# AdvPaint: Protecting Images from Inpainting Manipulation via Adversarial Attention Disruption

**Joonsung Jeon, Woo Jae Kim, Suhyeon Ha, Sooel Son**[*] **& Sung-Eui Yoon**[*]
Korea Advanced Institute of Science and Technology (KAIST)
{mikeraph,wkim97,suhyeon.ha,sl.son,sungeui}@kaist.ac.kr

## Abstract

The outstanding capability of diffusion models in generating high-quality images poses significant threats when misused by adversaries. In particular, we assume malicious adversaries exploiting diffusion models for inpainting tasks, such as replacing a specific region with a celebrity. While existing methods for protecting images from manipulation in diffusion-based generative models have primarily focused on image-to-image and text-to-image tasks, the challenge of preventing unauthorized inpainting has been rarely addressed, often resulting in suboptimal protection performance. To mitigate inpainting abuses, we propose AdvPaint, a novel defensive framework that generates adversarial perturbations that effectively disrupt the adversary's inpainting tasks. AdvPaint targets the self- and cross-attention blocks in a target diffusion inpainting model to distract semantic understanding and prompt interactions during image generation. AdvPaint also employs a two-stage perturbation strategy, dividing the perturbation region based on an enlarged bounding box around the object, enhancing robustness across diverse masks of varying shapes and sizes. Our experimental results demonstrate that AdvPaint's perturbations are highly effective in disrupting the adversary's inpainting tasks, outperforming existing methods; AdvPaint attains over a 100-point increase in FID and substantial decreases in precision. The code is available at https://github.com/JoonsungJeon/AdvPaint.

## 1 Introduction

The advent of diffusion models (Ho et al., 2020; Song et al., 2020; Rombach et al., 2022) and their applications has enabled the generation of a plethora of highly realistic and superior-quality images. For image-to-image tasks (Rombach et al., 2022), users input an image into a diffusion model to generate a modified version that aligns with a specified prompt. In inpainting tasks (Rombach et al., 2022), a diffusion model takes an input image with a masked region and replaces the masked area with new content that reflect a given prompt.

Meanwhile, the technical advancements in diffusion models have also posed significant threats of abuse due to their potential misuse. Unauthorized usage of diffusion models has raised copyright infringement concerns (Chatfield, 2023; BBC News, 2023) and has been exploited to spread fabricated content in fake news distributed across the Internet and social media platforms (Bloomberg, 2024). To mitigate this abuse, previous research has explored leveraging adversarial perturbations injected into images under protection. These perturbations aim to disrupt subsequent image manipulation tasks involving diffusion models.

However, the challenge of protecting images from inpainting abuses in diffusion models has received little attention in prior research. We posit that crafting adversarial perturbations to disrupt adversaries' inpainting tasks remains a challenging task. Figure 1 shows that prior defensive methods of injecting adversarial perturbations (Salman et al., 2023; Liang et al., 2023; Liang & Wu, 2023; Xu et al., 2024; Xue et al., 2024) provide insufficient protection, allowing adversaries to successfully perform inpainting despite the applied defenses.

---

[*]Co-corresponding authors

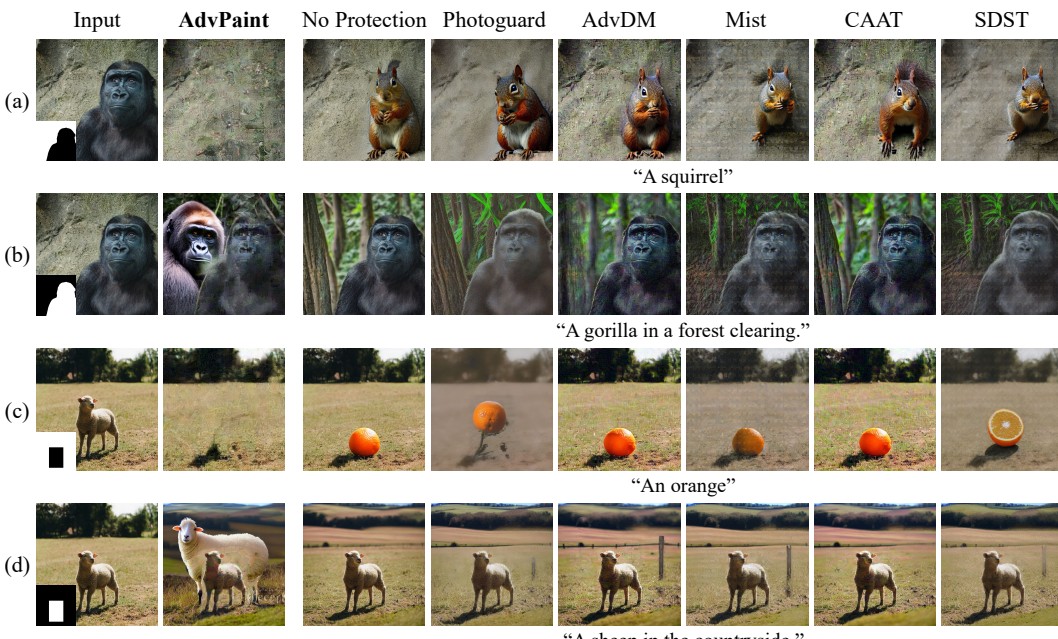

|  | Input | **AdvPaint** | No Protection | Photoguard | AdvDM | Mist | CAAT | SDST |

(a) "A squirrel"

(b) "A gorilla in a forest clearing."

(c) "An orange"

(d) "A sheep in the countryside."

Figure 1: Our proposed method effectively degrades the result images against various inpainting manipulations with huge spatial differences (*e.g.* removing objects or inserting new objects). The state-of-the-art adversarial examples show limitations in protecting input images, as the generated outputs still harmonize with the prompts. We apply (a) a segmentation mask $m^{seg}$, (b) its inverse, (c) a bounding box mask $m^{bb}$, (d) and its inverse, respectively.

We attribute this insufficient protection to the intrinsic nature of inpainting tasks, which leverage a mask for targeted image manipulation. When adversaries conduct foreground inpainting—replacing a *masked* region in an input image with new content specified by a prompt—the perturbations in the *unmasked* area should significantly disrupt this process. Similarly, when an inverted mask is applied for background inpainting, the perturbations in the *unmasked* foreground should disrupt the generation process in the *masked* background.

However, prior perturbation methods are designed to protect against whole-image manipulations by applying a single perturbation across the entire image. This approach is ineffective for inpainting tasks, where only the perturbations in the *unmasked* regions remain, leaving the *masked* areas vulnerable to manipulation by the adversary. Moreover, previous adversarial objectives have overlooked disrupting the diffusion model's ability to understand the semantics and spatial structure of the image and its conditioning prompts. Instead, they focused on finding shortcuts to make the latent representation of their adversarial examples similar to specific target representations, limiting their effectiveness in inpainting tasks.

In this work, we propose ADVPAINT, a novel defensive framework designed to protect images from inpainting tasks using diffusion models. ADVPAINT crafts imperceptible adversarial perturbations for an image by optimizing the perturbation to disrupt the attention mechanisms of a target inpainting diffusion model. Specifically, ADVPAINT targets both the cross-attention and self-attention blocks in the inpainting model, ensuring that the perturbation disrupts both foreground and background inpainting tasks. Unlike prior works that focused on latent space manipulation, ADVPAINT takes a direct approach to the generation process, ensuring that perturbations in the *unmasked* region effectively disrupt the inpainting process in the *masked* region.

The key idea of ADVPAINT is to maximize the differences in the components of the self- and cross-attention blocks between a clean image and its corresponding adversarial example, while ensuring the added perturbation remains imperceptible. For cross-attention blocks, ADVPAINT perturbs the query input to break the alignment between the latent image representation and external inputs. For self-attention blocks, ADVPAINT perturbs the three key elements—query, key, and value—to disrupt the inpainting model's ability to learn semantic relationships within the input image.

Moreover, ADVPAINT divides the perturbation regions into two using an enlarged bounding box around the object in the image and applies distinct perturbations inside and outside the box. This approach ensures that the adversarial examples enhance protection and remain robust to varying mask shapes, making the perturbations more effective and adaptable across diverse inpainting scenarios.

We demonstrate the effectiveness of our approach for both background and foreground inpainting tasks using masks varying in shapes and sizes, achieving superior performance compared to previous adversarial examples. Additionally, we conduct experiments on adversarial examples optimized with three objective functions from prior works, and validate the efficacy of our attention mechanism-based approach.

In summary, our contributions are as follows:

- We propose the first adversarial attack method designed to disrupt the attention mechanism of inpainting diffusion models, improving the protection of target images against inpainting manipulation abuses by adversaries.
- We introduce a method for dividing the perturbation region into two, based on the enlarged bounding box around the object in an image under protection, which contributes to ADVPAINT remaining effective across diverse adversarial scenarios exploiting various masks and inpainting types.
- We conduct extensive evaluations on ADVPAINT and demonstrate its superior performance compared to other baselines, improving FID by over a 100-point, and building upon existing methods by further enhancing protection specifically against abusive inpainting tasks.

## 2 RELATED STUDIES

### 2.1 ADVERSARIAL ATTACK

Goodfellow et al. (2014) highlighted the vulnerability of neural networks to adversarial examples—inputs perturbed to induce misclassification. Kurakin *et al.* extended this by introducing the Basic Iterative Method (Kurakin et al., 2016), which applies small perturbations iteratively, generating stronger adversarial examples. This iterative approach sparked further advancements in attack strategies, leading to more sophisticated methods and defenses. One notable advancement in this domain is Projected Gradient Descent (PGD) (Madry et al., 2017), a more robust iterative variant of FGSM. PGD includes a projection step that ensures the perturbed example $x'_i$ remains within a bounded $\eta$-ball around the original input $x$. Specifically, at each iteration $i$, the perturbed input is updated as follows:

$$x'_{i+1} = \text{Proj}_\eta(x'_i + \alpha \cdot \text{sign}(\nabla_{x'_i} J(\theta, x'_i, y)), \tag{1}$$

where $\alpha$ is the step size, $\text{Proj}_\eta$ enforces the constraint within the $\eta$-ball, $\nabla_{x'_i} J(\theta, x'_i, y)$ is the gradient of the loss function $J$, $\theta$ are the model parameters, and $y$ is the ground truth label. This refinement of FGSM via iterative updates and projection has become a representative method for generating adversarial examples.

### 2.2 DIFFUSION-BASED IMAGE GENERATION AND MANIPULATION

Ho et al. (2020) and Song et al. (2020) laid the foundation for modern diffusion models, with significant advancements like the latent diffusion model (LDM) introduced by Rombach et al. (2022). LDMs improve computational efficiency by encoding images $x \in \mathbb{R}^{3 \times H \times W}$ into latent vectors $z_0 \in \mathbb{R}^{4 \times h \times w}$ in lower dimension by the encoder $\mathcal{E}$. This reduction in dimensionality reduces computational cost while maintaining the model's ability to generate high-quality images.

LDMs consist of two processes: a forward process and a sampling process. In the forward process, Gaussian noise $\epsilon$ is incrementally added to the latent $z_0$ across timestep $t$, transforming it into pure Gaussian noise at $t = T$. The forward process results in a Markov Chain and can be expressed as $z_t = \sqrt{\bar{\alpha}_t} z_0 + \sqrt{1 - \bar{\alpha}_t} \epsilon$ where $\bar{\alpha}_t = \prod_{i=1}^{t} \alpha_i$ is pre-scheduled noise level and $\epsilon \sim \mathcal{N}(0, \mathbb{I})$. The reverse process, or sampling, denoises $z_t$ back to $z'_0$, using a noise prediction model (often a U-Net) trained to predict the added noise at each $t$. The training objective for the noise prediction is to minimize the following:

$$\mathcal{L}_{noise} = \mathbb{E}_{z_0,t,\epsilon \sim \mathcal{N}(0,\mathbb{I})} \left[ \|\epsilon - \epsilon_\theta(z_{t+1}, t)\|_2^2 \right], \tag{2}$$

where $\theta$ represents the parameters of the denoiser $\epsilon_\theta$. During sampling, the denoised latent at each timestep is computed via $z'_{t-1} = (1/\sqrt{\alpha_t})\left(z'_t - (1-\alpha_t)/\sqrt{1-\bar{\alpha}_t} \cdot \epsilon_\theta(z'_t, t)\right) + \sigma_t \epsilon$, where $\sigma_t$ controls the variance of the noise added back at each $t$, ensuring stochasticity. After the sampling process, the final latent $z'_0$ is decoded back into the image space via a decoder $\mathcal{D}$.

In addition to the default LDM, Rombach et al. (2022) introduced an inpainting-specific variant of the U-Net denoiser (see Appendix A.2.2). This model takes as input the original image $x$, the mask $m$, and the masked image $x^m = x \otimes m$, where $\otimes$ represents element-wise multiplication. The same encoder $\mathcal{E}$ is employed to create latent vectors $z_0$ and $z_0^m$ from both $x$ and $x^m$. The denoiser then takes as input a concatenation of the latent variable $z_t$, the masked latent $z_0^m$, and the resized mask $m' \in \mathbb{R}^{1 \times h \times w}$ at each $t$. Here, $z_0^m$ and $m'$ are inserted as input for every $t$, only denoising $z_t$ from $t = T$ to 0. This structure enables effective reconstruction of *masked* regions while preserving coherence with the surrounding *unmasked* areas. The loss function for the inpainting task is modified as minimizing the following:

$$\mathcal{L}_{noise} = \mathbb{E}_{z_0, z_0^m, m', t, \epsilon \sim \mathcal{N}(0,\mathbb{I})} \left[ \|\epsilon - \epsilon_\theta(z_{t+1}, z_0^m, m', t)\|_2^2 \right]. \tag{3}$$

## 2.3 ADVERSARIAL PERTURBATIONS TO PREVENT UNAUTHORIZED IMAGE USAGE

Malicious actors are certainly able to exploit the capabilities of diffusion models to generate high-quality and authentic-looking images. To mitigate such abuses, prior studies have explored methods for injecting imperceptible perturbations into images. These perturbations are designed to disrupt the image synthesis process, preventing diffusion models from effectively manipulating these perturbed images, thus protecting them against unauthorized and harmful usage. Several recent approaches have focused on protecting images from improper manipulation by leveraging *latent* representations in generative models. PhotoGuard (Salman et al., 2023) and Glaze (Shan et al., 2023) are designed to minimize the distance in latent space between the encoder output, $\mathcal{E}(x + \delta)$, and a target latent representation $z_{trg}$. The objective is to minimize the following:

$$\mathcal{L}_{latent} = \|z_{trg} - \mathcal{E}(x + \delta)\|_2^2, \tag{4}$$

where $\delta$ represents the perturbation applied to the image $x$ to ensure its encoded representation shifts towards the target latent vector $z_{trg}$.

Additionally, recent studies have focused on generating adversarial examples by utilizing the predicted noise from LDMs within the latent space. Anti-Dreambooth (Le et al., 2023) and AdvDM (Liang et al., 2023) adopt $\max_\delta$ Equation 2 to target text-to-image models. MetaCloak (Liu et al., 2023) introduces a meta-learning framework alongside $\max_\delta$ Equation 2 to address suboptimal optimization and vulnerability to data transformations. Mist (Liang & Wu, 2023) combines the two objectives – $\max_\delta$ Equation 2 and $\min_\delta$ Equation 4 – to strengthen image protection. CAAT (Xu et al., 2024) utilizes $\max_\delta$ Equation 2 for the optimization of its perturbation, and also finetunes the weights of key and value in the cross-attention blocks. Xue et al. (2024) propose SDST that enhances the efficiency of optimization by incorporating score distillation sampling (SDS) (Poole et al., 2022), which is applied alongside minimizing Equation 4.

## 3 PROBLEM STATEMENT

**Threat model.** We assume a malicious adversary who attempts to manipulate a published image by conducting inpainting tasks. The adversary's goal is to perform foreground or background inpainting using publicly available LDMs. They are motivated to fabricate contents using published images to spread fake news, potentially involving important public figures, or to infringe on the intellectual property of published artistic images.

We tackle a research question of how to compose an adversarial example for a given input image that effectively disrupts inpainting tasks abused by the adversary. Previous studies (Salman et al., 2023; Liang et al., 2023; Liang & Wu, 2023; Xu et al., 2024; Xue et al., 2024) have explored

diverse ways of crafting adversarial perturbations that disrupt adversaries' image-to-image and text-to-image tasks. However, we argue that these adversarial attacks are insufficient for protecting images from inpainting tasks.

Inpainting tasks inherently involve leveraging a mask for a specific target region to replace the *masked* area with desired prompts. However, in this process, the applied mask removes the perturbations embedded in the protected image, rendering them ineffective in disrupting the adversary's ability to inpaint the protected area. For example, the inpainted "orange" in AdvDM (Figure 1) exhibits a noisy background in the *unmasked* regions, while the perturbation has no impact on the generated "orange" object itself. We observed a similar limitation in Figure 1 "for every prior adversarial works", where the embedded perturbation only undermines inpainting tasks for the unmasked regions in protected images. This limitation arises because prior methods focus on optimizing objective functions (Equation 2 and 4) that seek shortcuts to align the latent representation of adversarial examples with their target representations.

Thus, this limitation presents a technical challenge: adversarial perturbations in the *unmasked* background should undermine the generation process in the *masked* area, thereby disrupting foreground inpainting. The same challenge applies to background inpainting with an inverted mask, where perturbations only on the *unmasked* foreground should disrupt the generation of the background.

To overcome this challenge, we propose two novel methods for generating adversarial perturbations: (1) generating perturbations that disrupt the attention mechanism of inpainting LDMs, and (2) applying distinct perturbations for a region covering a target object and the surrounding backgrounds outside that region.

These approaches work in tandem to effectively disrupt both foreground and background inpainting tasks. Section 4.1 describes our adversarial objectives that simultaneously disrupt cross-attention and self-attention mechanisms of inpainting LDMs. In Section 4.2, we describe the optimization strategy, which applies distinct perturbations to regions inside and outside the bounding boxes encompassing target objects, ensuring robustness across various mask shapes.

# 4 METHODOLOGY

## 4.1 ADVERSARIAL ATTACK ON ATTENTION BLOCKS

We propose ADVPAINT, a novel defensive framework to protect images from unauthorized inpainting tasks using LDMs. ADVPAINT generates an adversarial perturbation specifically designed to fool LDM-based inpainting model by disrupting their ability to capture correct attentions. These perturbations are designed to fundamentally destroy the model's image generation capability by tampering the cross- and self-attention mechanisms simultaneously.

The attention mechanism in the U-net denoiser $\epsilon_\theta$ of an LDM reconstructs an image from the latent vector via a denoising process. As depicted in Figure 2, each attention block–a sequence of residual, self-attention, and cross-attention blocks–is repeated after each down-sampling and upsampling operation. The self-attention block takes three components: query $q$, key $k$, and value $v$, each of which is a linear transformation of an input image. The cross-attention blocks obtains $k$ and $v$ sourced from external conditioning inputs (*e.g.* prompts). For each layer $l$, $q$ and $k$ are then multiplied to form the attention map $\mathcal{M}$, which then weights $v$, producing the block output $\mathcal{A}$:

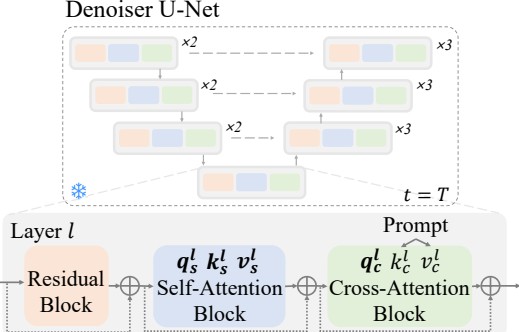

Figure 2: Attention mechanism in the U-Net denoiser. The bolded components in both blocks represent our target for disruption.

$$\mathcal{M} = \text{Softmax}\left(\frac{qk^T}{\sqrt{d}}\right), \quad \mathcal{A} = \mathcal{M} \cdot v. \tag{5}$$

Here, $d$ is the dimension of $q$ and $k$. Against such inpainting LDMs, we propose a novel optimization method for crafting adversarial perturbations that disrupt the functionalities of both self- and cross-

attention blocks. To attack cross-attention blocks, we design an objective function that disturbs the alignment between prompt tokens and their spatial positions in the input image. We aim to maximize the difference between the query $q$ of the clean image $x$ and that of the adversarial example $x + \delta$, thereby disrupting the cross-attention mechanism. Given $q = Q(\phi(x))$, where $\phi(\cdot)$ indicates the extracted features from the previous layer immediately before the self- or cross-attention block and $Q(\cdot)$ is the linear projection operator for $q$, we define the adversarial objective $\mathcal{L}_{cross}$ as follows:

$$\mathcal{L}_{cross} = \sum_l \left\| Q_c^l(\phi(x + \delta)) - Q_c^l(\phi(x)) \right\|^2 . \tag{6}$$

Here, $l$ denotes the $l$-th layer in the denoiser, and $c$ refers the cross-attention block. By pushing the query $q$ of $x + \delta$ away from that of $x$, this objective interferes the alignment with the key $k$ and value $v$, both of which are derived from the prompt conditions.

We propose another adversarial objective function that specifically targets the self-attention blocks. Unlike $\mathcal{L}_{cross}$, which only attacks $q$ that interacts with external conditioning inputs, this objective targets all input components—$q$, $k$, and $v$—in the self-attention block. This objective function $\mathcal{L}_{self}$ in Equation 7 is designed to maximize the difference between these three components of $x$ and $x + \delta$:

$$\mathcal{L}_{self} = \sum_l \Bigg( \left\| Q_s^l(\phi(x + \delta)) - Q_s^l(\phi(x)) \right\|^2$$
$$+ \left\| K_s^l(\phi(x + \delta)) - K_s^l(\phi(x)) \right\|^2 + \left\| V_s^l(\phi(x + \delta)) - V_s^l(\phi(x)) \right\|^2 \Bigg). \tag{7}$$

Here, $K(\cdot)$ and $V(\cdot)$ are linear projectors for $k$ and $v$, respectively, in the self-attention block $s$, where $k = K(\phi(x))$ and $v = V(\phi(x))$. By maximizing the difference across all components, we aim to disrupt the model's ability to interpret the semantics and spatial structure of the given image.

## 4.2 Separate Perturbations for Masked and Unmasked Regions

We also suggest an additional defensive measures of applying separate perturbations for possible objects that the adversary targets and their surrounding backgrounds. Specifically, given a target image to protect, ADVPAINT first divides the image into two regions—foreground and background–and applies distinct perturbations to ensure robust protection against masks of varying sizes and shapes. ADVPAINT identifies target objects that the adversary may target using Grounded SAM (Ren et al., 2024). To achieve this, ADVPAINT leverages a bounding box $m^{bb}$ around the identified objects. To fully cover the objects, this bounding box is then expanded by a factor of $\rho$ to form a new mask, $m$, by increasing its height and width while keeping the center coordinates of $m^{bb}$ intact. Using the two masks, $m$ and $1 - m$, ADVPAINT computes two separate perturbations for the regions inside and outside the boundary of $m$ based on the adversarial objective in Equation 8.

$$\delta := \arg \max_{\|\delta\|_\infty \leq \eta} \mathcal{L}_{attn} = \arg \max_{\|\delta\|_\infty \leq \eta} \left( \mathcal{L}_{cross} + \mathcal{L}_{self} \right). \tag{8}$$

In contrast to the single-stage perturbation approach used in prior works, where a single perturbation is applied to the entire image, our two-stage strategy provides enhanced protection and robustness against diverse mask configurations, as demonstrated in Section 5.5 and 5.6.

## 5 Experiments

### 5.1 Experimental Setup

We evaluate our proposed method on the pre-trained inpainting model from Stable Diffusion (Rombach et al., 2022), referred to as *SD inpainter* in this experiment. This model is widely used both in academia (*e.g.* Yu et al. (2023b); Xue et al. (2024)) and by the public[1] (*e.g.* von Platen et al. (2022)). Following prior studies (Salman et al., 2023; Liang et al., 2023; Xue et al., 2024), we collected 100 images from publicly available sources[2][3], which were then cropped and resized to $512 \times 512$

---

[1] https://huggingface.co/runwayml/stable-diffusion-inpainting
[2] https://www.pexels.com/
[3] https://unsplash.com/

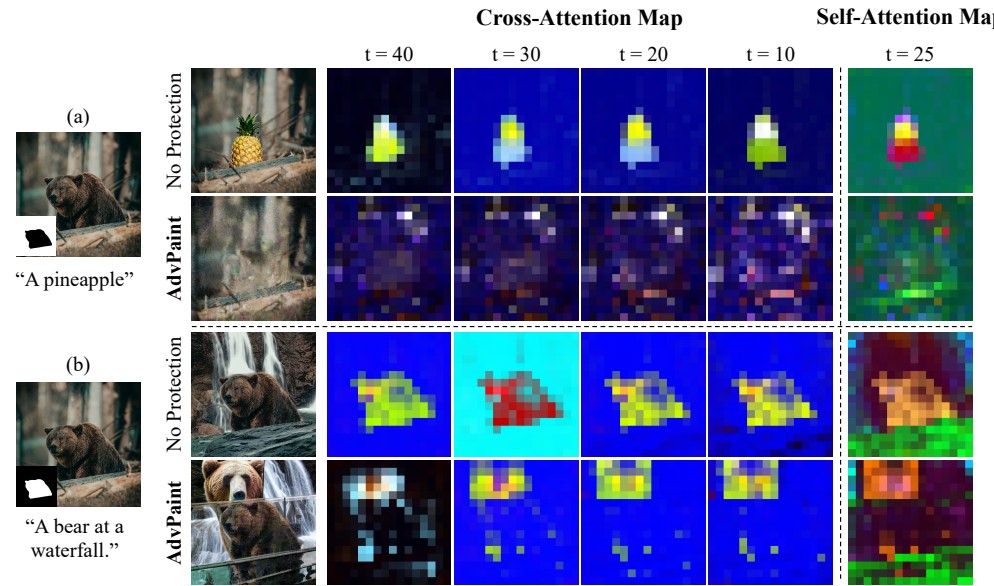

Figure 3: Visualization of cross- and self- attention maps for (a) foreground and (b) background inpainting manipulations. Our proposed method redirects the model's attention to other regions of the image, as shown in (a), while focusing attention on the newly generated object in (b).

resolutions. We applied Grounded SAM (Ren et al., 2024) to generate masks of various shapes and sizes. In computing adversarial perturbations, we enlarged the generated bounding box $m^{bb}$ to $m$ by a factor of $\rho = 1.2$, separating the regions for two-stage optimization. For text conditions, we generated 50 random prompts using ChatGPT (OpenAI, 2024). For example, {noun} and {noun, preposition, location} are randomly generated for foreground and background tasks, respectively. We applied Projected Gradient Descent (PGD) to optimize our perturbations exclusively at timestep $T$, over 250 iterations, starting with an initial step size of 0.03, which progressively decreased at each step. Importantly, we set $\eta$ as 0.06 for all adversarial examples, including those from prior works, to enforce consistent levels of imperceptible perturbations. All experiments were conducted using a single NVIDIA GeForce RTX 3090 GPU. Further implementation details can be found in the Appendix A.1.

## 5.2 ATTENTION MAPS

Figure 3 visualizes the cross- and self-attention maps for both the unprotected image $x$ and the adversarial example $x + \delta$ during the inpainting process. We compare foreground and background inpainting tasks under different textual conditions with the segmentation mask $m^{seg}$. For the prompts used to produce cross-attention maps, we used "pineapple" and "bear", respectively. The attention maps are visualized using PCA at a specific timestep $t$ within $T = 50$ inference steps.

In (a) foreground inpainting, the cross-attention map consistently focuses on the "pineapple" during the generation process of $x$, but our perturbation redirects attention to other areas of the unmasked region. Our method prevents the model from generating "pineapple", producing *nothing in the masked region*. In the self-attention map, the model shows a scattered image structure and semantics, indicating successful distraction. On the other hand, in (b) background inpainting, our perturbation causes the model to generate a new "bear" in the masked region, disregarding the original "bear" in the image. As shown in the cross-attention map, the prompt tokens fail to recognize the original object, leading to *the generation of a new "bear"*. Additionally, the self-attention block is tricked into overlooking the existing "bear", focusing instead on the newly generated one. These distorted attention maps demonstrate the effectiveness of our objective $\mathcal{L}_{attn}$ by (1) disrupting the linkage between image features and prompt tokens in the cross-attention blocks and (2) impairing the semantic understanding in the self-attention blocks.

| Optimization Methods | Foreground Inpainting | | | | | | Background Inpainting | | | | | |
|---|---|---|---|---|---|---|---|---|---|---|---|---|
| | $m^{seg}$ | | | $m^{bb}$ | | | $m^{seg}$ | | | $m^{bb}$ | | |
| | FID ↑ | Prec ↓ | LPIPS ↑ | FID ↑ | Prec ↓ | LPIPS ↑ | FID ↑ | Prec ↓ | LPIPS ↑ | FID ↑ | Prec ↓ | LPIPS ↑ |
| Photoguard | 230.49 | 0.5244 | 0.6494 | 185.86 | 0.7212 | 0.6236 | 118.85 | 0.4332 | 0.4141 | 132.51 | 0.1844 | 0.5220 |
| AdvDM | 232.39 | 0.3030 | 0.5287 | 181.13 | 0.4794 | 0.5231 | 94.49 | 0.5772 | 0.3111 | 116.60 | 0.2420 | 0.4191 |
| Mist | 235.81 | 0.4590 | 0.5541 | 191.00 | 0.6490 | 0.5421 | 123.48 | 0.4004 | 0.3852 | 155.57 | 0.1602 | 0.5016 |
| CAAT | 232.83 | 0.3430 | 0.5274 | 181.21 | 0.5314 | 0.5192 | 98.22 | 0.5414 | 0.3199 | 118.68 | 0.2382 | 0.4182 |
| SDST | 212.90 | 0.5658 | 0.5042 | 174.85 | 0.7244 | 0.4994 | 112.17 | 0.4406 | 0.3841 | 133.15 | 0.2054 | 0.4809 |
| SD Inpainter + $\min_\delta$ Eq. 4 | 211.35 | 0.5644 | 0.5780 | 180.40 | 0.7214 | 0.5894 | 128.01 | 0.4006 | 0.4745 | 146.39 | 0.1374 | 0.5914 |
| SD Inpainter + $\max_\delta$ Eq. 3 | 224.81 | 0.3860 | 0.4705 | 199.37 | 0.5186 | 0.4878 | 116.60 | 0.4832 | 0.3844 | 142.37 | 0.2078 | 0.4795 |
| SD Inpainter + $\min_\delta$ Eq. 3 | 182.12 | 0.6124 | 0.5267 | 154.27 | 0.7560 | 0.5273 | 97.44 | 0.5852 | 0.386 | 107.43 | 0.2692 | 0.4902 |
| ADVPAINT | **347.88** | **0.0570** | **0.6731** | **289.63** | **0.1536** | **0.6762** | **219.07** | **0.2148** | **0.5064** | **303.90** | **0.0936** | **0.6105** |

Table 1: Quantitative comparison with existing methods and objectives on foreground and background inpainting tasks. Metrics include FID, Precision (Prec), and LPIPS for segmentation ($m^{seg}$) and bounding box ($m^{bb}$) masks.

## 5.3 COMPARISON WITH EXISTING METHODS ON INPAINTING TASKS

We evaluate the performance of state-of-the-art adversarial attack methods, originally designed for disrupting image-to-image and text-to-image tasks (Salman et al., 2023; Liang et al., 2023; Liang & Wu, 2023; Xu et al., 2024; Xue et al., 2024), on inpainting tasks. Qualitative results on inpainting manipulations using the segmentation mask $m^{seg}$ and the bounding box mask $m^{bb}$ are shown in Figure 1. For foreground inpainting ((a), (c)), ADVPAINT-generated adversarial examples block the creation of objects specified by the given prompts. In contrast, previous methods allow the inpainter model to generate the synthetic objects as described in the prompts. For background inpainting ((b), (d)), our perturbations successfully mislead the model to ignore the original object and generate a new one from the prompt, while previous methods produce high-quality backgrounds aligned with the prompt. This highlights the challenge for prior approaches, which are less effective at disrupting the generation process within the *masked* area. More qualitative results of ADVPAINT are represented in Figure 4 and Appendix A.8.

To quantitatively compare our method with prior adversarial examples, we assess the performance using Frechet Inception Distance (FID) (Heusel et al., 2018), Precision (Kynkäänniemi et al., 2019), and LPIPS (Zhang et al., 2018). FID and LPIPS measure the feature distance between input images and the generated images. Precision denotes the proportion of generated images that fall within the distribution of real images. We used AlexNet (Krizhevsky et al., 2012) to compute LPIPS. In adversarial example methods, high FID, low precision, and high LPIPS are preferred.

In Table 1, we report the FID, Precision, and LPIPS scores for inpainting tasks using $m^{seg}$ and $m^{bb}$ masks. ADVPAINT consistently outperforms the state-of-the-art methods across various mask types while maintaining the same level of perturbation budget across all adversarial examples. For instance, when assuming the adversary conducting background inpainting, ADVPAINT using $m^{bb}$ achieves an FID of 303.90, outperforming Mist, the second-best method, by 148.33. We attribute this improvement to the fundamental difference in adversarial objectives. ADVPAINT targets the attention blocks and disrupts the functionalities of the self-attention block (image features-to-features) and the cross-attention block (image features-to-prompt) even with partially cropped perturbations. In contrast, the previous works focus solely on latent space representations (as seen in Equation 2, 3 and 4), without directly influencing the denoiser U-Net during the generation process.

## 5.4 COMPARISON WITH PRIOR OBJECTIVES FOR INPAINTING TASK

We assess the effectiveness of adversarial objective functions from prior works on inpainting manipulations by replacing their default LDM (image-to-image or text-to-image) with inpainting-specialized LDM (*SD inpainter*).

Rows 7–9 in Table 1 show the experimental results for the modified versions of ADVPAINT, utilizing the adversarial objectives from prior work (*i.e.* Equation 2 and 4). These results indicate that prior objectives do not lead to significant improvements, even when applied with the same SD inpainter, yielding subpar performance across all metrics. This is likely because previous spatial objectives, which rely primarily on latent space representations, lose effectiveness when image perturbations are masked out before being processed by the inpainting model.

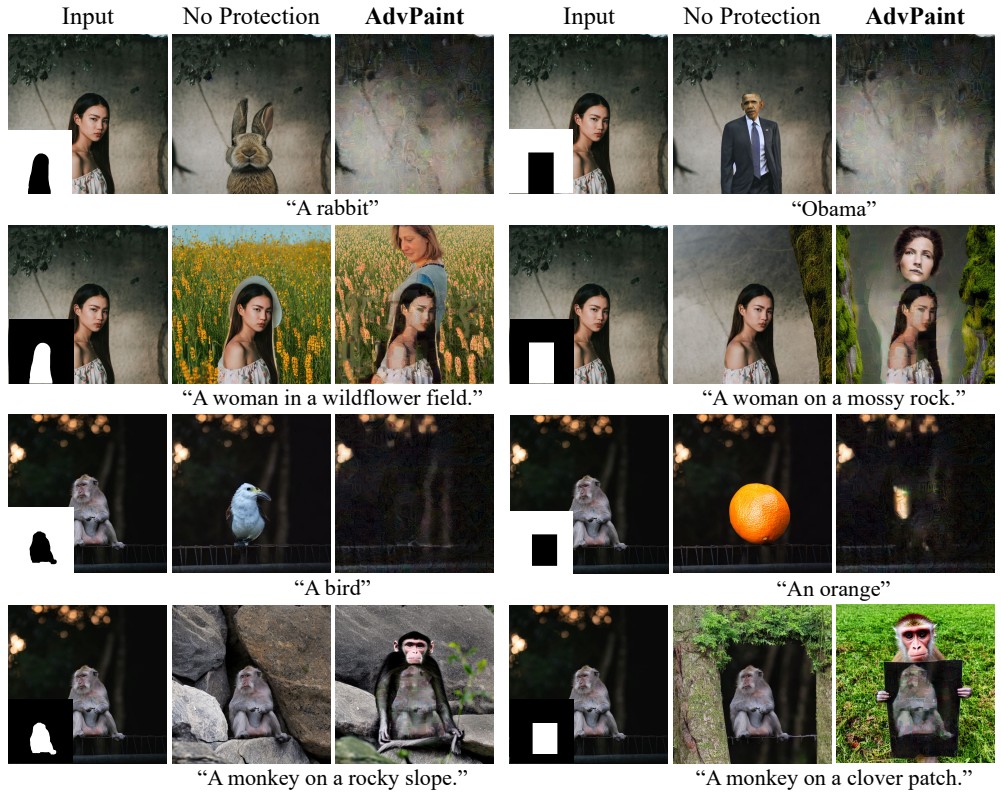

Figure 4: Our proposed adversarial examples on diverse inpainting types, masks, and prompts.

| | Foreground Inpainting | | | | | | Background Inpainting | | | | | |
|---|---|---|---|---|---|---|---|---|---|---|---|---|
| | $m^{seg}$ | | | $m^{bb}$ | | | $m^{seg}$ | | | $m^{bb}$ | | |
| Stage | FID ↑ | Prec ↓ | LPIPS ↑ | FID ↑ | Prec ↓ | LPIPS ↑ | FID ↑ | Prec ↓ | LPIPS ↑ | FID ↑ | Prec ↓ | LPIPS ↑ |
| 1 | 345.76 | 0.0628 | **0.6940** | 271.73 | 0.2056 | **0.6767** | 191.15 | 0.2418 | 0.4747 | 266.00 | 0.0938 | 0.5936 |
| 2 | **347.88** | **0.0570** | 0.6731 | **289.63** | **0.1536** | 0.6762 | **219.07** | **0.2148** | **0.5064** | **303.90** | **0.0936** | **0.6105** |

Table 2: Performance comparison according to optimization strategy. Combining our proposed objective with two-stage optimization consistently outperforms single-stage optimization.

## 5.5 EFFECTIVENESS OF SEPARATE PERTURBATIONS FOR IMAGE PROTECTION

We conduct quantitative evaluation to assess the effectiveness of ADVPAINT using separate perturbations based on the enlarged mask $m$, comparing it with the single perturbation approach. For the single-perturbation method, a white mask that covers *nothing* in the image is applied. Then, we optimize the single perturbation with the proposed objective $\mathcal{L}_{attn}$. Table 2 compares the protection effectiveness of this single-perturbation method with our approach, which utilizes two masks, $m$ and $1 - m$. The results demonstrate that using separate perturbations in ADVPAINT provides stronger image protection, yielding significant performance improvements across most metrics, regardless of the inpainting task type.

## 5.6 ROBUSTNESS OF ADVPAINT IN REAL-WORLD SCENARIO

Although ADVPAINT outperforms single perturbation methods, one might question whether AD-VPAINT remains effective when applied inpainting masks exceed the boundary. Thus, we further evaluate the robustness of ADVPAINT in real-world scenarios where masks are hand-crafted and exceed the boundaries of our optimization mask $m$. To simulate diverse user-defined masks, we randomly shift the original mask and consider two inpainting cases: one where the inference mask extends beyond $m$ and another where it remains within $m$. For randomly selected 25 images, we generate 10 segmentation masks per image and randomly shift them up, down, left, or right by a ran-

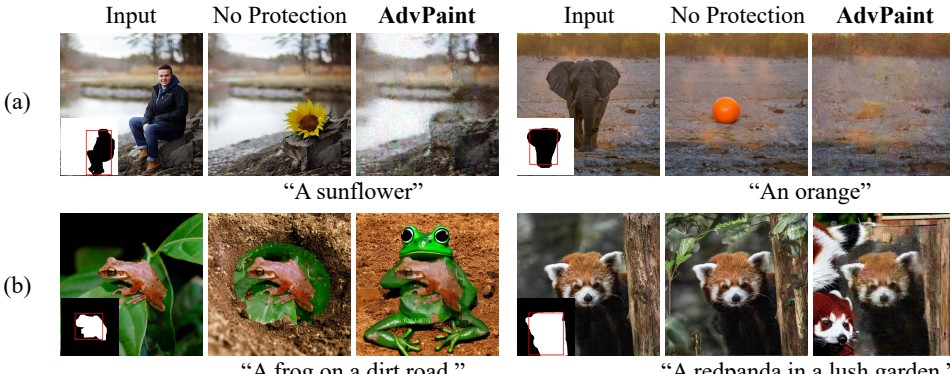

Figure 5: Results on image inpainting protection with exceeding masks. Hand-crafted binary masks depict real-world scenarios, with red bounding boxes indicating our optimization masks $m$. Examples of both (a) foreground and (b) background inpainting tasks are shown.

dom number of pixels, ensuring that at least one side exceeds the boundary, defining these as $m^{out}$. $m^{in}$ denotes masks that remain within the boundary after the same shifting process. Inpainting manipulations are then performed using 25 prompts for both foreground and background tasks.

| ADVPAINT | FG Inpainting | | | BG Inpainting | | |
|---|---|---|---|---|---|---|
| | FID ↑ | Prec ↓ | LPIPS ↑ | FID ↑ | Prec ↓ | LPIPS ↑ |
| $m^{in}$ | 294.91 | 0.0044 | 0.6743 | 225.3 | 0.0024 | 0.5754 |
| $m^{out}$ | 292.98 | 0.0058 | 0.6813 | 258.43 | 0.0036 | 0.6249 |

Table 3: Performance comparison of masks randomly shifted within the optimization boundary ($m^{in}$) and those exceeding the boundary ($m^{out}$).

Figure 5 shows the inpainting results of our approach using diverse, boundary-exceeding masks $m^{out}$, where our method successfully protects the image from both foreground and background inpainting. Table 3 compares the quantitative results of our approach in both $m^{in}$ and $m^{out}$ mask settings. This demonstrates ADVPAINT's robust protection in real-world scenarios, maintaining strong performance even when user-defined masks exceed the optimization boundaries.

## 5.7 DISCUSSION

**Transferability.** To demonstrate the transferability of our adversarial examples, we compare the effectiveness of ADVPAINT on image-to-image and text-to-image tasks using diffusion models with prior works in Appendix A.3. We observe that ADVPAINT exhibits comparable protections against image-to-image and text-to-image tasks, on par with the performance of prior methods that are solely designed for these tasks. We note that ADVPAINT is specifically designed for inpainting protection.

**Multi-object images.** We further evaluate the efficacy of ADVPAINT on images containing multiple objects by targeting the attention blocks for each object, as shown in Appendix A.8. We first optimize perturbations within each object's mask $m$ and then apply perturbations to the remaining background. ADVPAINT remains effective regardless of the number of objects; however, the computational cost increases as the number of target objects increases. We leave addressing these computational overheads for future work.

## 6 CONCLUSION

In this paper, we present a novel image protection perturbation designed to defend against inpainting LDMs, which can replace masked regions with highly realistic objects or backgrounds. We are the first to bring attention to the dangers of inpainting tasks in image abuse and demonstrate the limitations of previous adversarial approaches in providing sufficient protection. To address the challenge of preventing malicious alterations in *masked* regions with limited perturbations, ADVPAINT introduces attention disruption and a two-stage optimization strategy. By directly targeting the cross- and self-attention blocks, and optimizing separate perturbations for different object regions, ADVPAINT outperforms state-of-the-art methods in preventing inpainting manipulations. Additionally, ADV-PAINT exhibits robustness to various hand-crafted masks, demonstrating its practical applicability in real-world scenarios.

ACKNOWLEDGEMENTS

We would like to thank the anonymous reviewers for their constructive comments and suggestions. This work was supported by the National Research Foundation of Korea(NRF) grant funded by the Korea government(MSIT) (No. RS-2023-00208506) and the Institute of Information & Communications Technology Planning & Evaluation(IITP) grant funded by the Korea government (MSIT) (No. RS-2020-II200153, Penetration Security Testing of ML Model Vulnerabilities and Defense). Prof. Sung-Eui Yoon and Prof. Sooel Son are co-corresponding authors.

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
