# A APPENDIX

## A.1 IMPLEMENTATION DETAILS

### A.1.1 ALGORITHM OF ADVPAINT

---

**Algorithm 1** ADVPAINT

1: **Input:** Clean image $x$, perturbation $\delta$, mask set $M$, extracted feature $\phi$, optimization steps $N$, step size $\alpha$, perturbation budget $\eta$, total of $L$ layers in U-Net, timestep $T$
2: **Output:** Adversarial example $x'$
3: Initialize $\delta \sim \mathcal{U}(-\eta, \eta)$
4: $x' \leftarrow x + \delta$
5: **for** mask **in** $M$ **do**
6:     $m \leftarrow$ mask
7:     $x_0 \leftarrow x \otimes m$
8:     $x'_0 \leftarrow x' \otimes m$
9:     **for** $i = 0$ **to** $N-1$ **at timestep** $T$ **do**     $\triangleright$ Optimization is performed at timestep $T$ only
10:         **for** $l = 1$ **to** $L$ **do**
11:             $(q_s^l, k_s^l, v_s^l) \leftarrow (Q_s^l(\phi(x)), K_s^l(\phi(x)), V_s^l(\phi(x)))$
12:             $(q_s'^l, k_s'^l, v_s'^l) \leftarrow (Q_s^l(\phi(x'_i)), K_s^l(\phi(x'_i)), V_s^l(\phi(x'_i)))$
13:             $q_c^l, q_c'^l \leftarrow Q_c^l(\phi(x)), Q_c^l(\phi(x'_i))$
14:         **end for**
15:         $\mathcal{L}_{attn} \leftarrow \sum_l \left( \left\| q_s'^l - q_s^l \right\|^2 + \left\| k_s'^l - k_s^l \right\|^2 + \left\| v_s'^l - v_s^l \right\|^2 \right) + \sum_l \left( \left\| q_c'^l - q_c^l \right\|^2 \right)$
16:         $\delta \leftarrow \delta + \alpha \cdot \text{sign}(\nabla_{x'_i} \mathcal{L}_{attn})$
17:         $\delta \leftarrow \text{clip}(\delta, -\eta, \eta)$
18:         $x'_{i+1} \leftarrow x_0 + \delta$
19:     **end for**
20:     $x' \leftarrow x'_{N-1}$
21: **end for**

---

Algorithm 1 describes the perturbation generation process of ADVPAINT. Note that we optimize our perturbation only at timestep $T$, as considering additional timesteps significantly increase computational costs.

### A.1.2 PRIOR ADVERSARIAL METHODS

For all prior works used as our baselines (Salman et al., 2023; Liang et al., 2023; Liang & Wu, 2023; Xu et al., 2024; Xue et al., 2024), we follow their official implementations to optimize their perturbations. The only adjustment we made is to set the noise level by adjusting the hyperparameter $\eta$ to 0.06, ensuring that all methods operate under the same noise constraints. We note that all these baselines use PGD for optimizing their perturbations.

Several methods require setting a target latent for optimizing perturbations. For Photoguard (Salman et al., 2023), we use the zero vector as the target latent, which is their default setting. For Mist (Liang & Wu, 2023) and SDST (Xue et al., 2024), we use the target image of Mist for both implementations.

### A.1.3 THREAT MODEL

In this work, we evaluate our adversarial perturbations across a range of tasks, including inpainting, image-to-image, and text-to-image generation.

**Inpainting task:** We use the Stable Diffusion inpainting pipeline[4] provided by Diffusers (runwayml/stable-diffusion-inpainting). The default settings of the model are applied (inference step $T$=50, guidance scale=7.5, strength=1.0, etc.).

---

[4]https://huggingface.co/docs/diffusers/api/pipelines/stable_diffusion/inpaint

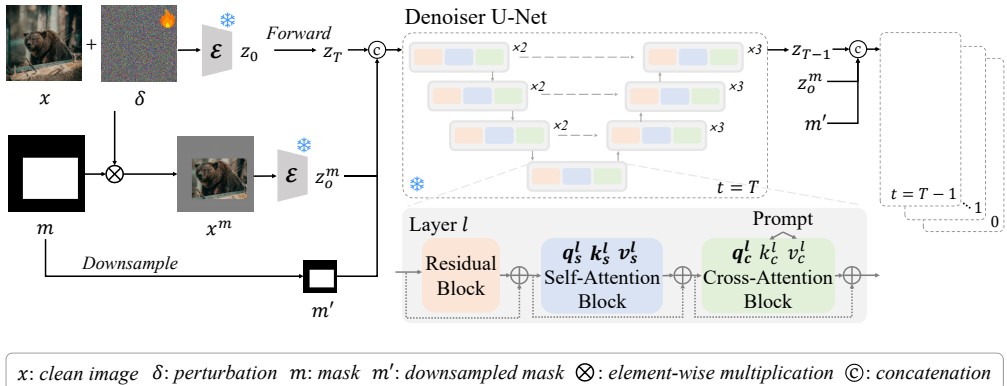

$x$: *clean image*  $\delta$: *perturbation*  $m$: *mask*  $m'$: *downsampled mask*  $\otimes$: *element-wise multiplication*  ©: *concatenation*

Figure 6: The architecture of the LDM denoiser specifically modified for inpainting tasks. The input image $x$ and the masked image $x^m$ share the same encoder $\varepsilon$. The latent $z_0^m$ and the resized mask $m'$ are fed into the model at every timestep. Here, $c$ denotes from cross-attention and $s$ stands for self-attention. We optimize the perturbation $\delta$ by targeting the bolded components in each block of each layer $l$.

**Image-to-image task:** For image-to-image translation, we use the Stable Diffusion image-to-image pipeline[5] provided by Diffusers (runwayml/stable-diffusion-v1-5). Specifically, we follow the default settings of the pipeline, where inference steps = 50, strength = 0.8, and guidance scale = 7.5.

**Text-to-image task:** We implement text-to-image generation using the Textual Inversion (Gal et al., 2022) model, following the official implementation and settings from the paper. Specifically, we set the inference steps to 50 and the guidance scale to 7.5. For the input images, where 3 to 5 images are required, we utilized the official dataset of DreamBooth (Ruiz et al., 2022). For both tasks, we used images of 512×512 size and randomly crafted the conditional prompts.

### A.1.4 GENERATING PROMPTS FOR OPTIMIZATION AND INPAINTING

In the process of generating adversarial perturbations using ADVPAINT, our target inpainting model requires a prompt input as an external condition. For simplicity, we manually set the prompt as a basic {noun} format (*e.g.*"A gorilla" for gorilla images, "A dog" for dog images).

In the inference phase, as described in 5.1, we generated 50 random prompts using ChatGPT (OpenAI, 2024). For foreground inpainting, the prompts followed the format of {noun} (*e.g.*"An orange", "A tiger"). For background inpainting, we generated prompts in the format of {preposition, location} (*e.g.*"at the riverside.", "at a wooden fence."), inserting the prompts used in the perturbation-generation step at the beginning of each generated prompt (*e.g.*"A gorilla at the riverside.", "A dog at a wooden fence."). We followed the prompt setup from Yu et al. (2023b) for fair comparisons.

### A.2 U-NET DENOISER MODIFIED FOR INPAINTING

### A.2.1 PRELIMINARY: ATTENTION BLOCKS IN LDMS

Rombach et al. (2022) proposed an LDM that leverages self- and cross-attention blocks. The self-attention blocks play a crucial role in generating high-dimensional images by capturing long-range dependencies between spatial regions of input images. Meanwhile, the cross-attention blocks are designed to align the latent image representation with external inputs, such as prompts, during the denoising process, ensuring that the generated image reflects the desired conditioning (Hertz et al., 2022; Tumanyan et al., 2022; Liu et al., 2024).

---

[5]https://huggingface.co/docs/diffusers/api/pipelines/stable_diffusion/img2img

| HD-Painter | Foreground Inpainting | | | | | | Background Inpainting | | | | | |
|---|---|---|---|---|---|---|---|---|---|---|---|---|
| | $m^{seg}$ | | | $m^{bb}$ | | | $m^{seg}$ | | | $m^{bb}$ | | |
| | FID ↑ | Prec ↓ | LPIPS ↑ | FID ↑ | Prec ↓ | LPIPS ↑ | FID ↑ | Prec ↓ | LPIPS ↑ | FID ↑ | Prec ↓ | LPIPS ↑ |
| Photoguard | 153.46 | 0.8552 | 0.5632 | 132.63 | 0.8962 | 0.5446 | 93.46 | 0.5978 | 0.3064 | 127.90 | 0.3246 | 0.4400 |
| AdvDM | 155.44 | 0.6180 | 0.4807 | 134.54 | 0.7032 | 0.4707 | 75.85 | 0.7278 | 0.2538 | 109.28 | 0.4738 | 0.3617 |
| SDST | 146.85 | 0.8568 | 0.4462 | 128.88 | 0.9038 | 0.4456 | 87.64 | 0.6042 | 0.2896 | 127.85 | 0.3366 | 0.4120 |
| **ADVPAINT** | **178.71** | **0.5350** | **0.5770** | **156.51** | **0.6276** | **0.5754** | **164.10** | **0.3310** | **0.3998** | **264.79** | **0.1748** | **0.5232** |

Table 4: Quantitative comparison for HD-Painter (Manukyan et al., 2023).

| DreamShaper | Foreground Inpainting | | | | | | Background Inpainting | | | | | |
|---|---|---|---|---|---|---|---|---|---|---|---|---|
| | $m^{seg}$ | | | $m^{bb}$ | | | $m^{seg}$ | | | $m^{bb}$ | | |
| | FID ↑ | Prec ↓ | LPIPS ↑ | FID ↑ | Prec ↓ | LPIPS ↑ | FID ↑ | Prec ↓ | LPIPS ↑ | FID ↑ | Prec ↓ | LPIPS ↑ |
| Photugard | 188.08 | 0.7422 | 0.5878 | 157.90 | 0.8544 | 0.5792 | 103.74 | 0.6112 | 0.3361 | 131.61 | 0.3074 | 0.4840 |
| AdvDM | 183.74 | 0.5114 | 0.5080 | 152.55 | 0.6540 | 0.5001 | 84.99 | 0.7102 | 0.2846 | 115.41 | 0.4340 | 0.3992 |
| SDST | 179.80 | 0.7792 | 0.4682 | 151.38 | 0.8656 | 0.4725 | 98.67 | 0.5954 | 0.3149 | 132.79 | 0.2978 | 0.4446 |
| **ADVPAINT** | **230.53** | **0.3856** | **0.6042** | **186.27** | **0.5196** | **0.6092** | **177.68** | **0.3160** | **0.4317** | **266.85** | **0.1622** | **0.5561** |

Table 5: Quantitative comparison for DreamShaper (DreamShaper, 2024).

| SD-2-Inp. | Foreground Inpainting | | | | | | Background Inpainting | | | | | |
|---|---|---|---|---|---|---|---|---|---|---|---|---|
| | $m^{seg}$ | | | $m^{bb}$ | | | $m^{seg}$ | | | $m^{bb}$ | | |
| | FID ↑ | Prec ↓ | LPIPS ↑ | FID ↑ | Prec ↓ | LPIPS ↑ | FID ↑ | Prec ↓ | LPIPS ↑ | FID ↑ | Prec ↓ | LPIPS ↑ |
| Photoguard | 239.73 | 0.5102 | 0.6226 | 199.21 | 0.7000 | 0.6071 | 110.33 | 0.4570 | 0.3798 | 126.74 | 0.1712 | 0.5094 |
| AdvDM | 249.57 | 0.1902 | 0.5393 | 199.22 | 0.3636 | 0.5246 | 89.66 | 0.5942 | 0.3027 | 114.39 | 0.2610 | 0.4197 |
| SDST | 231.96 | 0.5324 | 0.5001 | 201.65 | 0.6756 | 0.4996 | 106.61 | 0.4718 | 0.3569 | 130.00 | 0.1892 | 0.4710 |
| **ADVPAINT** | **325.14** | **0.0926** | **0.6452** | **264.72** | **0.2160** | **0.6443** | **198.32** | **0.2210** | **0.4633** | **267.91** | **0.0842** | **0.5756** |

Table 6: Quantitative comparison for Stable-Diffusion-2-Inpainting model.

### A.2.2 ARCHITECTURE OF INPAINTING LDM

We demonstrate the architecture of inpainting LDM in Figure 6. This model takes three inputs: the original image $x$, the mask $m$, and the masked image $x^m = x \otimes m$, where $\otimes$ represents element-wise multiplication. The shared encoder $\mathcal{E}$ produces two latent vectors, $z_0$ and $z_0^m$.

The denoiser U-Net consists of 16 layers, each comprising a sequence of residual, self-attention, and cross-attention blocks, along with skip connections. As in the default LDM, the denoiser predicts the noise added to the latent and denoises the latent $z_t$ at each timestep $t$. The resulting denoised latent $z_{t-1}$ is then concatenated with $z_0^m$ and $m'$ for the next denoising step. Note that After the denoising for inference steps $T$, the denoised latent $z_0'$ is then inserted to the same decoder $\mathcal{D}$ of default LDM to generate the inpainted image.

## A.3 TRANSFERABILITY OF ADVPAINT

### A.3.1 VARIANTS OF INPAINTING MODELS

We conducted extensive experiments on multiple inpainting model variants: HD-Painter (Manukyan et al., 2023), DreamShaper (DreamShaper, 2024), and the Stable Diffusion v2 inpainting model (Stability AI, n.d.). For all the variants, we follow the default settings of the official code and we also follow the default settings in the paper, only replacing the inpainting model to one of the variants.

In Table 4, 5, and 6, we evaluate ADVPAINT against these variants using FID, precision, and LPIPS metrics and compared it with other baseline protection methods. Even when the architecture differed significantly (e.g., HD-Painter) or when fine-tuning changed the model parameters (e.g., DreamShaper, SD-2-inpainting), ADVPAINT consistently outperform earlier protection methods across all metrics. Qualitative results are depicted in Figure 7, 8, and 9.

### A.3.2 IMAGE-TO-IMAGE AND TEXT-TO-IMAGE TASKS

We demonstrate the transferability of ADVPAINT to image-to-image and text-to-image tasks in Figure 10 and 11. While prior methods targeting these tasks effectively protect images from manipulations, our approach also delivers competitive safeguarding results.

### A.3.3 DIT-BASED GENERATION MODELS

DiT (Peebles & Xie, 2022) suggests a new paradigm in text-to-image generation tasks by applying vision transformers to Latent Diffusion Models, which decreases model complexity and increases generation quality. we evaluated the robustness of ADVPAINT against an adversary using the inpainting model of Flux (Labs, n.d.) and Stable Diffusion 3 (SD3) (Esser et al., 2024), and text-to-image model Pixart-$\delta$ (Chen et al., 2024) which leverages a diffusion transformer. Unlike Pixart-$\delta$, we note that models like DiT and Pixart-$\alpha$ (Chen et al., 2023) are designed for generating images solely from text prompts using diffusion transformer architectures, which make them unsuitable for our tasks that require accepting input images.

**Flux** provides an inpainting module based on multi-modal and parallel diffusion transformer blocks. We utilized the "black-forest-labs/FLUX.1-schnell" checkpoint and the image size was set to 512x512 to match our settings.

As shown in Figure 12, ADVPAINT effectively disrupts the inpainting process by causing misalignment between generated regions and unmasked areas. For example, it generates cartoon-style cows in (a) and adds a new rabbit in (b), while also producing noisy patterns in the unmasked areas of the images.

**SDS** is a text-to-image model built on the architecture of a Multi-modal DiT (MMDiT) and includes an inpainting pipeline, making it suitable for our experiments. We followed the official implementation, modifying only the image size to 512x512 to match our experimental settings.

As shown in the updated Figure 25, our results demonstrate the protective capabilities of AdvPaint against DiT-based inpainting tasks. Notably, we observed misalignment between generated images and unmasked regions. For instance, parts of a cat, lion, and watermelon are not fully generated and appear hidden behind the unmasked region in (a). In (b), which involves background inpainting tasks, the backgrounds are cartoonized, often disregarding pre-existing objects and generating new ones. This protective effect, which disrupts the semantic connection with unmasked objects, is also evident in the results for Flux.

**Chen et al.** have proposed Pixart-$\delta$ which incorporates DreamBooth (Ruiz et al., 2022) into DiT. We chose this work for the adversary's generative model since it supports feeding an input image along with a command prompt for performing generation.

As shown in Figure 13 (a), AdvPaint-generated perturbations consistently undermine the generation ability of Pixart-$\delta$. Furthermore, AdvPaint also renders noise patterns that degrade the image quality on the resulting output images of the diffusion model, which aligns with the behavior of previous methods (*i.e.* Photoguard, AdvDM, SDST).

ADVPAINT also effectively disrupts the original DreamBooth (Ruiz et al., 2022), as shown in Figure 13 (b). However, our findings indicate that ADVPAINT and the previous methods are less effective against Pixart-$\delta$ that leverages DiT, as shown in Figure 13 (a). Additionally, compared to the results of LDM-based inpainting models in Figure 1, current methods are less effective when applied to DiTs. Discernible objects are generated in the foreground inpainting tasks and new objects according to the prompts are not always generated. We believe this ineffectiveness stems from the distinct characteristic of DiT, which processes patchified latent representations. ADVPAINT and our baselines are specifically designed to target LDMs, which utilize the entire latent representation as input, allowing perturbations to be optimized over the complete latent space. Thus, when latents are patchified in DiTs, perturbations may become less effective at disrupting the model's processing, thereby diminishing their protective capability. This discrepancy necessitates further research to develop protection methods specifically tailored to safeguard images against the adversary misusing DiT-based models. For instance, optimizing perturbations at the patch level rather than across the entire latent representation could prove more effective in countering the unique paradigm of image generation in DiT-based models.

| Optim. Methods | (a) FG Inpainting | | | | | | (b) BG Inpainting | | | | | |
|---|---|---|---|---|---|---|---|---|---|---|---|---|
| | $m^{seg}$ | | | $m^{bb}$ | | | $m^{seg}$ | | | $m^{bb}$ | | |
| | FID ↑ | Prec ↓ | LPIPS ↑ | FID ↑ | Prec ↓ | LPIPS ↑ | FID ↑ | Prec ↓ | LPIPS ↑ | FID ↑ | Prec ↓ | LPIPS ↑ |
| Photoguard | 161.44 | 0.0874 | 0.6415 | 129.99 | 0.2158 | 0.6171 | 144.21 | 0.5230 | 0.4063 | 153.63 | 0.2280 | 0.5317 |
| AdvDM | 160.54 | 0.0658 | 0.5167 | 127.36 | 0.1266 | 0.5122 | 118.13 | 0.6228 | 0.3168 | 131.58 | 0.2720 | 0.4311 |
| SDST | 148.57 | 0.1340 | 0.4930 | 120.55 | 0.2456 | 0.4882 | 139.86 | 0.5112 | 0.3810 | 152.73 | 0.2280 | 0.4892 |
| ADVPAINT | **331.27** | **0.0036** | **0.6706** | **275.48** | **0.0264** | **0.6697** | **291.12** | **0.3490** | **0.4948** | **355.94** | **0.1152** | **0.6014** |

Table 7: Quantitative comparison with a diverse set of prompts that are likely candidates for foreground and background inpainting tasks. We set prompts as (a) {noun} that describes the *mask-covered* object and (b) {preposition, location}.

| IMPRESS | Foreground Inpainting | | | | | | Background Inpainting | | | | | | |
|---|---|---|---|---|---|---|---|---|---|---|---|---|---|
| | $m^{seg}$ | | | $m^{bb}$ | | | $m^{seg}$ | | | $m^{bb}$ | | | |
| | FID ↑ | Prec ↓ | LPIPS ↑ | FID ↑ | Prec ↓ | LPIPS ↑ | FID ↑ | Prec ↓ | LPIPS ↑ | FID ↑ | Prec ↓ | LPIPS ↑ | PSNR |
| Photugard | 182.62 | 0.6510 | 0.5564 | 151.05 | 0.7990 | 0.5522 | 106.54 | 0.4954 | 0.4333 | 118.30 | 0.2158 | 0.5361 | 28.5925 |
| AdvDM | 209.21 | 0.3764 | 0.5387 | 165.99 | 0.5708 | 0.5336 | 84.63 | 0.6132 | 0.3351 | 103.76 | 0.2734 | 0.4429 | 29.1283 |
| SDST | 199.28 | 0.6252 | 0.5307 | 164.13 | 0.7432 | 0.5271 | 104.75 | 0.4852 | 0.4124 | 121.10 | 0.2130 | 0.5090 | 28.8105 |
| ADVPAINT | **299.07** | **0.1614** | **0.6667** | **237.05** | **0.3300** | **0.6623** | **161.24** | **0.3230** | **0.4730** | **214.38** | **0.1360** | **0.5756** | 28.6303 |

Table 8: Quantitative evaluation of inpainting results after applying IMPRESS (Cao et al., 2023).

## A.4 DIFFERENT PROMPTS FOR VARIOUS INPAINTING TASKS

In real-world scenarios, the exact prompts used by adversaries to maliciously modify images remain unknown. To simulate and analyze potential attack vectors, we conduct experiments using a diverse set of prompts that are likely candidates for foreground and background inpainting tasks.

Please note that below experiments were conducted under the same default settings (*i.e.* using the Stable Diffusion Inpainting model with a total of 100 images and 50 prompts per image), ensuring a fair and consistent comparison.

### A.4.1 FOREGROUND INPAINTING

In the experiments throughout the paper, prompts follow the format of {noun} for foreground inpainting tasks. Here, we evaluate the robustness of ADVPAINT using a different kind of prompt: a prompt that describes the *mask-covered* object itself. For example, we used the prompt "A man" for an input image describing a male and performed an inpainting task to generate another male image.

As demonstrated in Table 7 (a), ADVPAINT successfully disrupted the adversary's inpainting task, resulting in the generation of an image with no discernible object. This is because the perturbation optimized to disrupt the attention mechanism successfully redirects the attention to other unmasked areas as explained in Section 5.2 and Figure 3. We demonstrate the qualitative examples in Figure 14.

### A.4.2 BACKGROUND INPAINTING

For background tasks throughout the paper, prompts follow the format of simple noun that describes the object in the image added to preposition, location. Here, we experiment with prompts where the noun describing the object is omitted and evaluate their effectiveness in undermining the adversary's background inpainting task. Specifically, we assumed the adversary might adjust the prompt to exclude the object (e.g., using "rocky slope" instead of "A monkey on a rocky slope") to mitigate artifacts. In all cases, ADVPAINT outperformed all baselines, as demonstrated in the Table 7 (b). Qualitative results are depicted in Figure 15.

## A.5 ROBUSTNESS OF ADVPAINT AGAINST PURIFICATION METHODS

We conducted experiments under the same settings as outlined in the paper (*i.e.* 100 images, 50 prompts per image, segmentation and bounding box masks, etc.) to evaluate the robustness of ADVPAINT against the recent purification techniques, including IMPRESS (Cao et al., 2023) and Honig et al. (2024). Please note that among the four suggested methods in Honig et al. (2024), we eval-

| Gaussian | Foreground Inpainting | | | | | | Background Inpainting | | | | | | |
|---|---|---|---|---|---|---|---|---|---|---|---|---|---|
| | $m^{seg}$ | | | $m^{bb}$ | | | $m^{seg}$ | | | $m^{bb}$ | | | |
| | FID ↑ | Prec ↓ | LPIPS ↑ | FID ↑ | Prec ↓ | LPIPS ↑ | FID ↑ | Prec ↓ | LPIPS ↑ | FID ↑ | Prec ↓ | LPIPS ↑ | PSNR |
| Photoguard | 185.20 | 0.6808 | 0.8665 | 156.79 | 0.7814 | 0.8382 | 127.17 | 0.4322 | 0.6111 | 136.26 | 0.1958 | **0.7659** | 20.1484 |
| AdvDM | 181.57 | 0.6730 | 0.8343 | 152.97 | 0.7864 | 0.8094 | 120.80 | 0.4460 | 0.5896 | 128.89 | 0.2084 | 0.7387 | 20.2824 |
| SDST | 185.04 | 0.6810 | 0.8507 | 154.38 | 0.7838 | 0.8228 | 123.37 | 0.4332 | 0.6006 | 135.07 | 0.2104 | 0.7546 | 20.2358 |
| ADVPAINT | **187.48** | **0.6682** | **0.8697** | **157.74** | **0.7804** | **0.8411** | **128.94** | **0.4056** | **0.6125** | **139.56** | **0.1820** | 0.7618 | 20.2410 |

Table 9: Quantitative evaluation of inpainting results after applying Gaussian Noise.

| Upscaling | Foreground Inpainting | | | | | | Background Inpainting | | | | | | |
|---|---|---|---|---|---|---|---|---|---|---|---|---|---|
| | $m^{seg}$ | | | $m^{bb}$ | | | $m^{seg}$ | | | $m^{bb}$ | | | |
| | FID ↑ | Prec ↓ | LPIPS ↑ | FID ↑ | Prec ↓ | LPIPS ↑ | FID ↑ | Prec ↓ | LPIPS ↑ | FID ↑ | Prec ↓ | LPIPS ↑ | PSNR |
| Photoguard | 136.96 | 0.8042 | 0.2476 | 111.39 | **0.8820** | 0.2562 | 60.49 | 0.8086 | 0.2639 | 62.65 | 0.5630 | 0.2842 | 30.2422 |
| AdvDM | **137.97** | 0.8078 | **0.3112** | **115.98** | 0.8844 | **0.3164** | 63.14 | 0.7886 | **0.2895** | 65.65 | 0.5428 | **0.3339** | 29.5016 |
| Mist | 136.57 | **0.8008** | 0.2474 | 112.77 | 0.8922 | 0.2576 | 61.18 | 0.7932 | 0.2632 | 64.92 | 0.5442 | 0.2823 | 30.0934 |
| ADVPAINT | 137.24 | 0.8132 | 0.2784 | 115.43 | 0.8844 | 0.2851 | **65.18** | **0.7782** | 0.2840 | **66.61** | **0.5376** | 0.3068 | 29.8244 |

Table 10: Quantitative evaluation of inpainting results after applying Upscaling method.

| JPEG | Foreground Inpainting | | | | | | Background Inpainting | | | | | | |
|---|---|---|---|---|---|---|---|---|---|---|---|---|---|
| | $m^{seg}$ | | | $m^{bb}$ | | | $m^{seg}$ | | | $m^{bb}$ | | | |
| | FID ↑ | Prec ↓ | LPIPS ↑ | FID ↑ | Prec ↓ | LPIPS ↑ | FID ↑ | Prec ↓ | LPIPS ↑ | FID ↑ | Prec ↓ | LPIPS ↑ | PSNR |
| Photugard | 178.67 | 0.7146 | 0.3830 | 144.72 | 0.8366 | 0.3790 | 101.84 | 0.5662 | 0.3736 | 117.19 | 0.2880 | 0.3969 | 29.6323 |
| AdvDM | **183.50** | **0.6800** | **0.4400** | **150.11** | 0.8126 | **0.4318** | 106.74 | 0.5394 | 0.3782 | 120.36 | **0.2614** | **0.4134** | 29.4626 |
| SDST | 179.31 | 0.7214 | 0.3956 | 145.99 | 0.8284 | 0.3914 | 104.13 | 0.5564 | 0.3783 | 118.56 | 0.2710 | 0.4003 | 29.5710 |
| ADVPAINT | 183.44 | 0.6894 | 0.4126 | 149.74 | **0.8110** | 0.4080 | **108.70** | **0.5150** | **0.3837** | **124.74** | 0.2712 | 0.4084 | 29.6232 |

Table 11: Quantitative evaluation of inpainting results after applying JPEG compression.

| | PSNR |
|---|---|
| Photoguard | 31.6608 |
| AdvDM | 32.5213 |
| SDST | 32.4273 |
| ADVPAINT | 32.3779 |

Table 12: PSNR comparison of ADVPAINT and baseline methods where they are equally set with $\eta = 0.06$.

uated the two methods for which official code is available in the current time of writing this paper—Gaussian noise addition and upscaling—while the others could not be tested due to the lack of accessible implementations. For the purification methods, we follow the Pytorch implementation for JPEG compression with quality 15 and official codes for other methods where Gaussian noise strength is set to 0.05.

In Table 8, ADVPAINT retains its protective ability even against IMPRESS, outperforming baseline methods in terms of FID, Precision, and LPIPS. Since IMPRESS uses LPIPS loss to ensure the purified image remains visually close to the perturbed image, we believe this objective inadvertently preserves a part of the adversarial perturbation. Qualitative results are depicted in Figure 16.

We observed that both ADVPAINT and the previous methods lose their ability to protect images when subjected to Gaussian noise addition, upscaling (Honig et al., 2024), and JPEG compression. In Table 9, 10, and 11, the FID, Precision, and LPIPS scores indicate significant degradation in protection under these conditions.

However, as depicted in the Figure 17 (a) and (c) regarding Gaussian noise addition and JPEG compression, *the inpainted results are noisy and blurry* (e.g. noisy backgrounds for (a) "sunflower" and (c) "bicycle" images) compared to images generated from non-protected input. This raises concerns about their visual quality. It calls into question the practicality of noise-erasing methods, as the generated images often fail to meet acceptable quality standards.

| Noise Level $\eta$ | Foreground Inpainting | | | | | | Background Inpainting | | | | | | |
|---|---|---|---|---|---|---|---|---|---|---|---|---|---|
| | $m^{seg}$ | | | $m^{bb}$ | | | $m^{seg}$ | | | $m^{bb}$ | | | |
| | FID ↑ | Prec ↓ | LPIPS ↑ | FID ↑ | Prec ↓ | LPIPS ↑ | FID ↑ | Prec ↓ | LPIPS ↑ | FID ↑ | Prec ↓ | LPIPS ↑ | PSNR |
| 0.04 | 319.54 | 0.1298 | 0.6056 | 268.58 | 0.2578 | 0.6138 | 170.37 | 0.3040 | 0.4603 | 247.31 | 0.1090 | 0.5602 | 35.2832 |
| AdvPaint (0.06) | **347.88** | **0.0570** | **0.6731** | **289.63** | **0.1536** | **0.6762** | **219.07** | **0.2148** | **0.5064** | **303.90** | **0.0936** | **0.6105** | **32.3779** |
| 0.08 | 368.37 | 0.0320 | 0.7446 | 311.58 | 0.0992 | 0.7447 | 250.44 | 0.1630 | 0.5506 | 330.50 | 0.0782 | 0.6575 | 29.9798 |
| 0.1 | 376.69 | 0.0226 | 0.7846 | 326.70 | 0.0642 | 0.7829 | 266.12 | 0.1432 | 0.5780 | 340.51 | 0.0818 | 0.6831 | 28.3171 |

Table 13: Quantitative evaluation of inpainting results for $\eta = 0.04, 0.06, 0.08, 0.1$. Results of AdvPaint are in bolded letters.

| Iter. Steps | Foreground Inpainting | | | | | | Background Inpainting | | | | | |
|---|---|---|---|---|---|---|---|---|---|---|---|---|
| | $m^{seg}$ | | | $m^{bb}$ | | | $m^{seg}$ | | | $m^{bb}$ | | |
| | FID ↑ | Prec ↓ | LPIPS ↑ | FID ↑ | Prec ↓ | LPIPS ↑ | FID ↑ | Prec ↓ | LPIPS ↑ | FID ↑ | Prec ↓ | LPIPS ↑ |
| 50 | 336.39 | 0.0826 | 0.6575 | 284.00 | 0.1872 | 0.6650 | 197.29 | 0.2736 | 0.4894 | 274.99 | 0.1122 | 0.5942 |
| 100 | 343.72 | 0.0728 | 0.6720 | 287.60 | 0.1744 | 0.6781 | 207.59 | 0.2308 | 0.5087 | 296.83 | 0.0894 | 0.6082 |
| 150 | 339.79 | 0.0794 | 0.6598 | 285.29 | 0.1898 | 0.6654 | 204.49 | 0.2672 | 0.4958 | 293.95 | 0.1178 | 0.5974 |
| AdvPaint (250) | **347.88** | **0.0570** | **0.6731** | **289.63** | **0.1536** | **0.6762** | **219.07** | **0.2148** | **0.5064** | **303.90** | **0.0936** | **0.6105** |

Table 14: Quantitative evaluation of inpainting results for iteration steps = $50, 100, 150, 250$. Results of AdvPaint are in bolded letters.

Additionally, we observed a critical drawback in existing purification methods: *they tend to degrade the quality of the purified image itself.* As shown in Table 12, AdvPaint and baseline methods in our experiments leveraged PGD with $\eta = 0.06$, ensuring adversarial examples retained a PSNR around 32 dB. On the other hand, after purification (e.g., via upscaling), we observed a PSNR drop of approximately 2.5 dB for AdvPaint, with similar reductions observed for other methods. This decline highlights a significant trade-off between the purification effectiveness and input image quality. Qualitative results after these purification methods are depicted in Figure 17.

## A.6 Ablation Study of Noise Levels and Iteration Steps

### A.6.1 Analysis of Noise Levels

In Table 13, we conducted an experiment with different values of $\eta$, ranging from 0.04 to 0.1. While the PSNR values of adversarial examples increase as $\eta$ increases, we observed consistent improvements across all evaluation metrics, including FID, precision, and LPIPS. For AdvPaint, we set $\eta$ to 0.06, as it effectively balances protection against inpainting tasks with the quality of the protected image, achieving a PSNR of approximately 32 dB.

### A.6.2 Analysis of Iteration Steps

In Table 14, we experimented with varying iteration steps. We evaluated iteration steps ranging from 50 to 150. Due to memory limitations, we set the default iteration steps to 250 in AdvPaint, as higher iterations result in memory overload. The results show that while there may not be significant improvement for iteration steps around 100 and 150, optimizing for 250 steps consistently outperforms lower iteration counts, validating our choice of 250 steps as the default setting for AdvPaint.

| Optim. Methods | Foreground Inpainting | | | | | | Background Inpainting | | | | | |
|---|---|---|---|---|---|---|---|---|---|---|---|---|
| | $m^{seg}$ | | | $m^{bb}$ | | | $m^{seg}$ | | | $m^{bb}$ | | |
| | FID ↑ | Prec ↓ | LPIPS ↑ | FID ↑ | Prec ↓ | LPIPS ↑ | FID ↑ | Prec ↓ | LPIPS ↑ | FID ↑ | Prec ↓ | LPIPS ↑ |
| LDM + $\mathcal{L}_{attn}$ | 258.63 | 0.3276 | 0.6221 | 209.34 | 0.5202 | 0.6057 | 103.89 | 0.5610 | 0.3872 | 135.07 | 0.2274 | 0.5010 |
| AdvPaint | **347.88** | **0.0570** | **0.6731** | **289.63** | **0.1536** | **0.6762** | **219.07** | **0.2148** | **0.5064** | **303.90** | **0.0936** | **0.6105** |

Table 15: Quantitative comparison with optimization applied to the default LDM using the same objective as AdvPaint. LDM refers to the model used in our baseline models (*e.g.* Photoguard, AdvDM, CAAT, etc.).

## A.7 Additional Quantitative Results

In Table 15, we conduct a simple experiment to evaluate the impact of replacing the objective functions in our baseline models. Specifically, we use the same Latent Diffusion Model (LDM) as the baselines but substitute their objective functions—replacing Eq. 2 (e.g., AdvDM, CAAT) and Eq. 4 (e.g., Photoguard) with our proposed attention loss (Eq. 8). Since attention blocks are also present in this *default LDM*, our attention loss is directly applicable. After optimizing perturbations targeting the LDM, we generate inpainted results using the Stable Diffusion inpainting model.

The results indicate that, while optimized with our proposed objective, the perturbations fail to provide effective protection against inpainting tasks, under-performing compared to ADVPAINT. Furthermore, compared to rows 2–6 in Table 1 (i.e., baseline models), replacing the objective function with our attention loss does not result in a significant improvement in performance.

We attribute this to the lack of direct targeting of inpainting models, which limits their ability to counter inpainting-specific attacks. This highlights a key limitation of current protection methods that rely on the *default LDM* for inpainting tasks and underscores the critical importance of designing defensive methods specifically tailored for such tasks.

## A.8 Additional Qualitative Results

### A.8.1 Single- and Multi-object Images

We present additional qualitative results for inpainting tasks of *single-object* images, comparing our method with prior protection approaches in Figure 18, demonstrating the effectiveness of ADVPAINT in protecting against both foreground and background inpainting with diverse masks in Figure 19. These results confirm the robustness of our method across various masks and prompts.

For the optimization process of *multi-object* images, we first position ourselves as content owners and select the objects that may be at risk of malicious inpainting modifications. Then, ADVPAINT performs PGD optimization for each object using enlarged bounding box masks generated by Grounded SAM. After optimizing each object, the leftover background regions, where objects potentially at risk do not exist, are also optimized. In Figure 20, we clarify the masks used to optimize multi-object images, aiding comprehension. After securing each object, we conducted experiments with a variety of mask types, including single-object masks, masks for other objects, combined-object masks, and their inverted versions. Figure 21 demonstrates the robustness of ADVPAINT for multi-object images. For example, since ADVPAINT optimizes each object individually, it ensures protection for each object, resulting in inpainted images that lack discernible objects in the foreground. Furthermore, ADVPAINT is robust to masks that encompass all objects, as shown by the absence of "two cameras" replacing "two dogs" in inpainted images. Additionally, the method effectively secures background regions when inverted masks are used for inpainting tasks. These results substantiate the effectiveness of ADVPAINT 's per-object protection method, even for complex multi-object scenarios.

### A.8.2 Alternative resources for prompt generation and mask creation

We conducted additional experiments employing alternative resources for prompt generation and mask creation to evaluate the robustness and generalizability of ADVPAINT 's protection performance. For prompt generation, in addition to ChatGPT, we utilized Claude 3.5 Sonnet to generate diverse prompts. For mask generation, we replaced Grounded SAM with the zero-shot segmentation method proposed by Yu et al. (2023a), which employs CLIP (Radford et al., 2021) to create object masks based on the given prompt. As depicted in Figure 22, ADVPAINT retains its protection performance for inpainting tasks, comparable to its performance when using ChatGPT and Grounded SAM. However, as shown in Figure 23, we observe that the segmentation results from Yu et al. (2023a) are generally less accurate compared to those generated by Grounded SAM. This reinforces our choice of Grounded SAM as the primary segmentation tool, while also validating ADVPAINT 's adaptability to alternative segmentation approaches.

### A.8.3 MASKS EXCEEDING OR OVERLAPPING THE OPTIMIZATION BOUNDARY

Since ADVPAINT leverages enlarged bounding box of the object in an image to optimize effective perturbations, one may be curious about if ADVPAINT is also robust to real-world inpainting scenarios where masks vary in sizes and shapes. In addition to the experiment conducted in Section 5.6, we conducted additional experiments using masks that exceed or overlap with the optimization boundary. Specifically, we visualized inpainting results where foreground masks were applied to regions without objects, simulating adversarial scenarios aimed at generating new objects in the background. As depicted in Figure 24, ADVPAINT remains robust in such diverse inpainting cases that reflect the potential threat from adversaries.

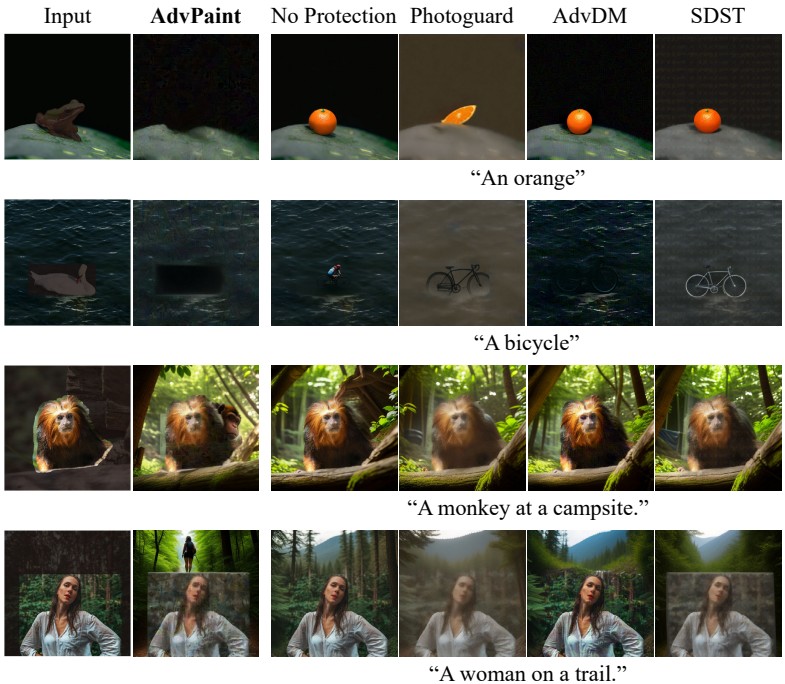

Figure 7: Qualitative inpainting results of HD-Painter (Manukyan et al., 2023).

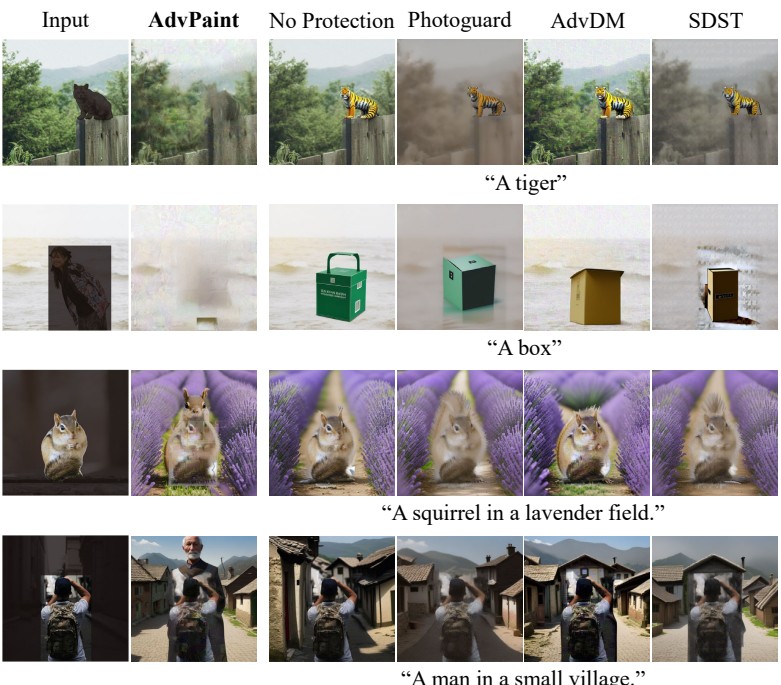

Figure 8: Qualitative inpainting results of DreamShaper (DreamShaper, 2024).

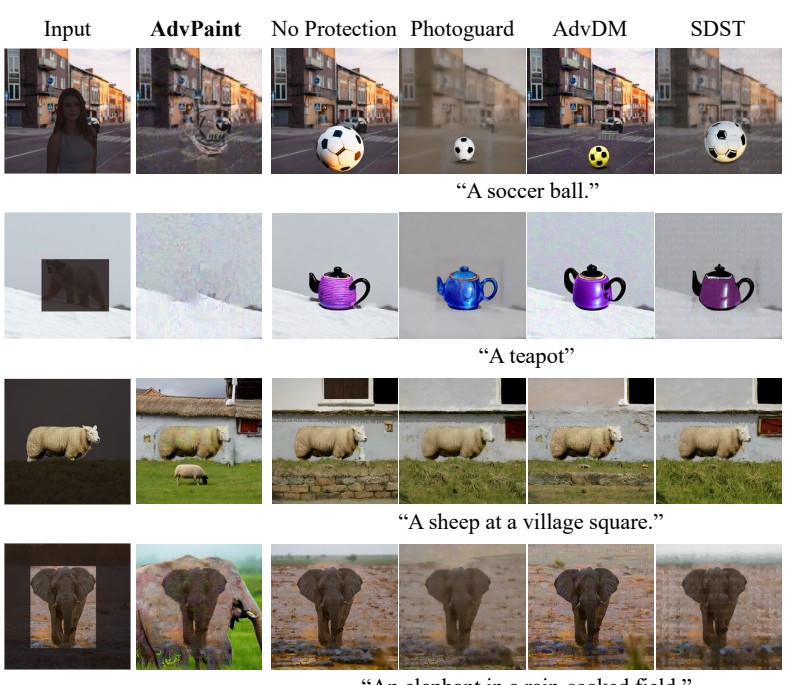

Figure 9: Qualitative inpainting results of Stable-Diffusion-2-Inpainting model.

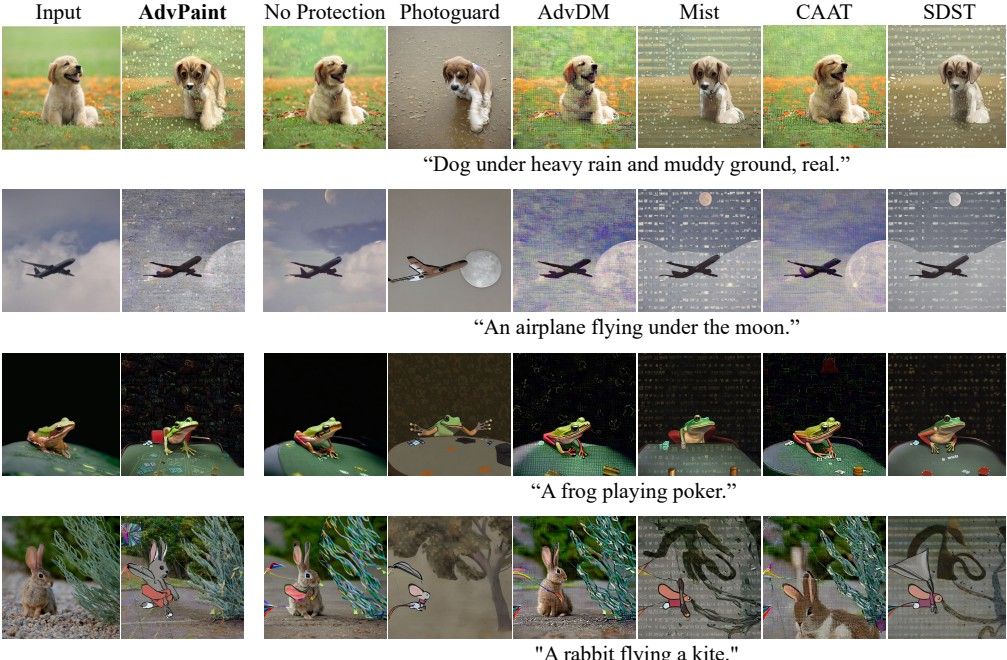

Figure 10: Comparison in image-to-image translation task. The results are generated via Stable Diffusion image-to-image pipeline.

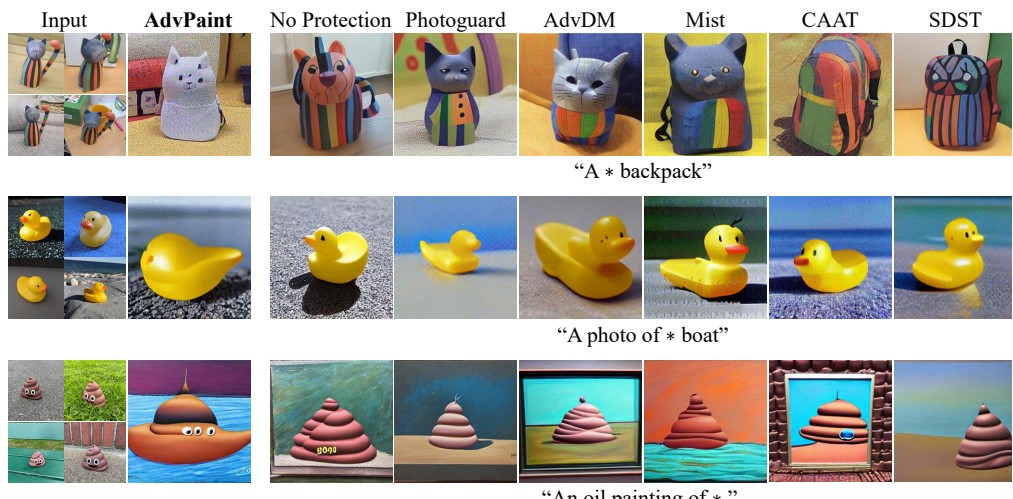

Figure 11: Comparison of text-to-image generation. The ∗ in the prompts indicates the representative prompt corresponding to the input images. The results are generated via Textual Inversion (Gal et al., 2022).

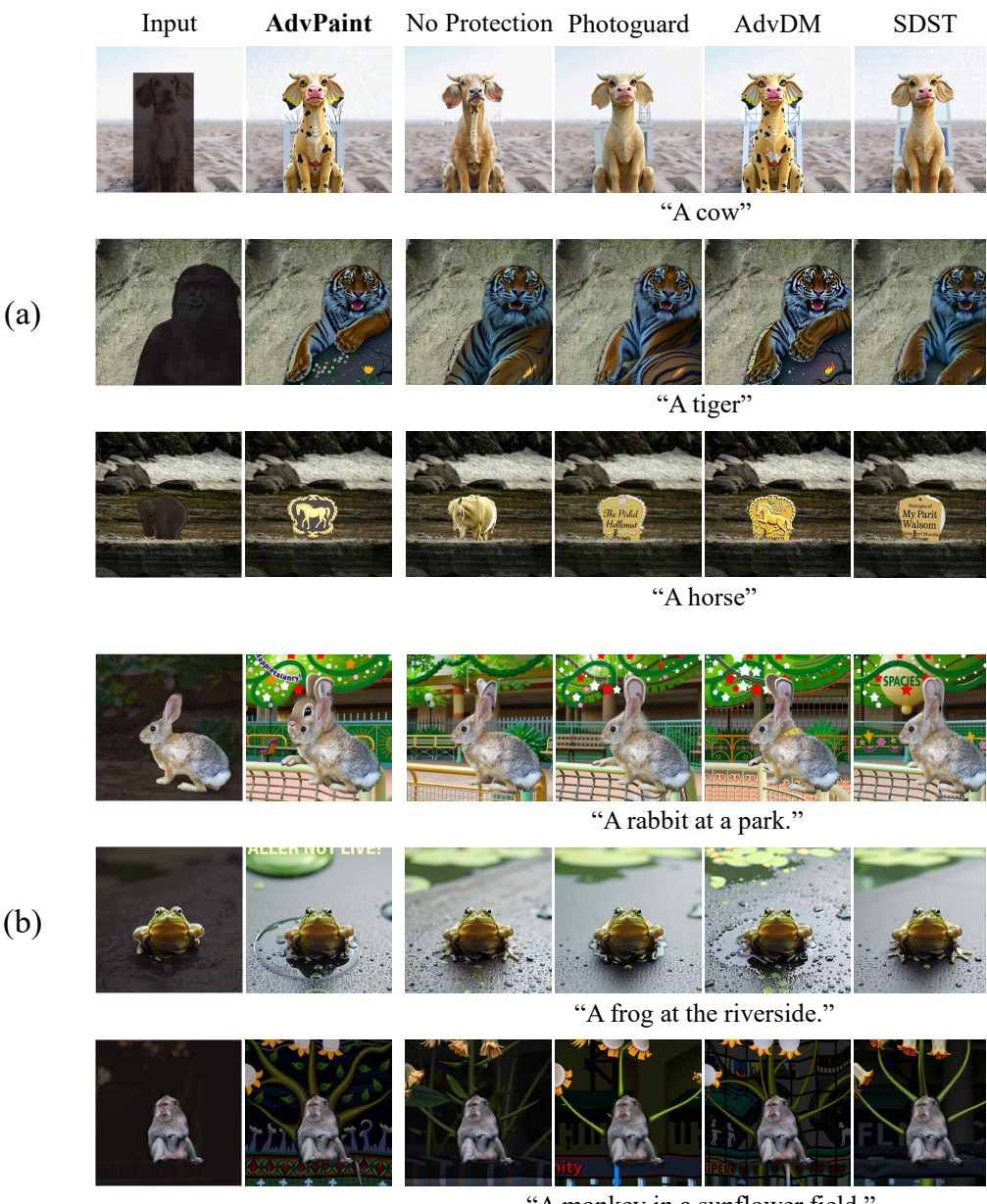

Figure 12: Qualitative results of ADVPAINT and baseline models applied to Flux (Labs, n.d.). Results demonstrate the transferability of ADVPAINT to DiT-based inpainting models, causing misalignment between generated regions and unmasked areas in both (a) foreground and (b) background inpainting tasks. Dark parts in the input image indicate the masked regions.

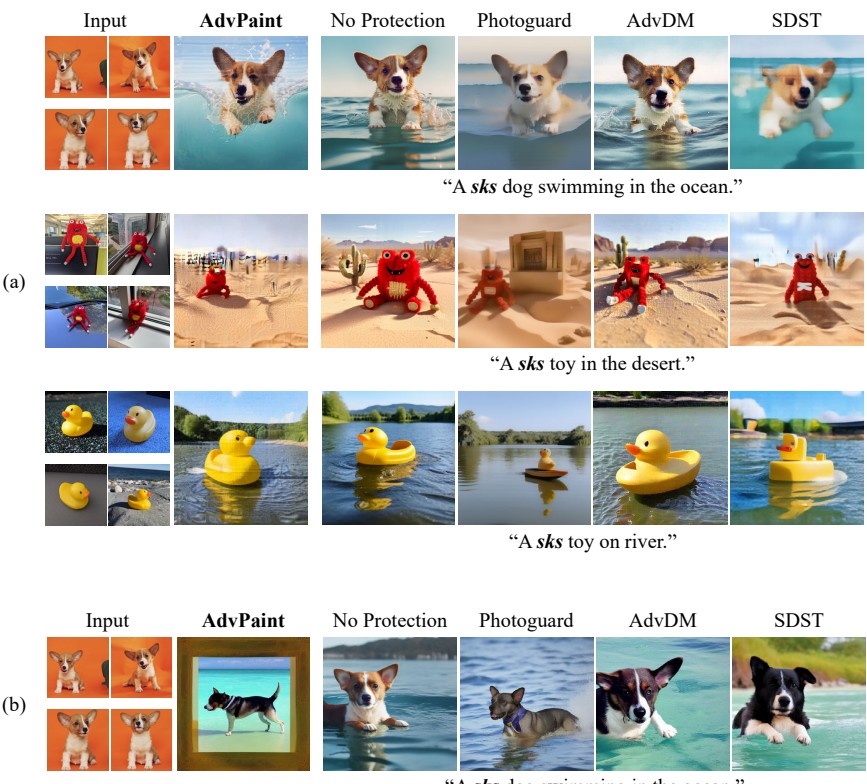

Figure 13: Comparison of text-to-image generation from (a) DiT-based DreamBooth (Chen et al., 2024) and (b) the original DreamBooth (Ruiz et al., 2022).

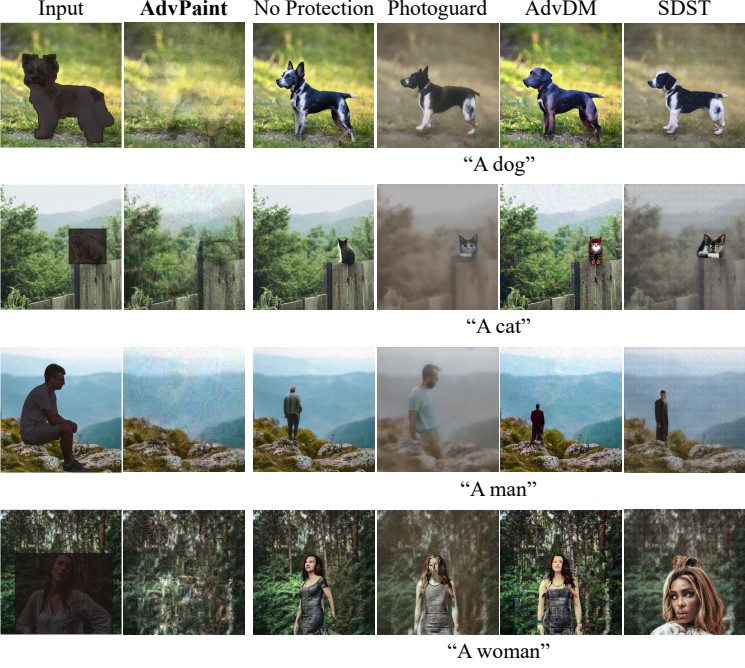

Figure 14: Qualitative results of foreground inpainting with prompts that describe the mask-covered object. Dark parts in the input image indicate the masked regions.

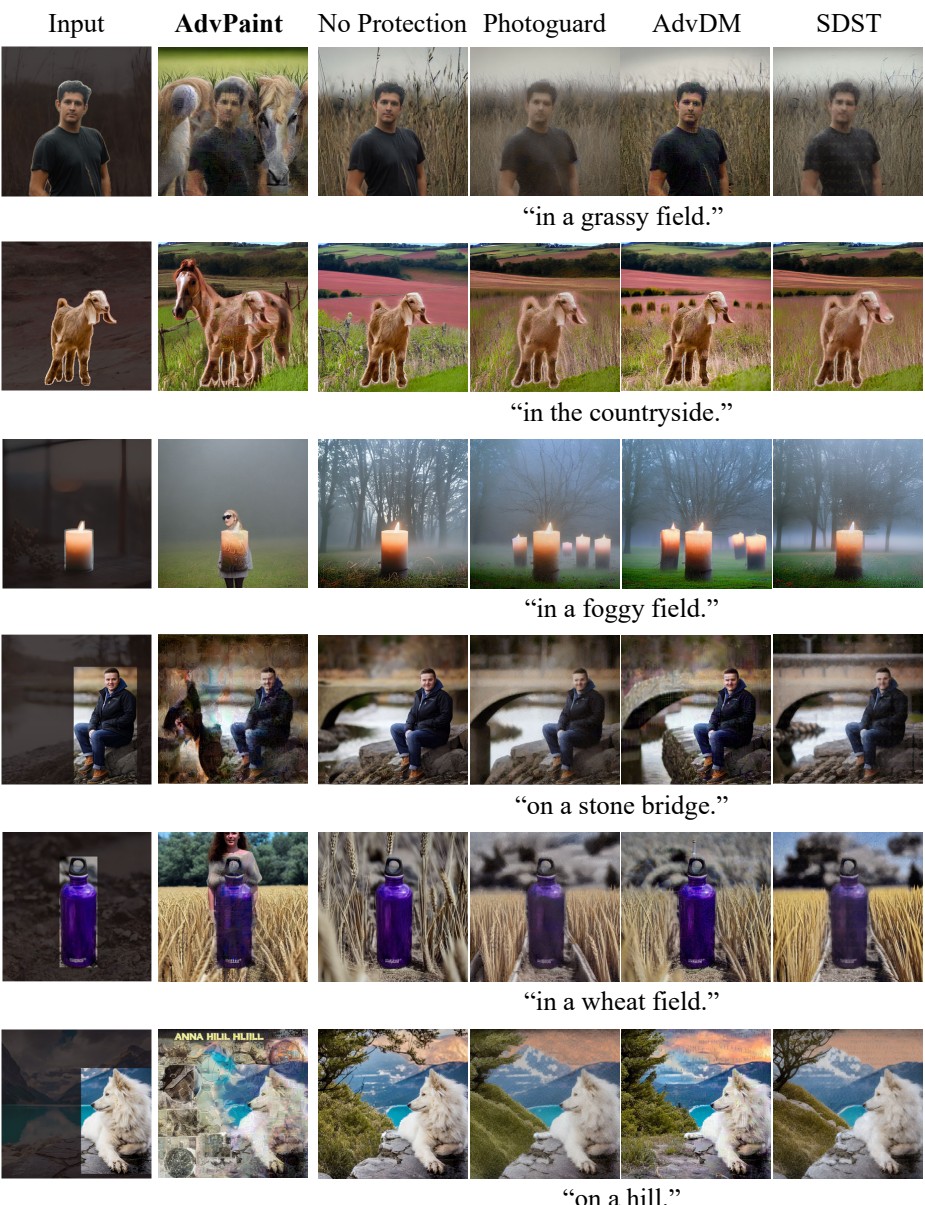

Figure 15: Qualitative results of background inpainting with prompts that follow the format of {preposition, location}. Dark parts in the input image indicate the masked regions.

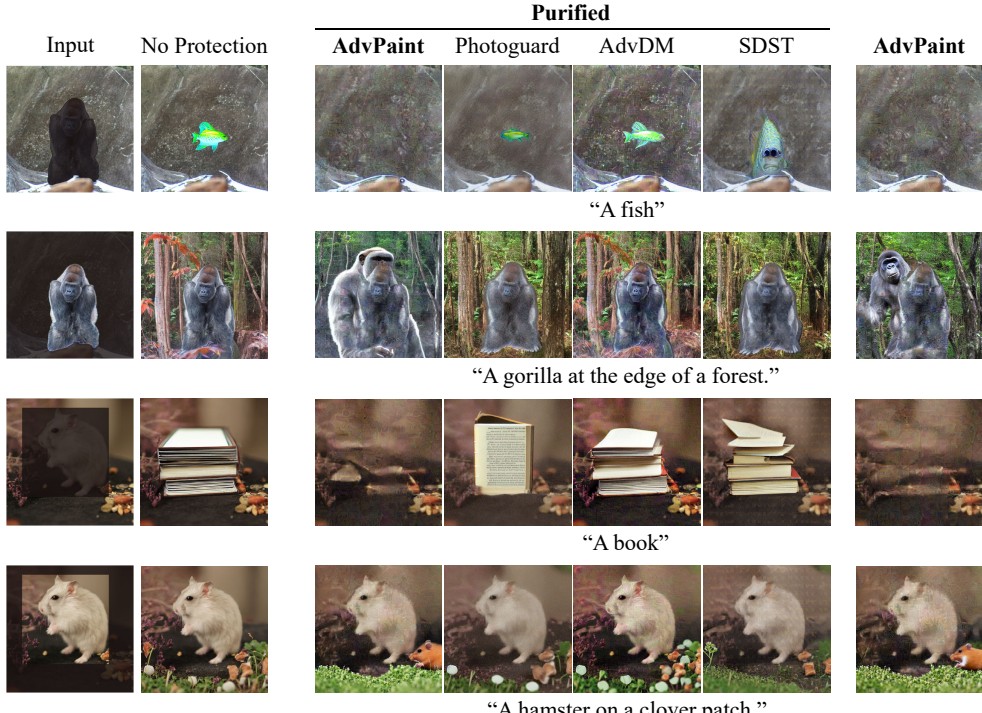

Figure 16: Qualitative inpainting results after applying IMPRESS (Cao et al., 2023). Dark parts in the input image indicate the masked regions.

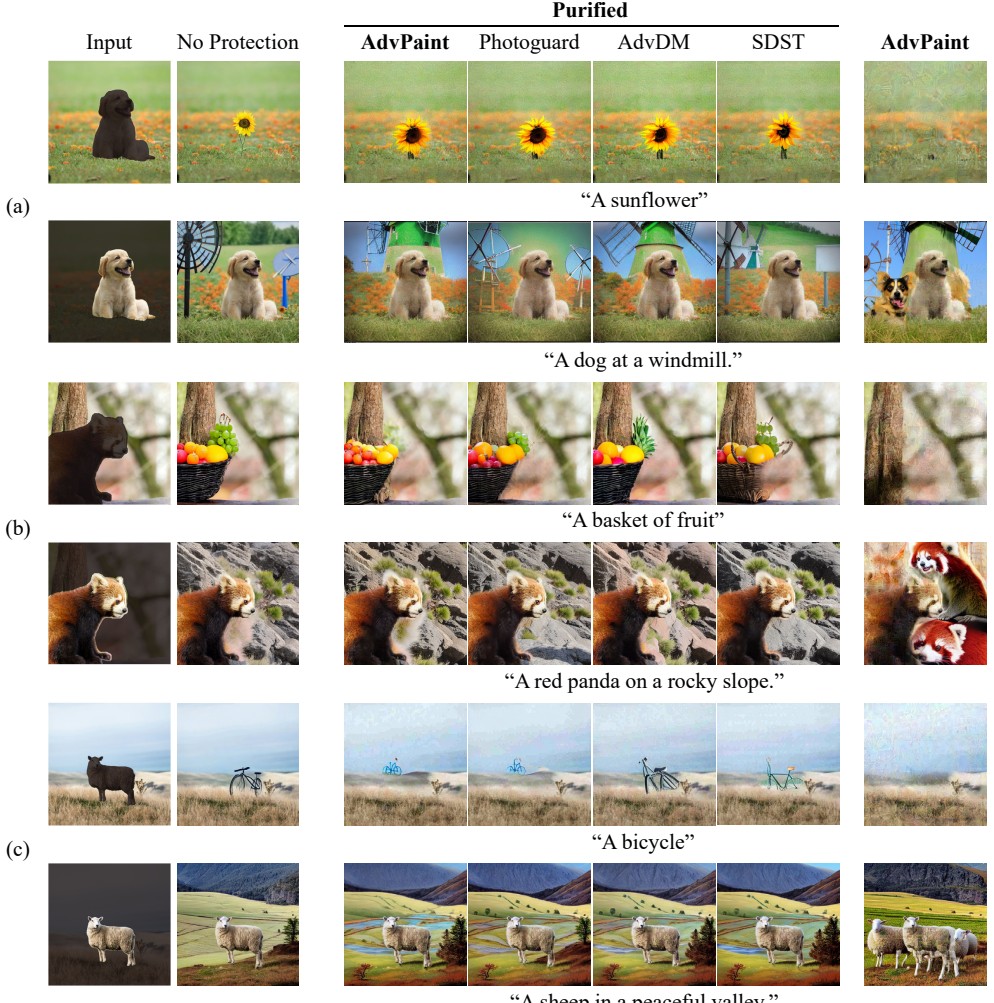

Figure 17: Qualitative inpainting results after applying (a) Gaussian Noise, (b) Upscaling (Honig et al., 2024), and (c) JPEG compression. Dark parts in the input image indicate the masked regions.

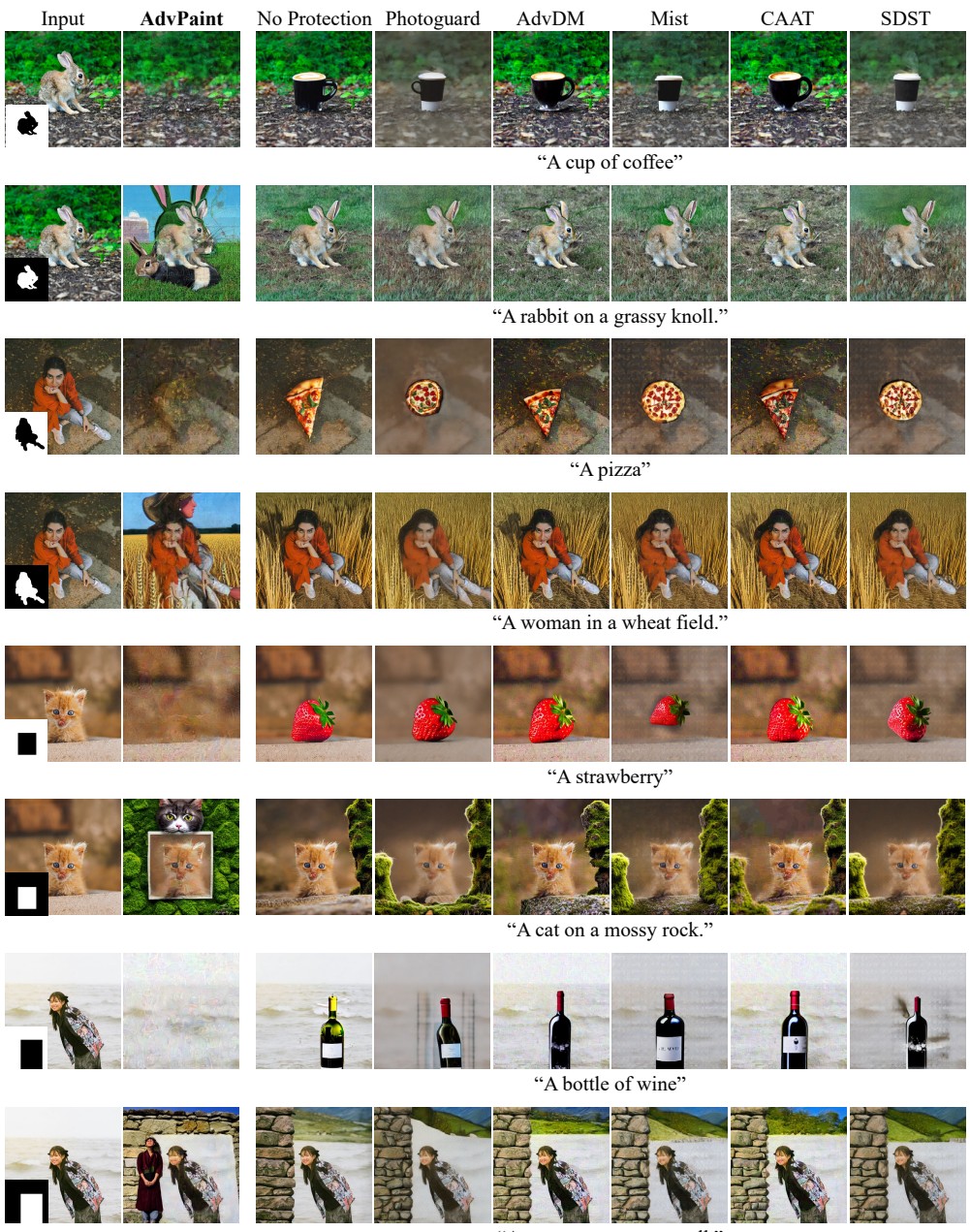

Figure 18: Qualitative results of inpainting tasks using segmentation mask $m^{seg}$ and bounding box mask $m^{bb}$, comparing with prior methods.

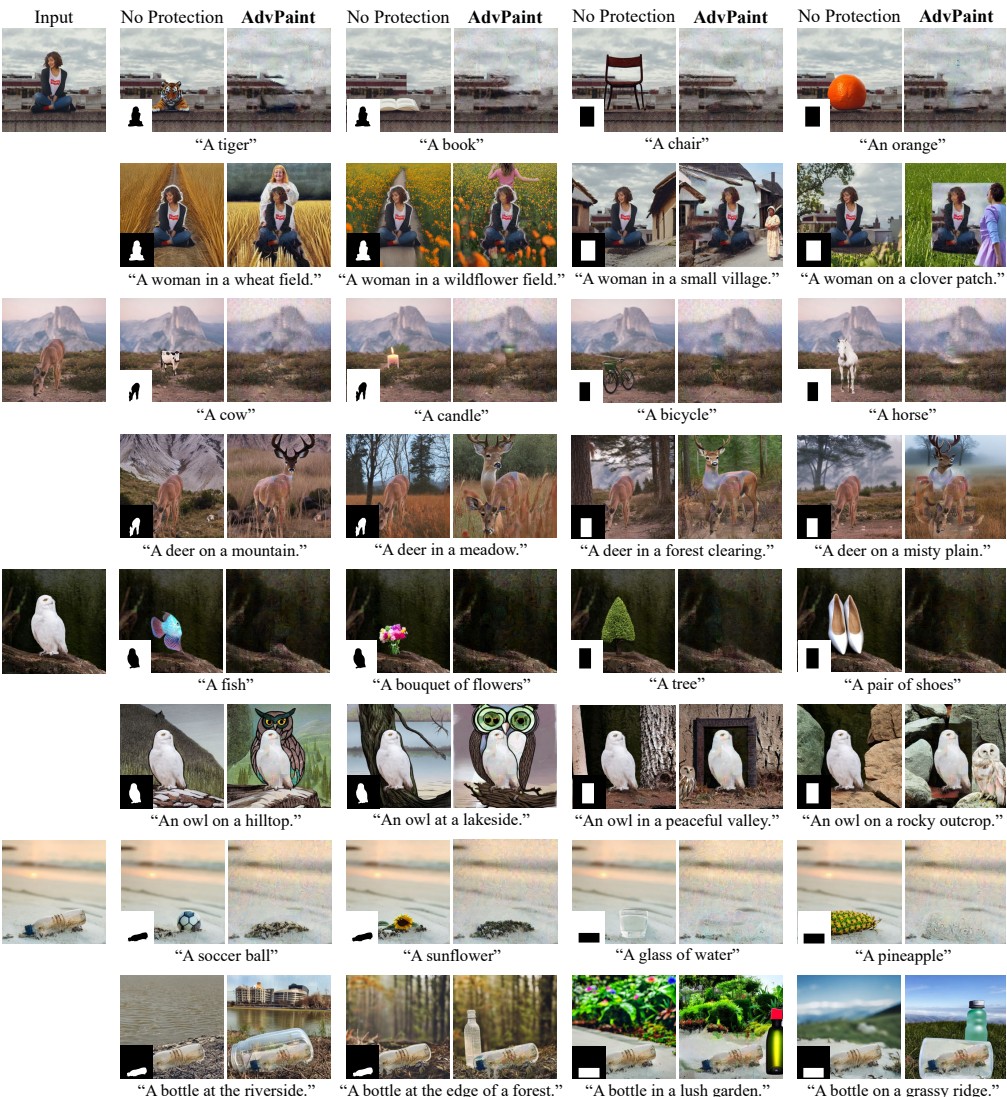

Figure 19: Qualitative results of our approach on inpainting tasks with masks $m^{seg}$, $m^{bb}$, and the optimization mask $m$.

Input          Applied Masks

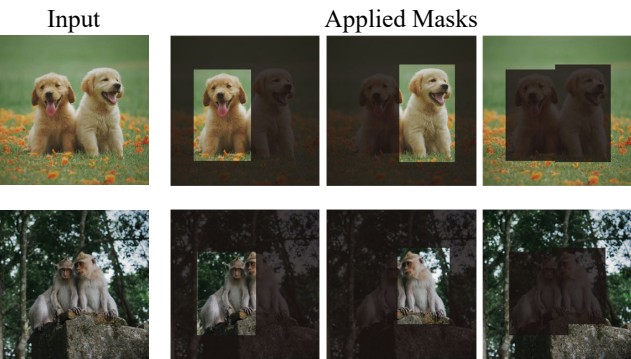

Figure 20: Masks used in optimization process of multi-object images. We utilize enlarged bounding box generated by Grounded SAM. Dark parts in the input image indicate the masked regions.

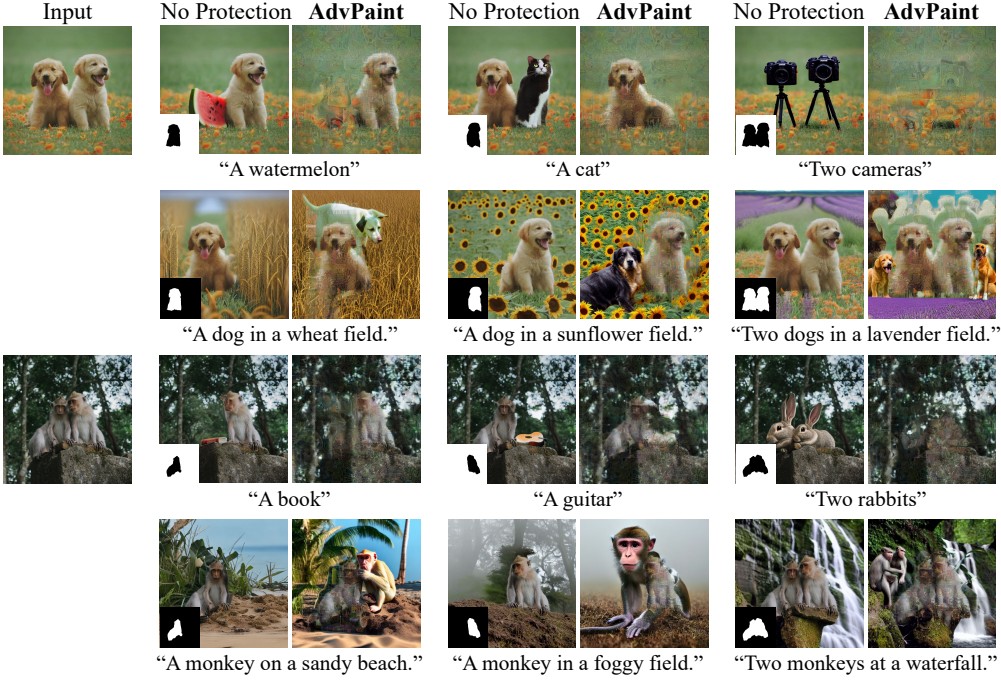

Figure 21: Qualitative results of inpainting tasks for multi-object images.

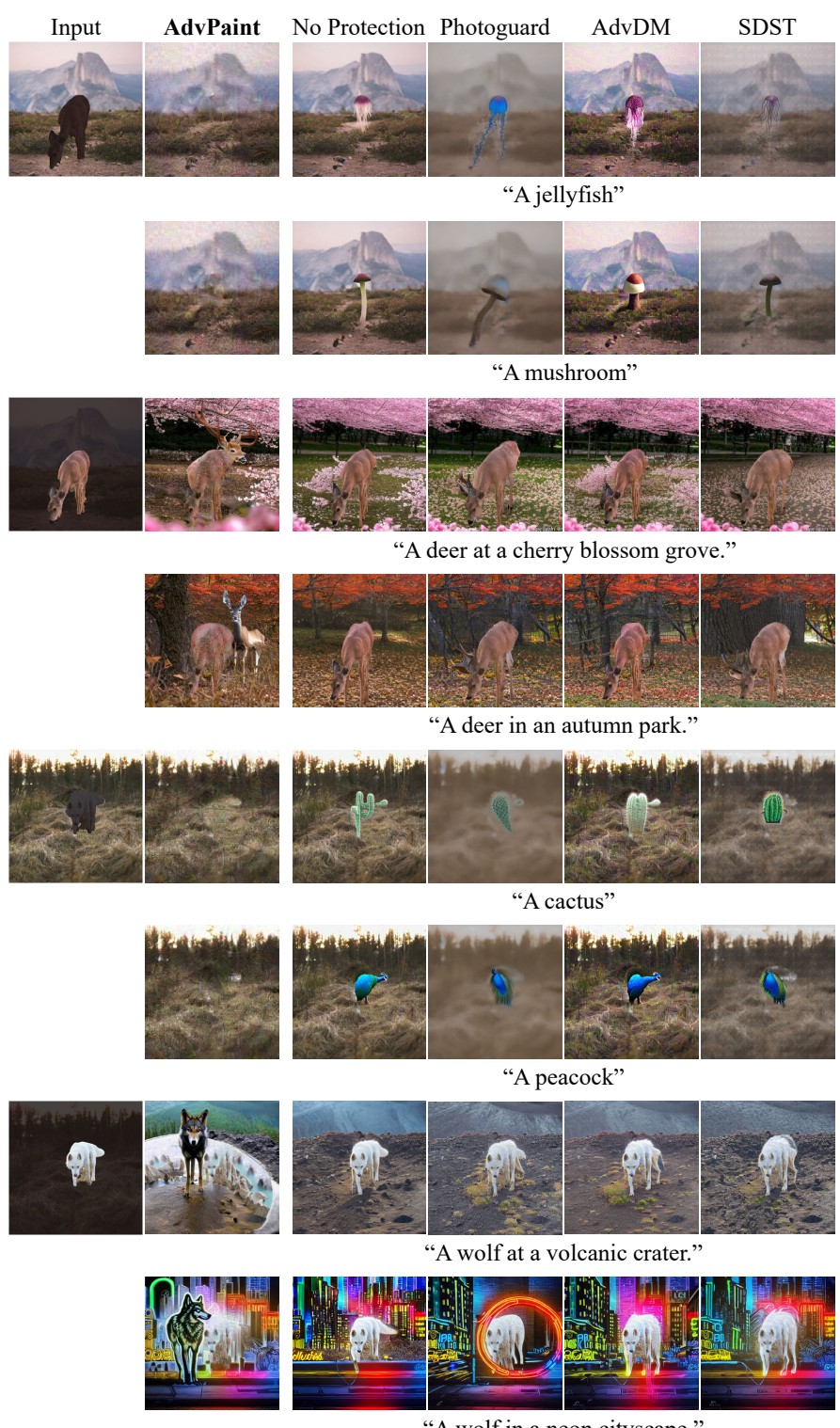

Figure 22: Qualitative results of inpainting tasks with prompts and masks generated from alternative resources. Dark parts in the input image indicate the masked regions.

(a) Yu *et al.*    (b) Grounded SAM

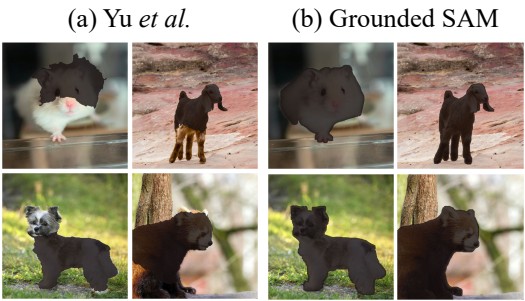

Figure 23: Qualitative results of inpainting tasks with prompts and masks generated from alternative resources. Dark parts in the input image indicate the masked regions.

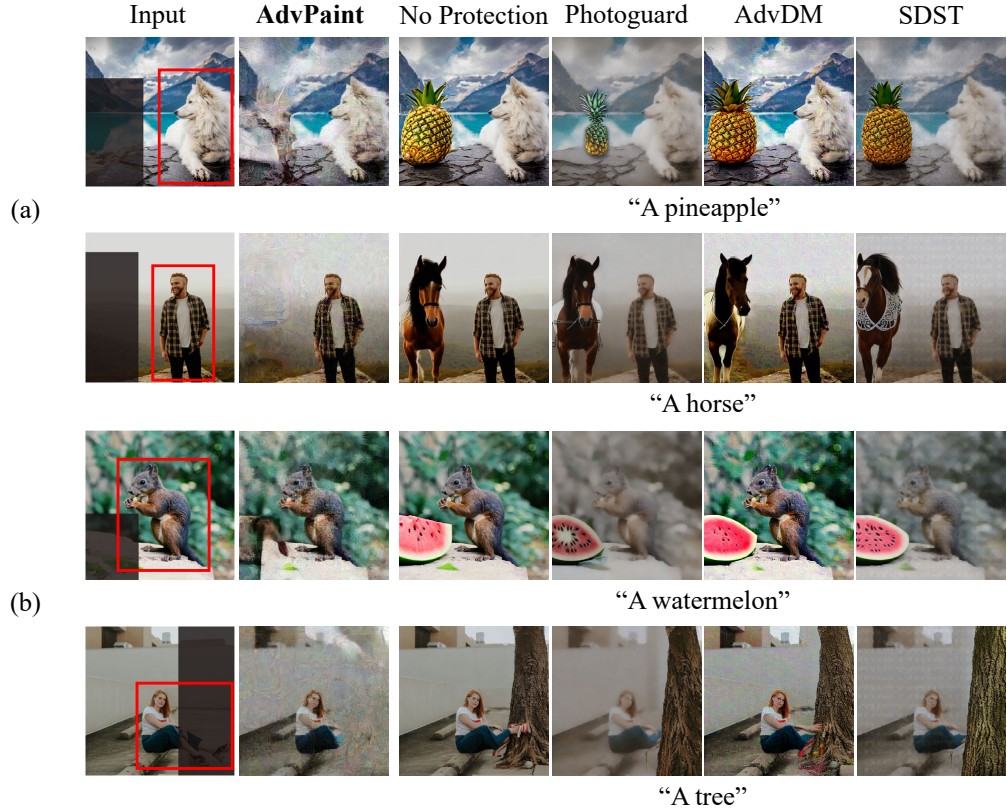

Figure 24: Qualitative inpainting results where (a) masks exceed or (b) overlap with the optimization boundary (highlighted in red lines). Dark parts in the input image indicate the masked regions.

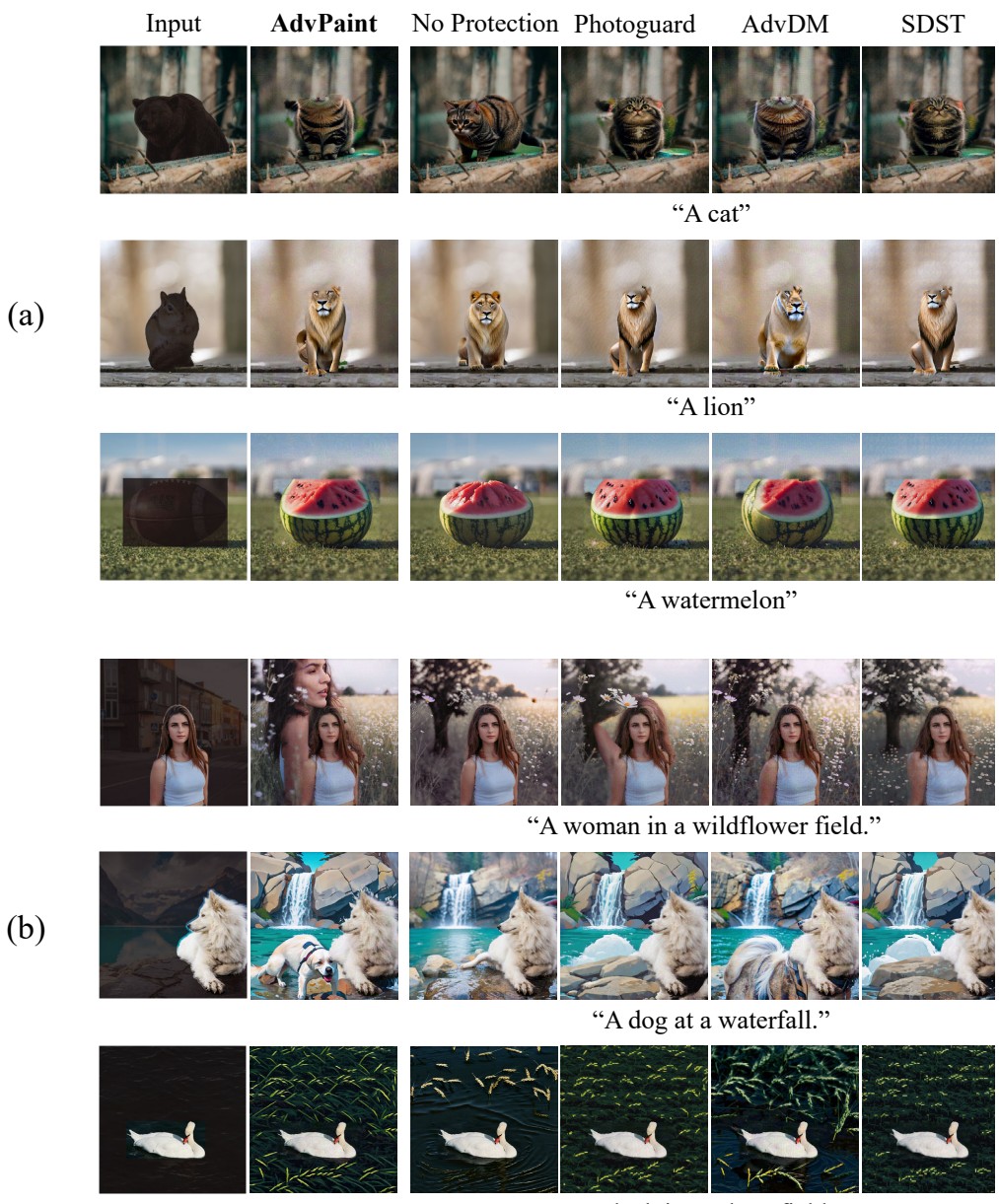

Figure 25: Qualitative results of ADVPAINT and baseline models applied to SD3 (Esser et al., 2024). Results demonstrate the transferability of ADVPAINT to DiT-based inpainting models, causing mis-alignment between generated regions and unmasked areas in both (a) foreground and (b) background inpainting tasks. Dark parts in the input image indicate the masked regions.