# OpenReview forum: "AdvPaint: Protecting Images from Inpainting Manipulation via Adversarial Attention Disruption"
_ICLR.cc/2025/Conference — ICLR 2025 Poster_

### Official Review · Reviewer_37iq · 2024-10-29

**Soundness:** 4
**Presentation:** 4
**Contribution:** 3
**Rating:** 8
**Confidence:** 5

**Summary:**

This paper proposes a defense framework named ADVPAINT, designed to protect images from inpainting attacks based on diffusion models, specifically to prevent malicious actors from making unauthorized changes to specific regions of an image using such models. The method generates perturbations to disrupt the attention mechanisms of diffusion models (including self-attention and cross-attention blocks), thereby disturbing semantic understanding and the generation process. Additionally, ADVPAINT employs a two-stage perturbation strategy, applying different perturbations to the target region and the background, which enhances robustness under various masking conditions.

**Strengths:**

ADVPAINT employs a two-stage perturbation generation approach, dividing the image into target and background regions and applying different perturbations to each. This strategy enhances the method’s robustness across various mask shapes and sizes, enabling ADVPAINT to maintain a high level of protection against diverse attack methods.

**Weaknesses:**

1. ADVPAINT's approach of dividing the target region and background is fixed, resulting in limited flexibility when facing custom masks created by adversaries. In real-world scenarios, an attacker could select masking regions that do not overlap or only partially overlap with the predefined target mask, potentially weakening ADVPAINT's protective effect. I recommend that the authors conduct experiments to demonstrate ADVPAINT’s robustness under various custom mask configurations.

2. The paper does not explore the effect of varying noise levels and different PGD iteration steps on the robustness of the ADVPAINT model. The experiments use a fixed noise budget and iteration count, leaving it unclear how the model's performance might change under different adversarial intensities. I suggest the authors conduct experiments to analyze the robustness trend across various noise magnitudes and iteration counts to provide a more comprehensive evaluation of ADVPAINT's protective capabilities.

3. The paper lacks experiments on scalable diffusion models with Transformers. The evaluation focuses on standard Stable Diffusion models, leaving uncertainty about ADVPAINT’s effectiveness and adaptability on larger, Transformer-based diffusion architectures. I recommend that the authors test ADVPAINT on scalable Transformer-based diffusion models to better assess its robustness in more extensive generative frameworks.

Peebles, William, and Saining Xie. "Scalable diffusion models with transformers." Proceedings of the IEEE/CVF International Conference on Computer Vision. 2023.

**Questions:**

See weakness

---

> ### Author Response · Authors · 2024-11-21
>
> Dear Reviewer 37iq,
>
> Thank you for your detailed comments and helpful suggestions regarding our manuscript. We have carefully considered each of your points and provided comprehensive responses below. If there are any aspects we may have inadvertently overlooked, please let us know, and we will gladly address them.
>
> # W1) Robustness of AdvPaint Against Custom Mask Configurations
>
> We thank the reviewer for highlighting this important aspect of real-world scenarios.
>
> In the real-world scenario, the purpose of inpainting may differ. To address these concerns, we conducted additional experiments in Fig.24 of Appendix A.8.3 using masks that (a) exceed or (b) overlap with the optimization boundary. Specifically, we visualized inpainting results where foreground masks were applied to regions without objects, simulating adversarial scenarios aimed at generating new objects in the background. These experiments demonstrate that **_AdvPaint_ remains robust in such diverse inpainting cases, generating inpainted images with no discernable objects.**
>
> Additionally, in Section 5.6, we present the inpainting results for two types of masks: $m^{out}$ and $m^{in}$. Since masks can be hand-crafted by adversaries and may exceed the enlarged bounding box used at the optimization step of AdvPaint, we simulate this scenario by randomly shifting the segmentation mask $m^{seg}$ in four directions (up, down, left, and right) to ensure that at least one boundary of the rectangular optimization box is exceeded. This shifted mask is referred to as $m^{out}$. In contrast, $m^{in}$ is generated by randomly shifting $m^{seg}$ while ensuring the entire mask remains inside the rectangular optimization box, reflecting scenarios where segmentation masks differ but are confined to the bounded region.
>
> Our experiments, as shown in Table 3, demonstrate that **_AdvPaint_ maintains robust performance for both $m^{out}$ and $m^{in}$.** Specifically, it achieves strong performance across key metrics, including FID, precision, and LPIPS scores, effectively safeguarding images under both the scenarios.
>
> We hope these additional scenarios in evaluating the robustness of AdvPaint  addresses the reviewer’s concerns.

---

> ### Author Response · Authors · 2024-11-21
>
> # W2) Analysis of Noise Levels and Iteration Steps
> We thank the reviewer for raising this important concern. To address it, we conducted two experiments, varying the noise budget $\eta$ and the number of iteration steps, and summarized the results in the tables below. The updated tables have also been included in Appendix A.6.
>
> ## 1. Noise Levels $\eta$
> First, we conducted an experiment with different values of $\eta$, ranging from 0.04 to 0.1. While the PSNR values of adversarial examples decrease as $\eta$ increases, we observed consistent improvements across all evaluation metrics, including FID, precision, and LPIPS. For AdvPaint, we set η to 0.06, as it effectively balances protection against inpainting tasks with the quality of the protected image, achieving a PSNR of approximately 32 dB.
>
> ## 2. Iteration Steps
> Second, we experimented with varying iteration steps. We evaluated iteration steps ranging from 50 to 150. Due to memory limitations, we set the default iteration steps to 250 in AdvPaint, as higher iterations result in memory overload. The results show that while there may not be significant improvement for iteration steps around 100 and 150, optimizing for 250 steps consistently outperforms lower iteration counts, validating our choice of 250 steps as the default setting for AdvPaint.
>
> We hope these additional analyses comprehensively address the reviewer’s concern and provide deeper insights into the robustness of AdvPaint under varying adversarial intensities.
>
> | $\eta$    | FG, $m^{seg}$, FID ↑ | FG, $m^{seg}$, Prec. ↓ | FG, $m^{seg}$, LPIPS ↑ | FG, $m^{bb}$, FID ↑ | FG, $m^{bb}$, Prec. ↓ | FG, $m^{bb}$, LPIPS ↑ | BG, $m^{seg}$, FID ↑ | BG, $m^{seg}$, Prec. ↓ | BG, $m^{seg}$, LPIPS ↑ | BG, $m^{bb}$, FID ↑ | BG, $m^{bb}$, Prec. ↓ | BG, $m^{bb}$, LPIPS ↑ | **PSNR**         |
> |-------------|----------------|------------------|------------------|---------------|------------------|------------------|----------------|------------------|------------------|---------------|------------------|------------------|-----------|
> | 0.04        | 319.54         | 0.1298           | 0.6056           | 268.58        | 0.2578           | 0.6138           | 170.37         | 0.3040           | 0.4603           | 247.31        | 0.1090           | 0.5602           | 35.2832   |
> | **0.06 (AdvPaint)** | **347.88**     | **0.0570**       | **0.6731**       | **289.63**    | **0.1536**       | **0.6762**       | **219.07**     | **0.2148**       | **0.5064**       | **303.90**    | **0.0936**       | **0.6105**       | **32.3779** |
> | 0.08        | 368.37         | 0.0320           | 0.7446           | 311.58        | 0.0992           | 0.7447           | 250.44         | 0.1630           | 0.5506           | 330.50        | 0.0782           | 0.6575           | 29.9798   |
> | 0.1         | 376.69         | 0.0226           | 0.7846           | 326.70        | 0.0642           | 0.7829           | 266.12         | 0.1432           | 0.5780           | 340.51        | 0.0818           | 0.6831           | 28.3171   |
>
>
> | *Iter. Steps*    | FG, $m^{seg}$, FID ↑ | FG, $m^{seg}$, Prec. ↓ | FG, $m^{seg}$, LPIPS ↑ | FG, $m^{bb}$, FID ↑ | FG, $m^{bb}$, Prec. ↓ | FG, $m^{bb}$, LPIPS ↑ | BG, $m^{seg}$, FID ↑ | BG, $m^{seg}$, Prec. ↓ | BG, $m^{seg}$, LPIPS ↑ | BG, $m^{bb}$, FID ↑ | BG, $m^{bb}$, Prec. ↓ | BG, $m^{bb}$, LPIPS ↑ |
> |---------------------|----------------|------------------|------------------|---------------|------------------|------------------|----------------|------------------|------------------|---------------|------------------|------------------|
> | 50                 | 336.39         | 0.0826           | 0.6575           | 284.00        | 0.1872           | 0.6650           | 197.29         | 0.2736           | 0.4894           | 274.99        | 0.1122           | 0.5942           |
> | 100                | 343.72         | 0.0728           | 0.6720           | 287.60        | 0.1744           | 0.6781           | 207.59         | 0.2308           | 0.5087           | 296.83        | 0.0894           | 0.6082           |
> | 150                | 339.79         | 0.0794           | 0.6598           | 285.29        | 0.1898           | 0.6654           | 204.49         | 0.2672           | 0.4958           | 293.95        | 0.1178           | 0.5974           |
> | **250 (AdvPaint)** | **347.88**     | **0.0570**       | **0.6731**       | **289.63**    | **0.1536**       | **0.6762**       | **219.07**     | **0.2148**       | **0.5064**       | **303.90**    | **0.0936**       | **0.6105**       |

---

> ### Author Response · Authors · 2024-11-21
>
> # W3) Application to Diffusion Transformers
>
> As the reviewer suggested, we evaluated the robustness of AdvPaint against an adversary using the inpainting model of Flux[1] and text-to-image model Pixart-$\delta$[2] where both  methods leverage diffusion transformers. Unlike [2], we note that models like DiT[3] and Pixart-$\alpha$[4] are designed for generating images solely from text prompts using diffusion transformer architectures, which make them unsuitable for our tasks that require accepting input images.
>
> ## Flux Inpainting Model
> *Flux* provides an inpainting module based on multimodal and parallel diffusion transformer blocks. We utilized the “black-forest-labs/FLUX.1-schnell” checkpoint, and the image size was set to 512x512 to match our settings.
>
> As shown in Figure 12 in Appendix A.3.3, **_AdvPaint_ effectively disrupts the inpainting process by causing misalignment between generated regions and unmasked areas.** For example, it generates cartoon-style cows in (a) and adds a new rabbit in (b), while also producing noisy patterns in the unmasked areas of the images.
>
> ## DreamBooth with DiT
> Chen et al. introduced *Pixart-$\delta$*, which integrates DreamBooth [5] into DiT. We selected this model as the adversary’s generative framework since it supports feeding an input image along with a command prompt for performing generation. Additionally, as discussed in Section A.3.2, we demonstrate the transferability of AdvPaint to text-to-image tasks.
>
> As shown in Fig. 13 (a) in Appendix A.3.3, **AdvPaint-generated perturbations consistently undermine the generation ability of Pixart-$\delta$.** Furthermore, AdvPaint also renders noise patterns that degrade the image quality on the resulting output images of the diffusion model, which aligns with the behavior of previous methods (i.e. Photoguard, AdvDM, SDST).
>
> ## Protection for DiT-based Models
> AdvPaint also effectively disrupts the original DreamBooth[5], as depicted in Fig. 13 (b). However, our findings indicate that AdvPaint and the previous methods are less effective against Pixart-$\delta$ that leverages DiT (Fig. 13 (a)). Additionally, compared to the results of LDM-based inpainting models in Figure 1, current methods are less effective when applied to DiTs. For example, in AdvPaint, discernible objects appear in the foreground inpainting tasks and new objects are not always generated in the background inpainting tasks. **We believe this ineffectiveness stems from the distinct characteristic of DiT, which processes patchified latent representations.** AdvPaint and our baselines are specifically designed to target LDMs, which utilize the entire latent representation as input, allowing perturbations to be optimized over the complete latent space. Thus, when latents are patchified in DiTs, perturbations may become less effective at disrupting the model's processing, thereby diminishing their protective capability. This discrepancy necessitates further research to develop protection methods specifically tailored to safeguard images against the adversary misusing DiT-based models. For instance, optimizing perturbations at the patch level rather than across the entire latent representation could prove more effective in countering the unique paradigm of image generation in DiT-based models.
>
> [1] Black Forest Labs. (n.d.). Home. Retrieved November 21, 2024, from https://blackforestlabs.ai/
>
> [2] Chen, J., Wu, Y., Luo, S., Xie, E., Paul, S., Luo, P., Zhao, H., & Li, Z. (2024). PIXART-δ: Fast and Controllable Image Generation with Latent Consistency Models. ArXiv, abs/2401.05252.
>
> [3] Peebles, W.S., & Xie, S. (2022). Scalable Diffusion Models with Transformers. 2023 IEEE/CVF International Conference on Computer Vision (ICCV), 4172-4182.
>
> [4] Chen, J., Yu, J., Ge, C., Yao, L., Xie, E., Wu, Y., Wang, Z., Kwok, J.T., Luo, P., Lu, H., & Li, Z. (2023). PixArt-α: Fast Training of Diffusion Transformer for Photorealistic Text-to-Image Synthesis. ArXiv, abs/2310.00426.
>
> [5] Ruiz, N., Li, Y., Jampani, V., Pritch, Y., Rubinstein, M., & Aberman, K. (2022). DreamBooth: Fine Tuning Text-to-Image Diffusion Models for Subject-Driven Generation. 2023 IEEE/CVF Conference on Computer Vision and Pattern Recognition (CVPR), 22500-22510.

---

> ### Author Response · Authors · 2024-11-21
>
> # Experimental settings and baseline models
> Please note that experiments were conducted under the same settings as in the main paper (i.e., using the Stable Diffusion Inpainting model with a total of 100 images, 50 prompts per image where we use only a single seed for a single prompt), ensuring a fair and consistent comparison.
>
> Additionally, we chose the baseline models—Photoguard, AdvDM, and SDST—that employ one of three existing objective functions (i.e. minimize Eq.2, maximize Eq.2, minimize Eq.4) from prior protection methods mentioned in the main paper.

---

> ### Comment · Reviewer_37iq · 2024-11-26
>
> The author's response addressed all my concerns. Therefore I will raise the score to 8.

---

> > ### Author Response · Authors · 2024-11-26
> >
> > Dear Reviewer 37iq,
> >
> > Thank you for your thoughtful review and for taking the time to reassess our work. We greatly appreciate your constructive feedback and engagement in the review process.

---

### Official Review · Reviewer_5BhD · 2024-11-02

**Soundness:** 2
**Presentation:** 2
**Contribution:** 2
**Rating:** 6
**Confidence:** 4

**Summary:**

The authors propose ADVPAINT, a defensive framework that generates adversarial perturbations to mitigate inpainting abuses of diffusion models, such as replacing a specific region with a celebrity. ADVPAINT targets the self- and cross-attention blocks in a target diffusion inpainting model and employs a two-stage perturbation strategy which divides the perturbation region based on an enlarged bounding box around the object. Experimental results demonstrate that ADVPAINT’s perturbations can disrupt the adversary's inpainting tasks.

**Strengths:**

1. The inpainting abuse of diffusion models studied in this paper is an important problem.

2. The authors propose a method for disrupting inpainting tasks, while previous works mainly focused on image-to-image tasks or text-to-image tasks.

**Weaknesses:**

1. It's unclear whether ADVPAINT will be effective if some countermeasures against adversarial perturbations, such as Gaussian noise, JPEG compression and super-resolution, are applied to the perturbed images.

2. Lack of important details in the experiments.

(1) ADVPAINT uses a set of masks to optimize perturbation, but the paper does not specify whether $m^{out}$ in Section 5.6 exceeds all masks to optimize.

(2) Section A.2.2 shows different performances when using different models to optimize perturbations. However, the model used to perform inpainting remains unknown.

(3) Image-to-image tasks and text-to-image tasks are not specified in the paper, so it's confusing how the experiments in Figure 7 and Figure 8 are conducted.

3. There is no comparison of ADVPAINT's performance when models to optimize perturbation and models to perform inpainting are different, which means that most experiments may be conducted as white-box attacks. It is impractical to mainly consider the white-box settings.

**Questions:**

1. Will ADVPAINT be effective if perturbed images are processed with Gaussian noise, JPEG compression or super-resolution before performing inpainting?

2. Does $m^{out}$ in Section 5.6 mean the generated segmentation masks used to perform inpainting exceed all masks used to optimize perturbation?

3. Does Section A.2.2 use the same model to perform inpainting as that to optimize perturbation?

4. Can the authors detail the experiments of Figure 7 and Figure 8?

5. Do the authors only use one version of the inpainting model in their main experiments?

---

> ### Author Response · Authors · 2024-11-21
>
> Dear Reviewer 5BhD,
>
> We are truly grateful for your thoughtful comments and constructive feedback on our manuscript. Below, we have addressed each of your questions and concerns in detail. Should there be any additional points or suggestions that we may have missed, we would be happy to address them. Please do not hesitate to share further feedback with us.
>
> # W1, Q1) Resistance against purification methods
> We thank the reviewer for raising this concern regarding the robustness of AdvPaint against the purification methods designed to remove adversarial perturbations. To address this, we conducted experiments to evaluate the robustness of AdvPaint against the recent purification techniques, including IMPRESS [1] and Honig et al. [2]. Please note that among the four suggested methods in [2], we evaluated two methods for which official code is available—Gaussian noise addition and upscaling—while the others could not be tested due to the lack of accessible implementations. For the purification methods, we follow the Pytorch implementation for JPEG compression with quality 15 and official codes for other methods where Gaussian noise strength is set to 0.05.
>
>
> ## IMPRESS
> Our results show that **_AdvPaint_ retains its protective ability even against IMPRESS, outperforming baseline methods in terms of FID, Precision, and LPIPS.** Since IMPRESS uses LPIPS loss to ensure the purified image remains visually close to the perturbed image, we believe this objective inadvertently preserves a part of the adversarial perturbation. While the protection effectiveness of AdvPaint has decreased compared to Table 1 in the main paper, the residual perturbation appears to maintain sufficient protection against malicious inpainting, as evidenced by the continued high FID values. Additional qualitative results and quantitative evaluations have been provided in Fig. 16 and Table 8 of Appendix A.5.
>
> | *IMPRESS*     | FG, $m^{seg}$, FID ↑ | FG, $m^{seg}$, Prec. ↓ | FG, $m^{seg}$, LPIPS ↑ | FG, $m^{bb}$, FID ↑ | FG, $m^{bb}$, Prec. ↓ | FG, $m^{bb}$, LPIPS ↑ | BG, $m^{seg}$, FID ↑ | BG, $m^{seg}$, Prec. ↓ | BG, $m^{seg}$, LPIPS ↑ | BG, $m^{bb}$, FID ↑ | BG, $m^{bb}$, Prec. ↓ | BG, $m^{bb}$, LPIPS ↑ | **PSNR**         |
> |-------------|----------------|------------------|------------------|---------------|------------------|------------------|----------------|------------------|------------------|---------------|------------------|------------------|---------------------|
> | Photoguard  | 182.62         | 0.6510           | 0.5564           | 151.05        | 0.7990           | 0.5522           | 106.54         | 0.4954           | 0.4333           | 118.30        | 0.2158           | 0.5361           | **28.5925**             |
> | AdvDM       | 209.21         | 0.3764           | 0.5387           | 165.99        | 0.5708           | 0.5336           | 84.63          | 0.6132           | 0.3351           | 103.76        | 0.2734           | 0.4429           | **29.1283**             |
> | SDST        | 199.28         | 0.6252           | 0.5307           | 164.13        | 0.7432           | 0.5271           | 104.75         | 0.4852           | 0.4124           | 121.10        | 0.2130           | 0.5090           | **28.8105**             |
> | **AdvPaint**| **299.07**     | **0.1614**       | **0.6667**       | **237.05**    | **0.3300**       | **0.6623**       | **161.24**     | **0.3230**       | **0.4730**       | **214.38**    | **0.1360**       | **0.5756**       | **28.6303**         |
>
> [1] Cao, B., Li, C., Wang, T., Jia, J., Li, B., & Chen, J. (2023). IMPRESS: Evaluating the Resilience of Imperceptible Perturbations Against Unauthorized Data Usage in Diffusion-Based Generative AI. ArXiv, abs/2310.19248.
>
> [2] Honig, R., Rando, J., Carlini, N., & Tramèr, F. (2024). Adversarial Perturbations Cannot Reliably Protect Artists From Generative AI. ArXiv, abs/2406.12027.

---

> ### Author Response · Authors · 2024-11-21
>
> ## Gaussian Noise, Honig et al., and JPEG Compression
> We calculate the FID, precision, and LPIPS metrics evaluated after applying purification processes, including Gaussian noise addition, upscaling, and JPEG compression. Tables are demonstrated in the *next comment*.
>
> We observed that both AdvPaint and the previous methods lose their ability to protect images when subjected to Gaussian noise addition, upscaling, and JPEG compression. As shown in the below tables, the FID, Precision, and LPIPS scores indicate significant degradation in protection under these conditions.
>
> However, as depicted in the Fig.17 (a) Gaussian noise addition (c) JPEG compression, **the inpainted results are noisy and blurry** (e.g. noisy backgrounds for (a) “sunflower” and (c) “bicycle” images) compared to images generated from non-protected input. This raises concerns about their visual quality. It calls into question the practicality of noise-erasing methods, as the generated images often fail to meet acceptable quality standards.
>
> Additionally, we observed a critical drawback in the existing purification methods: **they tend to degrade the quality of the purified image itself.** For instance, AdvPaint and baseline methods in our experiments leveraged PGD with $\eta$=0.06, ensuring adversarial examples retained a PSNR around 32 dB (Table 12 in the Appendix). On the other hand, after purification (e.g., via upscaling), we observed a PSNR drop of approximately 2.5 dB for AdvPaint, with similar reductions observed for other methods. **This decline highlights a significant trade-off between the purification effectiveness and input image quality.** Additional qualitative results and quantitative evaluations have been provided in Fig.17 and Table 9, 10, and 11 of Appendix A.5.
>
> Based on these findings, we argue that while establishing the robustness against purification methods is indeed critical, the quality degradation of purified images should also be considered. Therefore, the protection capability of AdvPaint is valuable since it increases the adversary’s cost in abusing AdvPaint-perturbed images. We hope this addresses the reviewer’s concern and emphasizes the practical significance of our work.

---

> ### Author Response · Authors · 2024-11-21
>
> Here, we present the FID, precision, and LPIPS metrics evaluated after applying purification methods of Gaussian noise, upscaling, and JPEG compression.
>
> | *Gaussian*     | FG, $m^{seg}$, FID ↑ | FG, $m^{seg}$, Prec. ↓ | FG, $m^{seg}$, LPIPS ↑ | FG, $m^{bb}$, FID ↑ | FG, $m^{bb}$, Prec. ↓ | FG, $m^{bb}$, LPIPS ↑ | BG, $m^{seg}$, FID ↑ | BG, $m^{seg}$, Prec. ↓ | BG, $m^{seg}$, LPIPS ↑ | BG, $m^{bb}$, FID ↑ | BG, $m^{bb}$, Prec. ↓ | BG, $m^{bb}$, LPIPS ↑ | **PSNR**         |
> |-------------|----------------|------------------|------------------|---------------|------------------|------------------|----------------|------------------|------------------|---------------|------------------|------------------|------------------|
> | Photoguard  | 185.20         | 0.6808           | 0.8665           | 156.79        | 0.7814           | 0.8382           | 127.17         | 0.4322           | 0.6111           | 136.26        | 0.1958           | **0.7659**           | **20.1484**      |
> | AdvDM       | 181.57         | 0.6730           | 0.8343           | 152.97        | 0.7864           | 0.8094           | 120.80         | 0.4460           | 0.5896           | 128.89        | 0.2084           | 0.7387           | **20.2824**      |
> | SDST        | 185.04         | 0.6810           | 0.8507           | 154.38        | 0.7838           | 0.8228           | 123.37         | 0.4332           | 0.6006           | 135.07        | 0.2104           | 0.7546           | **20.2358**      |
> | **AdvPaint**| **187.48**     | **0.6682**       | **0.8697**       | **157.74**    | **0.7804**       | **0.8411**       | **128.94**     | **0.4056**       | **0.6125**       | **139.56**    | **0.1820**       | 0.7618       | **20.2410**      |
>
>
> | *Upscaling*     | FG, $m^{seg}$, FID ↑ | FG, $m^{seg}$, Prec. ↓ | FG, $m^{seg}$, LPIPS ↑ | FG, $m^{bb}$, FID ↑ | FG, $m^{bb}$, Prec. ↓ | FG, $m^{bb}$, LPIPS ↑ | BG, $m^{seg}$, FID ↑ | BG, $m^{seg}$, Prec. ↓ | BG, $m^{seg}$, LPIPS ↑ | BG, $m^{bb}$, FID ↑ | BG, $m^{bb}$, Prec. ↓ | BG, $m^{bb}$, LPIPS ↑ | **PSNR**         |
> |-------------|----------------|------------------|------------------|---------------|------------------|------------------|----------------|------------------|------------------|---------------|------------------|------------------|----------------|
> | Photoguard  | 136.96         | 0.8042           | 0.2476           | 111.39        | **0.8820**           | 0.2562           | 60.49          | 0.8086           | 0.2639           | 62.65         | 0.5630           | 0.2842           | **30.2422**    |
> | AdvDM       | **137.97**         | 0.8078           | **0.3112**           | **115.98**        | 0.8844           | **0.3164**           | 63.14          | 0.7886           | **0.2895**           | 65.65         | 0.5428           | **0.3339**           | **29.5016**    |
> | SDST        | 136.57         | **0.8008**           | 0.2474           | 112.77        | 0.8922           | 0.2576           | 61.18          | 0.7932           | 0.2632           | 64.92         | 0.5442           | 0.2823           | **30.0934**    |
> | AdvPaint    | 137.24         | 0.8132           | 0.2784           | 115.43        | 0.8844           | 0.2851           | **65.18**          | **0.7782**           | 0.2840           | **66.61**         | **0.5376**           | 0.3068           | **29.8244**    |
>
>
> | *JPEG*     | FG, $m^{seg}$, FID ↑ | FG, $m^{seg}$, Prec. ↓ | FG, $m^{seg}$, LPIPS ↑ | FG, $m^{bb}$, FID ↑ | FG, $m^{bb}$, Prec. ↓ | FG, $m^{bb}$, LPIPS ↑ | BG, $m^{seg}$, FID ↑ | BG, $m^{seg}$, Prec. ↓ | BG, $m^{seg}$, LPIPS ↑ | BG, $m^{bb}$, FID ↑ | BG, $m^{bb}$, Prec. ↓ | BG, $m^{bb}$, LPIPS ↑ | **PSNR**         |
> |-------------|----------------|------------------|------------------|---------------|------------------|------------------|----------------|------------------|------------------|---------------|------------------|------------------|----------------|
> | Photoguard  | 178.67         | 0.7146           | 0.3830           | 144.72        | 0.8366           | 0.3790           | 101.84         | 0.5662           | 0.3736           | 117.19        | 0.2880           | 0.3969           | **29.6323**    |
> | AdvDM       | **183.50**         | **0.6800**           | **0.4400**           | **150.11**        | 0.8126           | **0.4318**           | 106.74         | 0.5394           | 0.3782           | 120.36        | **0.2614**           | **0.4134**           | **29.4626**    |
> | SDST        | 179.31         | 0.7214           | 0.3956           | 145.99        | 0.8284           | 0.3914           | 104.13         | 0.5564           | 0.3783           | 118.56        | 0.2710           | 0.4003           | **29.5710**    |
> | AdvPaint    | 183.44         | 0.6894           | 0.4126           | 149.74        | **0.8110**           | 0.4080           | **108.70**         | **0.5150**           | **0.3837**           | **124.74**        | 0.2712           | 0.4084           | **29.6232**    |

---

> ### Author Response · Authors · 2024-11-21
>
> # W2) Experimental Details
> ## Q2) Explanation about $m^{out}$ in Section 5.6
> In Section 5.6, we present the inpainting results for two types of masks: $m^{out}$ and $m^{in}$. Since masks can be hand-crafted by adversaries and may exceed the enlarged bounding box used at the optimization step of AdvPaint, we simulate this scenario by randomly shifting the segmentation mask $m^{seg}$ in four directions (up, down, left, and right) to ensure that at least one boundary of the rectangular optimization box is exceeded. This shifted mask is referred to as $m^{out}$. In contrast, $m^{in}$ is generated by randomly shifting $m^{seg}$ while ensuring the entire mask remains inside the rectangular optimization box, reflecting scenarios where segmentation masks differ but are confined to the bounded region.
>
> Our experiments, as shown in Table 3, demonstrate that **_AdvPaint_ maintains robust performance for both $m^{out}$ and $m^{in}$.** Specifically, it achieves strong performance across key metrics, including FID, precision, and LPIPS scores, effectively safeguarding images under both the scenarios.
>
> ### Evaluation on Masks Exceeding or Overlapping the Optimization Boundary
> To address concerns of real-world inpainting scenarios where masks vary in sizes and shapes, we conducted additional experiments in Fig.24 of Appendix A.8.3 using masks that (a) exceed or (b) overlap with the optimization boundary. Specifically, we visualized inpainting results where foreground masks were applied to regions without objects, simulating adversarial scenarios aimed at generating new objects in the background. These experiments demonstrate that **_AdvPaint_ remains robust in such diverse inpainting cases, generating inpainted images with no discernable objects.** We hope this additional analysis effectively addresses the reviewer’s concerns.
>
> ## Q3) Clarification for Section A.2.2 and Table 4
> ### 1. Section A.2.2
> AdvPaint aims to protect images from potential malicious abuse of various inpainting tasks, where there exists a typical architecture (depicted in Figure 6 in Section A.2.2) that takes three types of input: *original image*, *mask*, and the *masked image*. This model also serves as the baseline for other inpainting models [1, 2, 3, 4]. Thus, to clarify, **we utilize the same inpainting model described in Section A.2.2 to perform both the optimization and inpainting tasks in AdvPaint.**
>
> ### 2. Table 4
> In Table 15 (formerly Table 4 in the pre-updated version), we conduct an experiment to evaluate the impact of replacing the objective functions in our baseline models. Specifically, we use the same Latent Diffusion Model (LDM) as the baselines but substitute their objective functions—replacing Eq. (2) (e.g., AdvDM, CAAT) and Eq. (4) (e.g., Photoguard) with our proposed attention loss (Eq. 8). **In other words, the only difference between the two rows of Table 15 is the targeting model (LDM (top) vs. SD Inpainting model (bottom). The objective function stays the same.** Since attention blocks are also present in this *default LDM*, our attention loss is directly applicable. After optimizing perturbations targeting this LDM, we generate inpainted results using the same Stable Diffusion inpainting model referenced in our main paper.
>
> The results indicate that, while trained with our proposed objective, the perturbations fail to provide effective protection against inpainting tasks, underperforming compared to AdvPaint. Furthermore, compared to rows 2–6 in Table 1 (i.e., baseline models), replacing the formal objective functions with our attention loss does not result in a significant improvement in performance.
>
> We attribute this to the lack of direct targeting of inpainting models, which limits their ability to counter inpainting-specific attacks. **This highlights a key limitation of current protection methods that rely on the _default LDM_ for inpainting tasks and underscores the critical importance of designing defensive methods specifically tailored for such tasks.**
>
> To alleviate any confusion caused by the experimental settings in this section, we have updated the explanations in Appendix A.7 to provide clarity.
>
> [1] Manukyan, H., Sargsyan, A., Atanyan, B., Wang, Z., Navasardyan, S., & Shi, H. (2023). HD-Painter: High-Resolution and Prompt-Faithful Text-Guided Image Inpainting with Diffusion Models. ArXiv, abs/2312.14091.
>
> [2] DreamShaper. (2024). DreamShaper: A model available on CivitAI. Retrieved from https://civitai.com/models/4384/dreamshaper
>
> [3] Xie, S., Zhang, Z., Lin, Z., Hinz, T., & Zhang, K. (2022). SmartBrush: Text and Shape Guided Object Inpainting with Diffusion Model. 2023 IEEE/CVF Conference on Computer Vision and Pattern Recognition (CVPR), 22428-22437.
>
> [4] Tang, L., Ruiz, N., Chu, Q., Li, Y., Holynski, A., Jacobs, D.E., Hariharan, B., Pritch, Y., Wadhwa, N., Aberman, K., & Rubinstein, M. (2023). RealFill: Reference-Driven Generation for Authentic Image Completion. ACM Trans. Graph., 43, 135:1-135:12.

---

> ### Author Response · Authors · 2024-11-21
>
> ## Q4) Experimental Details for Figure 7 and 8
> We apply AdvPaint to both image-to-image and text-to-image diffusion models to demonstrate the transferability of AdvPaint. For image-to-image tasks, we implement the Stable Diffusion image-to-image pipeline using Diffusers (runwayml/stable-diffusion-v1-5). Specifically, we follow the default settings of the pipeline, where inference steps = 50, strength = 0.8, and guidance scale = 7.5.
>
> For text-to-image tasks, we leverage Textual Inversion [1] and follow the official implementation and settings. Specifically, we set the inference steps to 50 and the guidance scale to 7.5. For the input images, where 3 to 5 images are required, we utilized the official dataset of DreamBooth. For both tasks, we used images of 512×512 size and randomly crafted the conditional prompts.
>
> We have updated the above information in Section A.1.3 to clarify the experimental settings for Figure 10 and 11 (formerly Figure 7 and 8). We hope this additional detail resolves confusion regarding how these experiments were conducted.
>
> # W3, Q5) Evaluation on Inpainting Model Variants
> We thank the reviewer for highlighting the concern regarding the use of a single inpainting model in the main experiments and the practicality of considering white-box settings. To address this, we conducted additional experiments on multiple inpainting model variants: HD-Painter [2], DreamShaper [3], and the Stable Diffusion v2 inpainting model [4].
>
> HD-Painter proposes a training-free approach to inpainting and replaces the self-attention block in the Stable Diffusion (SD) inpainting model with a block designed to align the prompt more closely with the generated image. DreamShaper is a fine-tuned version of the SD inpainting model, widely available on public platforms. In all experiments, we followed the default settings as in their respective papers.
>
> We evaluated AdvPaint against these variants using FID, precision, and LPIPS metrics and compared it with other baseline protection methods. Even when the architecture differed significantly (i.e., HD-Painter) or when fine-tuning changed the model parameters (i.e., DreamShaper, [4]), **AdvPaint consistently outperformed earlier protection methods across all metrics.** These results demonstrate the transferability of AdvPaint to different inpainting model variants, providing robust protection against potential attacks. We also provide the corresponding qualitative results in Figure 7,8, and 9 (Appendix A.3.1) for a comprehensive comparison.
>
> We present the quantitative results of HD-Painter in the *current comment* and both DreamShaper and Stable Diffusion v2 inpainting model in the *next comment*.
>
> | *HD-Painter*     | FG, $m^{seg}$, FID ↑ | FG, $m^{seg}$, Prec. ↓ | FG, $m^{seg}$, LPIPS ↑ | FG, $m^{bb}$, FID ↑ | FG, $m^{bb}$, Prec. ↓ | FG, $m^{bb}$, LPIPS ↑ | BG, $m^{seg}$, FID ↑ | BG, $m^{seg}$, Prec. ↓ | BG, $m^{seg}$, LPIPS ↑ | BG, $m^{bb}$, FID ↑ | BG, $m^{bb}$, Prec. ↓ | BG, $m^{bb}$, LPIPS ↑ |
> |-------------|----------------|------------------|------------------|---------------|------------------|------------------|----------------|------------------|------------------|---------------|------------------|------------------|
> | Photoguard  | 153.46         | 0.8552           | 0.5632           | 132.63        | 0.8962           | 0.5446           | 93.46          | 0.5978           | 0.3064           | 127.90        | 0.3246           | 0.4400           |
> | AdvDM       | 155.44         | 0.6180           | 0.4807           | 134.54        | 0.7032           | 0.4707           | 75.85          | 0.7278           | 0.2538           | 109.28        | 0.4738           | 0.3617           |
> | SDST        | 146.85         | 0.8568           | 0.4462           | 128.88        | 0.9038           | 0.4456           | 87.64          | 0.6042           | 0.2896           | 127.85        | 0.3366           | 0.4120           |
> | **AdvPaint**| **178.71**     | **0.5350**       | **0.5770**       | **156.51**    | **0.6276**       | **0.5754**       | **164.10**     | **0.3310**       | **0.3998**       | **264.79**    | **0.1748**       | **0.5232**       |
>
>
> [1] Gal, R., Alaluf, Y., Atzmon, Y., Patashnik, O., Bermano, A.H., Chechik, G., & Cohen-Or, D. (2022). An Image is Worth One Word: Personalizing Text-to-Image Generation using Textual Inversion. ArXiv, abs/2208.01618.
>
> [2] Manukyan, H., Sargsyan, A., Atanyan, B., Wang, Z., Navasardyan, S., & Shi, H. (2023). HD-Painter: High-Resolution and Prompt-Faithful Text-Guided Image Inpainting with Diffusion Models. ArXiv, abs/2312.14091.
>
> [3] DreamShaper. (2024). DreamShaper: A model available on CivitAI. Retrieved from https://civitai.com/models/4384/dreamshaper
>
> [4] Stability AI. (n.d.). Stable Diffusion 2 Inpainting. Retrieved November 21, 2024, from https://huggingface.co/stabilityai/stable-diffusion-2-inpainting

---

> ### Author Response · Authors · 2024-11-21
>
> We present the quantitative results of DreamShaper and Stable Diffusion v2 inpainting model.
>
> | *DreamShaper*     | FG, $m^{seg}$, FID ↑ | FG, $m^{seg}$, Prec. ↓ | FG, $m^{seg}$, LPIPS ↑ | FG, $m^{bb}$, FID ↑ | FG, $m^{bb}$, Prec. ↓ | FG, $m^{bb}$, LPIPS ↑ | BG, $m^{seg}$, FID ↑ | BG, $m^{seg}$, Prec. ↓ | BG, $m^{seg}$, LPIPS ↑ | BG, $m^{bb}$, FID ↑ | BG, $m^{bb}$, Prec. ↓ | BG, $m^{bb}$, LPIPS ↑ |
> |-------------|----------------|------------------|------------------|---------------|------------------|------------------|----------------|------------------|------------------|---------------|------------------|------------------|
> | Photoguard  | 188.08         | 0.7422           | 0.5878           | 157.90        | 0.8544           | 0.5792           | 103.74         | 0.6112           | 0.3361           | 131.61        | 0.3074           | 0.4840           |
> | AdvDM       | 183.74         | 0.5114           | 0.5080           | 152.55        | 0.6540           | 0.5001           | 84.99          | 0.7102           | 0.2846           | 115.41        | 0.4340           | 0.3992           |
> | SDST        | 179.80         | 0.7792           | 0.4682           | 151.38        | 0.8656           | 0.4725           | 98.67          | 0.5954           | 0.3149           | 132.79        | 0.2978           | 0.4446           |
> | **AdvPaint**| **230.53**     | **0.3856**       | **0.6042**       | **186.27**    | **0.5196**       | **0.6092**       | **177.68**     | **0.3160**       | **0.4317**       | **266.85**    | **0.1622**       | **0.5561**       |
>
>
> | *SD v2 Inp.*     | FG, $m^{seg}$, FID ↑ | FG, $m^{seg}$, Prec. ↓ | FG, $m^{seg}$, LPIPS ↑ | FG, $m^{bb}$, FID ↑ | FG, $m^{bb}$, Prec. ↓ | FG, $m^{bb}$, LPIPS ↑ | BG, $m^{seg}$, FID ↑ | BG, $m^{seg}$, Prec. ↓ | BG, $m^{seg}$, LPIPS ↑ | BG, $m^{bb}$, FID ↑ | BG, $m^{bb}$, Prec. ↓ | BG, $m^{bb}$, LPIPS ↑ |
> |-------------|----------------|------------------|------------------|---------------|------------------|------------------|----------------|------------------|------------------|---------------|------------------|------------------|
> | Photoguard  | 239.73         | 0.5102           | 0.6226           | 199.21        | 0.7000           | 0.6071           | 110.33         | 0.4570           | 0.3798           | 126.74        | 0.1712           | 0.5094           |
> | AdvDM       | 249.57         | 0.1902           | 0.5393           | 199.22        | 0.3636           | 0.5246           | 89.66          | 0.5942           | 0.3027           | 114.39        | 0.2610           | 0.4197           |
> | SDST        | 231.96         | 0.5324           | 0.5001           | 201.65        | 0.6756           | 0.4996           | 106.61         | 0.4718           | 0.3569           | 130.00        | 0.1892           | 0.4710           |
> | **AdvPaint**| **325.14**     | **0.0926**       | **0.6452**       | **264.72**    | **0.2160**       | **0.6443**       | **198.32**     | **0.2210**       | **0.4633**       | **267.91**    | **0.0842**       | **0.5756**       |

---

> ### Author Response · Authors · 2024-11-21
>
> # Experimental Settings and Baseline Models
> Please note that experiments were conducted under the same settings as in the main paper (i.e., using the Stable Diffusion Inpainting model with a total of 100 images, 50 prompts per image where we use only a single seed for a single prompt), ensuring a fair and consistent comparison.
>
> Additionally, we chose the baseline models—Photoguard, AdvDM, and SDST—that employ one of three existing objective functions (i.e. minimize Eq.2, maximize Eq.2, minimize Eq.4) from prior protection methods mentioned in the main paper.

---

> ### Author Response · Authors · 2024-11-27
>
> # Reminder for Discussions
>
> Dear Reviewer 5BhD,
>
> We would like to remind you that we are open to further discussions on our submission and additional experiments. Considering the deadline for uploading the revised supplementary materials is tomorrow, we ask if you have any additional questions or concerns that we should address.
>
> Thank you for your time and effort in reviewing our work.

---

> > ### Comment · Reviewer_5BhD · 2024-11-27
> >
> > Thank you for your response, which has dispelled my doubts above, except Q5.
> >
> > In your answer to Q5, you have provided extra experiment results in Table 4, 5, and 6. However, there is only one model mentioned in each table, so in each table, the model to optimize perturbations and the model to repaint perturbed images are not clarified. This is necessary, for the adversary may try different inpainting models to repaint the image and choose the result of the best-performed model, while you could only choose one model to optimize perturbations before releasing the image to the Internet.

---

> > > ### Author Response · Authors · 2024-11-28
> > >
> > > Dear Reviewer 5BhD,
> > >
> > > Thank you for your thoughtful review and for raising this concern regarding Table 4, 5, and 6. In our experiments, the target model for optimizing perturbations in **_AdvPaint_**  is always the **Stable Diffusion inpainting model** (*runwayml/stable-diffusion-inpainting*). However, in Tables 4, 5, and 6, the models used for inpainting inference are **HD-Painter**[1], **DreamShaper**[2], and **Stable Diffusion v2 inpainting model**[3] (*stabilityai/stable-diffusion-2-inpainting*), respectively.
> > >
> > > As discussed in the response to Q5, these models differ from the optimization target either in architecture (e.g., HD-Painter [1]) or model parameters (e.g., DreamShaper [2] and Stable Diffusion v2 inpainting model [3]). This setup aligns with our goal of evaluating AdvPaint's robustness against variant models of inpainting without requiring any prior knowledge of their architectures or parameters.
> > >
> > >
> > > To clarify further, the adversarial perturbations optimized using the Stable Diffusion inpainting model were applied consistently across all variants, ensuring a fair and robust evaluation. We provide the following chart summarizing the models used for optimization and inpainting inference in the corresponding tables:
> > >
> > > | Table Number | Target Model for Optimization     | Model Used for Inpainting Inference         |
> > > |--------------|-----------------------------------|---------------------------------------------|
> > > | Table 4      | Stable Diffusion inpainting model | HD-Painter                                  |
> > > | Table 5      | Stable Diffusion inpainting model | DreamShaper                                 |
> > > | Table 6      | Stable Diffusion inpainting model | Stable Diffusion v2 inpainting model        |
> > >
> > >
> > >
> > > By doing so, we emphasize that AdvPaint does not target any of the three variant models during the optimization stage, demonstrating its effectiveness in a black-box scenario where the adversary cannot adaptively optimize against these variants.
> > >
> > >
> > > We hope this clarification addresses your concerns regarding Table 4, 5, and 6. Thank you again for your thoughtful comments, and we kindly request your consideration of our discussions and experimental results when revising your review scores.
> > >
> > >
> > > [1] Manukyan, H., Sargsyan, A., Atanyan, B., Wang, Z., Navasardyan, S., & Shi, H. (2023). HD-Painter: High-Resolution and Prompt-Faithful Text-Guided Image Inpainting with Diffusion Models. ArXiv, abs/2312.14091.
> > >
> > >
> > > [2] DreamShaper. (2024). DreamShaper: A model available on CivitAI. Retrieved from https://civitai.com/models/4384/dreamshaper
> > >
> > >
> > > [3] Stability AI. (n.d.). Stable Diffusion 2 Inpainting. Retrieved November 21, 2024, from https://huggingface.co/stabilityai/stable-diffusion-2-inpainting

---

> > > > ### Comment · Reviewer_5BhD · 2024-11-28
> > > >
> > > > Thanks for your further clarification! My major concerns have been addressed. Therefore, I have increased the score.

---

> > > > > ### Author Response · Authors · 2024-11-28
> > > > >
> > > > > Dear Reviewer 5BhD,
> > > > >
> > > > > Thank you for your thoughtful review and for taking the time to reassess our work.

---

### Official Review · Reviewer_BUBL · 2024-11-03

**Soundness:** 2
**Presentation:** 3
**Contribution:** 2
**Rating:** 6
**Confidence:** 3

**Summary:**

This paper proposes a method of adding adversarial noise into images, aiming to prevent unauthorized use of datasets (inpainting tasks). Specifically, the method leverages SAM to divide  Images into two regions: foreground and background, then optimizes to disrupt the cross- and self-attention block.

**Strengths:**

1.	Considering the popularity of text-to-image models, the topic of preventing image abuse holds practical significance.

2.	Based on the experiments presented in the paper, the proposed method shows promising results.

**Weaknesses:**

1.	**The discussion with related work is insufficient.** Considering that there is a similar work [1] utilizing adversarial noise to disrupt attention layers, discussion about the technical difference between these methods is necessary.

2.	The proposed method is complex, involving different processes like (1) generating prompts using ChatGPT, (2) using SAM for segmentation. However, **the evaluation experiments are not comprehensive** for these pre-progresses. Is it guaranteed that these preprocessing steps are 100% accurate? Do different methods of prompt generation and segmentation produce the same final results?

3.	**The assumptions may not align with real-world scenarios.** It seems the authors assume that the object divided by SAM is the exactly target for inpainting. In multi-object images, is the inpainting target always the object that SAM segmented, or would users focus on different objects?

4.	**A minor concern**: Though numerous works (AdvDM, Mist, CAAT, SDST) were proposed to prevent unauthorized usage by adding adversarial noise, some works [2,3] point out that the noise generated by these methods can be easily disturbed and lose effectiveness. Given this, I believe that, compared to the protection effectiveness (e.g. FID rise and ACC decline), the resistance against these disturbing works [2,3] is more critical. Otherwise, these works may lack practical significance.


[1] Xu, Jingyao, et al. "Perturbing Attention Gives You More Bang for the Buck: Subtle Imaging Perturbations That Efficiently Fool Customized Diffusion Models." Proceedings of the IEEE/CVF Conference on Computer Vision and Pattern Recognition. 2024.

[2] Cao, Bochuan, et al. "Impress: Evaluating the resilience of imperceptible perturbations against unauthorized data usage in diffusion-based generative ai." Advances in Neural Information Processing Systems 36 (2023): 10657-10677.

[3] Hönig, Robert, et al. "Adversarial Perturbations Cannot Reliably Protect Artists From Generative AI." arXiv preprint arXiv:2406.12027 (2024).

**Questions:**

See Weakness.

---

> ### Author Response · Authors · 2024-11-21
>
> Dear Reviewer BUBL,
>
> We sincerely appreciate your valuable comments and insightful suggestions regarding our manuscript. Below, we provide detailed responses to each of your questions and concerns. If there are any additional points or feedback that we may have overlooked, please feel free to let us know.
>
>
> # W1) Technical Comparison with CAAT
> We appreciate this insightful feedback and clarify the technical differences between AdvPaint and CAAT [1].
>
> CAAT first optimizes its perturbation using PGD by maximizing the objective function of Eq.2 in the main paper, **which does not directly target the attention mechanism of a target LDM.** Then, within the same iteration step, it fine-tines the parameters of key and value of the cross-attention layer of the denoiser. The fundamental motivation behind fine-tuning cross-attention parameters is to make the diffusion model more robust, thereby making the perturbation harder to attack and enabling the generation of a more effective perturbation in the subsequent iteration steps.
>
> However, we question whether fine-tuning the parameters of the denoiser U-Net genuinely disrupts the mapping of text and image features. While CAAT does not explore this, AdvDM [2], which utilizes the same objective function of maximizing Eq.2 but omits the fine-tuning process, shows visually similar results. Specifically, in Figures 10, 11, and 18 of the Appendix, the outcomes of image-to-image, text-to-image, and inpainting tasks for both CAAT and AdvDM exhibit similar trends, often manifesting as noisy and plaid patterns. We believe these similar patterns are primarily driven by the shared loss function, and the fine-tuning process in CAAT has a negligible impact on disrupting the attention mechanism.
>
> In contrast, **_AdvPaint_ directly targets both the cross and self-attention layers**, impairing their representation capabilities without requiring an additional fine-tuning process. Moreover, AdvPaint introduces a two-stage optimization process (Section 4.2), applying different perturbations to the target mask regions (i.e., foreground) and the remaining regions (i.e., background). Thus, this approach, specifically designed for undermining adversarial inpainting tasks, effectively disrupts the adversary’s tasks more reliably than the previous methods, including CAAT and AdvDM.
>
> [1] Xu, J., Lu, Y., Li, Y., Lu, S., Wang, D., & Wei, X. (2024). Perturbing Attention Gives You More Bang for the Buck: Subtle Imaging Perturbations That Efficiently Fool Customized Diffusion Models. 2024 IEEE/CVF Conference on Computer Vision and Pattern Recognition (CVPR), 24534-24543.
>
> [2] Liang, C., Wu, X., Hua, Y., Zhang, J., Xue, Y., Song, T., Xue, Z., Ma, R., & Guan, H. (2023). Adversarial Example Does Good: Preventing Painting Imitation from Diffusion Models via Adversarial Examples. International Conference on Machine Learning.

---

> ### Author Response · Authors · 2024-11-21
>
> # W2) Prompts and masks from different models other than ChatGPT and Grounded SAM
>
> We appreciate the reviewer’s insightful feedback on the preprocessing steps used in our method. To address these concerns, we conducted additional experiments employing alternative resources for prompt generation and mask creation to evaluate the robustness and generalizability of AdvPaint’s protection performance.
>
> For prompt generation, in addition to ChatGPT, we utilized **Claude 3.5 Sonnet** to generate diverse prompts. Specifically, for foreground inpainting, we requested prompts consisting of {noun} phrases (e.g., “a jellyfish,” “a mushroom”), while for background inpainting, we generated prompts with {preposition, location} phrases (e.g., “at a cherry blossom grove,” “at a volcanic crater”) combined with a brief description of the input image (e.g., “A deer at a cherry blossom grove,” “A wolf at a volcanic crater”). This prompt structure is consistent with the main paper.
>
> For mask generation, we replaced Grounded SAM with the zero-shot segmentation method proposed by **Yu et al. [1]**, which employs CLIP [2] to create object masks based on the given prompt. Using the same input images as in the main paper, we ran [1] to generate segmentation masks, selecting one mask for foreground inpainting and using its inverted counterpart for background inpainting. These newly generated prompts and masks were then used with the same Stable Diffusion inpainting model as in the main paper.
>
> In Fig. 22 of Appendix A.8.2, we added qualitative results using this alternative setup. These experiments demonstrate that **_AdvPaint_ retains its protection performance for inpainting tasks, comparable to its performance when using ChatGPT and Grounded SAM.** Additionally, we observed that while the exact outputs might vary due to differences in prompt wording or mask generation, the relative protective effectiveness of AdvPaint against earlier protection methods remains consistent. This highlights the robustness of AdvPaint to diverse inpainting scenarios, even when the preprocessing steps involve varying methods.
>
> Note that, as updated in Fig. 23, we observed that the segmentation results from [1] were generally less accurate compared to those generated by Grounded SAM. Masks often fail to fully cover the object, leading to incorrect inpainting results. We report that Grounded SAM may be more suitable for inpainting tasks, given its demonstrated effectiveness and its adoption in related works such as SDST [3] for generating masks in similar contexts. This reinforces our choice of Grounded SAM as the primary segmentation tool in the main paper, while also validating AdvPaint’s adaptability to alternative segmentation approaches.
>
> [1] Yu, S., Seo, P.H., & Son, J. (2023). Zero-shot Referring Image Segmentation with Global-Local Context Features. 2023 IEEE/CVF Conference on Computer Vision and Pattern Recognition (CVPR), 19456-19465.
>
> [2]  Radford, A., Kim, J.W., Hallacy, C., Ramesh, A., Goh, G., Agarwal, S., Sastry, G., Askell, A., Mishkin, P., Clark, J., Krueger, G., & Sutskever, I. (2021). Learning Transferable Visual Models From Natural Language Supervision. International Conference on Machine Learning.
>
> [3] Xue, H., Liang, C., Wu, X., & Chen, Y. (2023). Toward effective protection against diffusion based mimicry through score distillation. ArXiv, abs/2311.12832.

---

> ### Author Response · Authors · 2024-11-21
>
> # W3) Explanations of AdvPaint and Multi-Object Images
> We appreciate your insightful comments. We will closely explain more about the optimization process of multi-object images as the extension of Section 5.7 and the effectiveness of AdvPaint regardless of which masks are used for the multi-object images.
>
> For the optimization process of multi-object images, we first position ourselves as content owners and select the objects that may be at risk of malicious inpainting modifications. Then, AdvPaint performs PGD optimization for each object using enlarged bounding box masks generated by Grounded SAM. After optimizing for each object, the leftover background regions, where objects potentially at risk do not exist, are also optimized. We have updated Fig. 20 in the Appendix to clarify the masks used to optimize multi-object images, aiding comprehension.
>
> After securing each object, we conducted experiments with a variety of mask types, including single-object masks, masks for other objects, combined-object masks, and their inverted versions. **Figure 21 demonstrates the robustness of _AdvPaint_ for multi-object images.** For example, since AdvPaint optimizes each object individually, it ensures protection for each object, resulting in inpainted images that lack discernable objects in the foreground. Furthermore, AdvPaint is robust to masks that encompass all objects, as shown by the absence of "two cameras" replacing "two dogs". Additionally, the method effectively secures background regions when inverted masks are used for inpainting tasks. These results substantiate the effectiveness of AdvPaint’s per-object protection method, even for complex multi-object scenarios.
>
> We have updated the explanations of multi-object images in Appendix A.8.1. We hope this clarifies our approach and demonstrates AdvPaint’s real-world applicability to multi-object image scenarios.
>
> ## Evaluation on Masks Exceeding or Overlapping the Optimization Boundary
> In the real-world scenario, the purpose of inpainting may differ. To address these concerns, we conducted additional experiments in Fig.24 of Appendix A.8.3 using masks that (a) exceed or (b) overlap with the optimization boundary. Specifically, we visualized inpainting results where foreground masks were applied to regions without objects, simulating adversarial scenarios aimed at generating new objects in the background. These experiments demonstrate that **_AdvPaint_ remains robust in such diverse inpainting cases, generating inpainted images with no discernable objects.** We hope this additional analysis effectively addresses the reviewer’s concerns.

---

> ### Author Response · Authors · 2024-11-21
>
> # W4) Resistance Against Purification Methods
> We thank the reviewer for raising this concern regarding the robustness of AdvPaint against the purification methods designed to remove adversarial perturbations. To address this, we conducted experiments to evaluate the robustness of AdvPaint against the recent purification techniques, including IMPRESS [1] and Honig et al. [2]. Please note that among the four suggested methods in [2], we evaluated two methods for which official code is available—Gaussian noise addition and upscaling—while the others could not be tested due to the lack of accessible implementations. For the purification methods, we follow the Pytorch implementation for JPEG compression with quality 15 and official codes for other methods where Gaussian noise strength is set to 0.05.
>
>
> ## IMPRESS
> Our results show that **_AdvPaint_ retains its protective ability even against IMPRESS, outperforming baseline methods in terms of FID, Precision, and LPIPS.** Since IMPRESS uses LPIPS loss to ensure the purified image remains visually close to the perturbed image, we believe this objective inadvertently preserves a part of the adversarial perturbation. While the protection effectiveness of AdvPaint has decreased compared to Table 1 in the main paper, the residual perturbation appears to maintain sufficient protection against malicious inpainting, as evidenced by the continued high FID values. Additional qualitative results and quantitative evaluations have been provided in Fig. 16 and Table 8 of Appendix A.5.
>
> | *IMPRESS*     | FG, $m^{seg}$, FID ↑ | FG, $m^{seg}$, Prec. ↓ | FG, $m^{seg}$, LPIPS ↑ | FG, $m^{bb}$, FID ↑ | FG, $m^{bb}$, Prec. ↓ | FG, $m^{bb}$, LPIPS ↑ | BG, $m^{seg}$, FID ↑ | BG, $m^{seg}$, Prec. ↓ | BG, $m^{seg}$, LPIPS ↑ | BG, $m^{bb}$, FID ↑ | BG, $m^{bb}$, Prec. ↓ | BG, $m^{bb}$, LPIPS ↑ | **PSNR**         |
> |-------------|----------------|------------------|------------------|---------------|------------------|------------------|----------------|------------------|------------------|---------------|------------------|------------------|---------------------|
> | Photoguard  | 182.62         | 0.6510           | 0.5564           | 151.05        | 0.7990           | 0.5522           | 106.54         | 0.4954           | 0.4333           | 118.30        | 0.2158           | 0.5361           | **28.5925**             |
> | AdvDM       | 209.21         | 0.3764           | 0.5387           | 165.99        | 0.5708           | 0.5336           | 84.63          | 0.6132           | 0.3351           | 103.76        | 0.2734           | 0.4429           | **29.1283**             |
> | SDST        | 199.28         | 0.6252           | 0.5307           | 164.13        | 0.7432           | 0.5271           | 104.75         | 0.4852           | 0.4124           | 121.10        | 0.2130           | 0.5090           | **28.8105**             |
> | **AdvPaint**| **299.07**     | **0.1614**       | **0.6667**       | **237.05**    | **0.3300**       | **0.6623**       | **161.24**     | **0.3230**       | **0.4730**       | **214.38**    | **0.1360**       | **0.5756**       | **28.6303**         |
>
> [1] Cao, B., Li, C., Wang, T., Jia, J., Li, B., & Chen, J. (2023). IMPRESS: Evaluating the Resilience of Imperceptible Perturbations Against Unauthorized Data Usage in Diffusion-Based Generative AI. ArXiv, abs/2310.19248.
>
> [2] Honig, R., Rando, J., Carlini, N., & Tramèr, F. (2024). Adversarial Perturbations Cannot Reliably Protect Artists From Generative AI. ArXiv, abs/2406.12027.

---

> ### Author Response · Authors · 2024-11-21
>
> ## Gaussian Noise, Honig et al., and JPEG Compression
> We calculate the FID, precision, and LPIPS metrics evaluated after applying purification processes, including Gaussian noise addition, upscaling, and JPEG compression. Tables are demonstrated in the *next comment*.
>
> We observed that both AdvPaint and the previous methods lose their ability to protect images when subjected to Gaussian noise addition, upscaling, and JPEG compression. As shown in the below tables, the FID, Precision, and LPIPS scores indicate significant degradation in protection under these conditions.
>
> However, as depicted in the Fig.17 (a) Gaussian noise addition (c) JPEG compression, **the inpainted results are noisy and blurry** (e.g. noisy backgrounds for (a) “sunflower” and (c) “bicycle” images) compared to images generated from non-protected input. This raises concerns about their visual quality. It calls into question the practicality of noise-erasing methods, as the generated images often fail to meet acceptable quality standards.
>
> Additionally, we observed a critical drawback in the existing purification methods: **they tend to degrade the quality of the purified image itself.** For instance, AdvPaint and baseline methods in our experiments leveraged PGD with $\eta$=0.06, ensuring adversarial examples retained a PSNR around 32 dB (Table 12 in the Appendix). On the other hand, after purification (e.g., via upscaling), we observed a PSNR drop of approximately 2.5 dB for AdvPaint, with similar reductions observed for other methods. **This decline highlights a significant trade-off between the purification effectiveness and input image quality.** Additional qualitative results and quantitative evaluations have been provided in Fig.17 and Table 9, 10, and 11 of Appendix A.5.
>
> Based on these findings, we argue that while establishing the robustness against purification methods is indeed critical, the quality degradation of purified images should also be considered. Therefore, the protection capability of AdvPaint is valuable since it increases the adversary’s cost in abusing AdvPaint-perturbed images. We hope this addresses the reviewer’s concern and emphasizes the practical significance of our work.

---

> ### Author Response · Authors · 2024-11-21
>
> Here, we present the FID, precision, and LPIPS metrics evaluated after applying purification methods of Gaussian noise, upscaling, and JPEG compression.
>
> | *Gaussian*     | FG, $m^{seg}$, FID ↑ | FG, $m^{seg}$, Prec. ↓ | FG, $m^{seg}$, LPIPS ↑ | FG, $m^{bb}$, FID ↑ | FG, $m^{bb}$, Prec. ↓ | FG, $m^{bb}$, LPIPS ↑ | BG, $m^{seg}$, FID ↑ | BG, $m^{seg}$, Prec. ↓ | BG, $m^{seg}$, LPIPS ↑ | BG, $m^{bb}$, FID ↑ | BG, $m^{bb}$, Prec. ↓ | BG, $m^{bb}$, LPIPS ↑ | **PSNR**         |
> |-------------|----------------|------------------|------------------|---------------|------------------|------------------|----------------|------------------|------------------|---------------|------------------|------------------|------------------|
> | Photoguard  | 185.20         | 0.6808           | 0.8665           | 156.79        | 0.7814           | 0.8382           | 127.17         | 0.4322           | 0.6111           | 136.26        | 0.1958           | **0.7659**           | **20.1484**      |
> | AdvDM       | 181.57         | 0.6730           | 0.8343           | 152.97        | 0.7864           | 0.8094           | 120.80         | 0.4460           | 0.5896           | 128.89        | 0.2084           | 0.7387           | **20.2824**      |
> | SDST        | 185.04         | 0.6810           | 0.8507           | 154.38        | 0.7838           | 0.8228           | 123.37         | 0.4332           | 0.6006           | 135.07        | 0.2104           | 0.7546           | **20.2358**      |
> | **AdvPaint**| **187.48**     | **0.6682**       | **0.8697**       | **157.74**    | **0.7804**       | **0.8411**       | **128.94**     | **0.4056**       | **0.6125**       | **139.56**    | **0.1820**       | 0.7618       | **20.2410**      |
>
>
> | *Upscaling*     | FG, $m^{seg}$, FID ↑ | FG, $m^{seg}$, Prec. ↓ | FG, $m^{seg}$, LPIPS ↑ | FG, $m^{bb}$, FID ↑ | FG, $m^{bb}$, Prec. ↓ | FG, $m^{bb}$, LPIPS ↑ | BG, $m^{seg}$, FID ↑ | BG, $m^{seg}$, Prec. ↓ | BG, $m^{seg}$, LPIPS ↑ | BG, $m^{bb}$, FID ↑ | BG, $m^{bb}$, Prec. ↓ | BG, $m^{bb}$, LPIPS ↑ | **PSNR**         |
> |-------------|----------------|------------------|------------------|---------------|------------------|------------------|----------------|------------------|------------------|---------------|------------------|------------------|----------------|
> | Photoguard  | 136.96         | 0.8042           | 0.2476           | 111.39        | **0.8820**           | 0.2562           | 60.49          | 0.8086           | 0.2639           | 62.65         | 0.5630           | 0.2842           | **30.2422**    |
> | AdvDM       | **137.97**         | 0.8078           | **0.3112**           | **115.98**        | 0.8844           | **0.3164**           | 63.14          | 0.7886           | **0.2895**           | 65.65         | 0.5428           | **0.3339**           | **29.5016**    |
> | SDST        | 136.57         | **0.8008**           | 0.2474           | 112.77        | 0.8922           | 0.2576           | 61.18          | 0.7932           | 0.2632           | 64.92         | 0.5442           | 0.2823           | **30.0934**    |
> | AdvPaint    | 137.24         | 0.8132           | 0.2784           | 115.43        | 0.8844           | 0.2851           | **65.18**          | **0.7782**           | 0.2840           | **66.61**         | **0.5376**           | 0.3068           | **29.8244**    |
>
>
> | *JPEG*     | FG, $m^{seg}$, FID ↑ | FG, $m^{seg}$, Prec. ↓ | FG, $m^{seg}$, LPIPS ↑ | FG, $m^{bb}$, FID ↑ | FG, $m^{bb}$, Prec. ↓ | FG, $m^{bb}$, LPIPS ↑ | BG, $m^{seg}$, FID ↑ | BG, $m^{seg}$, Prec. ↓ | BG, $m^{seg}$, LPIPS ↑ | BG, $m^{bb}$, FID ↑ | BG, $m^{bb}$, Prec. ↓ | BG, $m^{bb}$, LPIPS ↑ | **PSNR**         |
> |-------------|----------------|------------------|------------------|---------------|------------------|------------------|----------------|------------------|------------------|---------------|------------------|------------------|----------------|
> | Photoguard  | 178.67         | 0.7146           | 0.3830           | 144.72        | 0.8366           | 0.3790           | 101.84         | 0.5662           | 0.3736           | 117.19        | 0.2880           | 0.3969           | **29.6323**    |
> | AdvDM       | **183.50**         | **0.6800**           | **0.4400**           | **150.11**        | 0.8126           | **0.4318**           | 106.74         | 0.5394           | 0.3782           | 120.36        | **0.2614**           | **0.4134**           | **29.4626**    |
> | SDST        | 179.31         | 0.7214           | 0.3956           | 145.99        | 0.8284           | 0.3914           | 104.13         | 0.5564           | 0.3783           | 118.56        | 0.2710           | 0.4003           | **29.5710**    |
> | AdvPaint    | 183.44         | 0.6894           | 0.4126           | 149.74        | **0.8110**           | 0.4080           | **108.70**         | **0.5150**           | **0.3837**           | **124.74**        | 0.2712           | 0.4084           | **29.6232**    |

---

> ### Author Response · Authors · 2024-11-21
>
> # Experimental Settings and Baseline Models
> Please note that experiments were conducted under the same settings as in the main paper (i.e., using the Stable Diffusion Inpainting model with a total of 100 images, 50 prompts per image where we use only a single seed for a single prompt), ensuring a fair and consistent comparison.
>
> Additionally, we chose the baseline models—Photoguard, AdvDM, and SDST—that employ one of three existing objective functions (i.e. minimize Eq.2, maximize Eq.2, minimize Eq.4) from prior protection methods mentioned in the main paper.

---

> > ### Comment · Reviewer_BUBL · 2024-11-26
> >
> > Thank you for the feedback, which has addressed almost all of my concerns.
> >
> > Additionally, I have a minor question: In Equation (7), based on my understanding of Stable Diffusion, the input to \\( Q(\\cdot) \\) comes from the image, while the inputs to \\( K(\\cdot) \\) and \\( V(\\cdot) \\) come from the text.
> > This indicates that the \\( x \\) in different terms in Equation (7) are not always consistent, as they actually originate from both text and image inputs.
> > In this case, could the formulation of Equation (7) be improved to make it clearer?

---

> > > ### Author Response · Authors · 2024-11-26
> > >
> > > Dear Reviewer BUBL,
> > >
> > > We are pleased to hear that most of your concerns have been addressed. Regarding Equation (7), we clarify the distinctions between the cross-attention and self-attention mechanisms in Stable Diffusion.
> > >
> > > As you stated, in the **cross-attention** blocks, only the *query* originates from the image features, while the *key* and *value* stem from the conditioning prompt. Accordingly, AdvPaint targets only the *query* in the cross-attention blocks, as shown in Equation (6).
> > >
> > > However, in the **self-attention** blocks, to which that Equation (7) pertains,  all three components (*query*, *key*, and *value*) are derived from linear transformations of the image features. This is because the self-attention layers do not incorporate the prompt condition as an input. That is, the input image $x$ is the source for $Q(\cdot)$, $K(\cdot)$, and $V(\cdot)$ in these blocks. AdvPaint targets all three components (*query*, *key*, and *value*) in the self-attention block, ensuring consistency since there is no prompt-induced variance.
> > >
> > > We hope this clarification resolves any ambiguities regarding Equation (7). Thank you again for your thoughtful comments, and we kindly request that you consider reflecting our discussions and experimental results in your review scores.

---

> > > > ### Comment · Reviewer_BUBL · 2024-11-26
> > > >
> > > > I'd like to thank the authors for their clarification. I found it was an unintentional misunderstanding on my side.
> > > >
> > > > Taking into account the quality of the rebuttal and the comprehensive experiments in the paper, I will raise my rating to 6.

---

> > > > > ### Author Response · Authors · 2024-11-27
> > > > >
> > > > > Dear Reviewer BUBL,
> > > > >
> > > > > Thank you for your thoughtful review and for taking the time to carefully reassess our work.

---

### Official Review · Reviewer_e2Ae · 2024-11-04

**Soundness:** 3
**Presentation:** 3
**Contribution:** 2
**Rating:** 6
**Confidence:** 4

**Summary:**

The authors propose AdvPaint, a method for crafting adversarial image to degrade the performance of Stable Diffusion inpainting model. The perturbation is crafted by maximizing the distance between the cross-attention queries, and self-attention qkv between the perturbed and clean image. These objectives are applied to both foreground and background masks.
Qualitative results show that AdvPaint effectively confuses the attention mask, leading to incorrect inpainting. And quantitative results show consistent improvement in FID and LPIPS scores over previous methods

**Strengths:**

- The writing is clear and easy to follow, with well-presented method and good coverage of literature review
- The protection effects are interesting: foreground inpainting produces a "nothing in the masked regions" effect, while background inpainting creates a "blind spot" effect, making generated regions repetitive and lacking harmony with the foreground object - This protection effects are novel compared to previous chaotic/noisy effects

**Weaknesses:**

- A potential weakness is that the method was tested only on Stable Diffusion Inpainting; evaluating its performance against Diffusion Transformer architectures (e.g., SD3/Flux) would be interesting, given their different patchify and norm mechanisms.
While crafting perturbations for these models may require significant modification, the authors could first show the result of perturbation crafted by SD Inpainting on these models to show its robustness.
- Another weakness is that the method appears effective only when the foreground inpainting prompt is a noun phrase, or when the background prompt includes a phrase where the noun represents the foreground object (that lead to the repetitive effect). Given that an attacker could easily modify the prompt, it would be interesting to test the method on a wide range of prompts (suggested in questions section)

**Questions:**

- What are the run time and memory requirements to run AdvInpaint?
- What is the condition prompt when crafting the perturbation?
- When testing, if I change the prompt to inpaint the foreground object, with the actual object itself (e.g in figure 5, if I change the prompt to `A person` instead of `A sunflower`), will the "nothing in the masked regions" effect still apply?
- Also, on the background prompt, the notable artifacts will happen when the editing prompt contains the object. I wonder if omitting the object from the inpainting prompt (e.g in figure 4, instead of "A monkey on a rocky slope", I just use the prompt "rocky slope") still affect the quality of the image. The problem is that the attacker can quickly adjust the prompt, so in this case, the noisy artifacts from previous method might be advantageous than AdvInpaint.

---

> ### Author Response · Authors · 2024-11-21
>
> Dear Reviewer e2Ae,
>
> We sincerely appreciate your thoughtful comments and suggestions in reviewing our manuscript. We address each of your questions and concerns individually below. Please let us know if you have any comments/concerns that we have not addressed.
>
> # W1) Application to Diffusion Transformers
> As the reviewer suggested, we evaluated the robustness of AdvPaint against an adversary using the inpainting model of Flux[1] and text-to-image model Pixart-$\delta$[2] where both  methods leverage diffusion transformers. Unlike [2], we note that models like DiT[3] and Pixart-$\alpha$[4] are designed for generating images solely from text prompts using diffusion transformer architectures, which make them unsuitable for our tasks that require accepting input images.
>
> ## Flux Inpainting Model
> *Flux* provides an inpainting module based on multimodal and parallel diffusion transformer blocks. We utilized the “black-forest-labs/FLUX.1-schnell” checkpoint, and the image size was set to 512x512 to match our settings.
>
> As shown in Figure 12 in Appendix A.3.3, **_AdvPaint_ effectively disrupts the inpainting process by causing misalignment between generated regions and unmasked areas.** For example, it generates cartoon-style cows in (a) and adds a new rabbit in (b), while also producing noisy patterns in the unmasked areas of the images.
>
> ## DreamBooth with DiT
> Chen et al. introduced *Pixart-$\delta$*, which integrates DreamBooth [5] into DiT. We selected this model as the adversary’s generative framework since it supports feeding an input image along with a command prompt for performing generation. Additionally, as discussed in Section A.3.2, we demonstrate the transferability of AdvPaint to text-to-image tasks.
>
> As shown in Fig. 13 (a) in Appendix A.3.3, **AdvPaint-generated perturbations consistently undermine the generation ability of Pixart-$\delta$.** Furthermore, AdvPaint also renders noise patterns that degrade the image quality on the resulting output images of the diffusion model, which aligns with the behavior of previous methods (i.e. Photoguard, AdvDM, SDST).
>
> ## Protection for DiT-based Models
> AdvPaint also effectively disrupts the original DreamBooth[5], as depicted in Fig. 13 (b). However, our findings indicate that AdvPaint and the previous methods are less effective against Pixart-$\delta$ that leverages DiT (Fig. 13 (a)). Additionally, compared to the results of LDM-based inpainting models in Figure 1, current methods are less effective when applied to DiTs. For example, in AdvPaint, discernible objects appear in the foreground inpainting tasks and new objects are not always generated in the background inpainting tasks. **We believe this ineffectiveness stems from the distinct characteristic of DiT, which processes patchified latent representations.** AdvPaint and our baselines are specifically designed to target LDMs, which utilize the entire latent representation as input, allowing perturbations to be optimized over the complete latent space. Thus, when latents are patchified in DiTs, perturbations may become less effective at disrupting the model's processing, thereby diminishing their protective capability. This discrepancy necessitates further research to develop protection methods specifically tailored to safeguard images against the adversary misusing DiT-based models. For instance, optimizing perturbations at the patch level rather than across the entire latent representation could prove more effective in countering the unique paradigm of image generation in DiT-based models.
>
>
> [1] Black Forest Labs. (n.d.). Home. Retrieved November 21, 2024, from https://blackforestlabs.ai/
>
> [2] Chen, J., Wu, Y., Luo, S., Xie, E., Paul, S., Luo, P., Zhao, H., & Li, Z. (2024). PIXART-δ: Fast and Controllable Image Generation with Latent Consistency Models. ArXiv, abs/2401.05252.
>
> [3] Peebles, W.S., & Xie, S. (2022). Scalable Diffusion Models with Transformers. 2023 IEEE/CVF International Conference on Computer Vision (ICCV), 4172-4182.
>
> [4] Chen, J., Yu, J., Ge, C., Yao, L., Xie, E., Wu, Y., Wang, Z., Kwok, J.T., Luo, P., Lu, H., & Li, Z. (2023). PixArt-α: Fast Training of Diffusion Transformer for Photorealistic Text-to-Image Synthesis. ArXiv, abs/2310.00426.
>
> [5] Ruiz, N., Li, Y., Jampani, V., Pritch, Y., Rubinstein, M., & Aberman, K. (2022). DreamBooth: Fine Tuning Text-to-Image Diffusion Models for Subject-Driven Generation. 2023 IEEE/CVF Conference on Computer Vision and Pattern Recognition (CVPR), 22500-22510.

---

> ### Author Response · Authors · 2024-11-21
>
> # Q1) Runtime and Memory Requirements
> ## 1. Memory Requirements
> The memory usage of AdvPaint primarily depends on the size of an input image, the number of optimization stages, and the number of time-steps in the diffusion models. At the first optimization iteration step, we observe that the minimum memory storage needed is approximately 20GB VRAM, which mainly comes from the Stable Diffusion inpainting model. All the optimization and experiments for AdvPaint were conducted on an *NVIDIA GeForce RTX 3090 (24GB VRAM)* using a *512x512* image size, with optimization performed for *250 steps per mask* at the initial time step T of the sampling process.
>
> ## 2. Runtime
> For each applied mask, AdvPaint performs PGD for 250 steps, which takes approximately *2 minutes*. With a reasonable number of masks applied to each target image, AdvPaint delivers an efficient runtime, averaging 2 minutes per mask (Min: 110 seconds, Max: 128 seconds).
>
> # Q2) Condition Prompt for Crafting Perturbation
> We would like to refer the reviewer to Section A.1.4 in our supplementary material. The condition prompts were designed to indicate an object in the scene (e.g., "a woman," "a man," "a dog"). This conditioning format was designed to simplify the optimization process, making it easier for the image owners to protect their images easily using a straightforward prompt.
>
> This approach aligns with the settings used by our baseline models: AdvDM and Mist employ prompts such as "A photo" or "A painting," while CAAT utilizes prompts in the format "A photo of [noun]," where the [noun] specifies the image content (e.g., "person" for human images).

---

> ### Author Response · Authors · 2024-11-21
>
> #  W2, Q3) FG Inpainting: prompts describing the object itself
>
> We conducted additional experiments using prompts that specify the object in each given scene. For example, we used the prompt “A man” for an input image describing a male and performed an inpainting task to generate another male image.
>
> **In all cases, AdvPaint successfully disrupted the adversary's inpainting task, resulting in the generation of an image with no discernible object.** The following table describes the foreground inpainting performance of AdvPaint for this set of prompts, outperforming baseline methods in terms of FID, Precision, and LPIPS. This is because the perturbation optimized to disrupt the attention mechanism successfully redirects the attention to other unmasked areas as explained in Section 5.2 and Figure 3.
>
> We included the qualitative examples and experimental results in Appendix A.4.1 (Figure 14) of the revised supplementary material.
>
> | Method      | $m^{seg}$, FID ↑ | $m^{seg}$, Prec. ↓ | $m^{seg}$, LPIPS ↑ | $m^{bb}$, FID ↑ | $m^{bb}$, Prec. ↓ | $m^{bb}$, LPIPS ↑ |
> |-------------|-------------|----------------|----------------|-------------|----------------|----------------|
> | Photoguard  | 161.44      | 0.0874         | 0.6415         | 129.99      | 0.2158         | 0.6171         |
> | AdvDM       | 160.54      | 0.0658         | 0.5167         | 127.36      | 0.1266         | 0.5122         |
> | SDST        | 148.57      | 0.1340         | 0.4930         | 120.55      | 0.2456         | 0.4882         |
> | **AdvPaint**| **331.27**  | **0.0036**     | **0.6706**     | **275.48**  | **0.0264**     | **0.6697**     |
>
>
> # W2, Q4) BG Inpainting: prompts without nouns
>
> We experimented with prompts where the noun describing the object is omitted and evaluated their effectiveness in undermining the adversary’s background inpainting task. Specifically, we assumed the adversary might adjust the prompt to exclude the object (e.g., in Figure 4, using "rocky slope" instead of "A monkey on a rocky slope") to mitigate artifacts. **In all cases, AdvPaint outperformed all baselines**, as demonstrated in the table below and updated in Appendix A.4.2.
>
> Additionally, in the qualitative results included in Fig. 15 of Appendix A.4.2, we observed a consistent similarity across prompts with and without nouns: **in both cases for AdvPaint, the inpainting models fail to recognize the object in the unmasked region.** When using prompts that contain nouns (originally used prompts), the models generate new objects in the masked area based on the noun in the prompt (e.g., in (b) and (d) of Fig. 1), disregarding the presence of the object in the unmasked region. Similarly, when prompts are provided without nouns (Fig. 15), the inpainting models fail to recognize the object, blending it into the unmasked region and often generating background components based on the prompt description (e.g., adding a "horse" to the background for the prompt "in the countryside").
>
> We believe this similarity arises from AdvPaint's targeted disruption of the self- and cross-attention mechanisms in inpainting models, which impairs their ability to maintain semantic relationships and align the generated content with the condition prompt, regardless of whether the prompt includes a noun describing the object. These findings highlight AdvPaint’s robustness in protecting images, as it consistently prevents recognition of objects in the unmasked region, even when the attacker adjusts the prompt to exclude the object.
>
> | Method      | $m^{seg}$, FID ↑ | $m^{seg}$, Prec. ↓ | $m^{seg}$, LPIPS ↑ | $m^{bb}$, FID ↑ | $m^{bb}$, Prec. ↓ | $m^{bb}$, LPIPS ↑ |
> |-------------|-------------|----------------|----------------|-------------|----------------|----------------|
> | Photoguard  | 144.21      | 0.5230         | 0.4063         | 153.63      | 0.2280         | 0.5317         |
> | AdvDM       | 118.13      | 0.6228         | 0.3168         | 131.58      | 0.2720         | 0.4311         |
> | SDST        | 139.86      | 0.5112         | 0.3810         | 152.73      | 0.2280         | 0.4892         |
> | **AdvPaint**| **291.12**  | **0.3490**     | **0.4948**     | **355.94**  | **0.1152**     | **0.6014**     |

---

> ### Author Response · Authors · 2024-11-21
>
> # Experimental Settings and Baseline Models
> Please note that experiments were conducted under the same settings as in the main paper (i.e., using the Stable Diffusion Inpainting model with a total of 100 images, 50 prompts per image where we use only a single seed for a single prompt), ensuring a fair and consistent comparison.
>
> Additionally, we chose the baseline models—Photoguard, AdvDM, and SDST—that employ one of three existing objective functions (i.e. minimize Eq.2, maximize Eq.2, minimize Eq.4) from prior protection methods mentioned in the main paper.

---

> > ### Comment · Reviewer_e2Ae · 2024-11-25
> >
> > I’m pleased that the authors addressed all of my concerns and included results on personalization. Although the performance of DiT-based approaches remains inconsistent, I believe this is generally a good work.

---

> > > ### Author Response · Authors · 2024-11-26
> > >
> > > We sincerely appreciate your acknowledgement of our discussions and contributions. We want to point out that AdvPaint still disrupts the inpainting task using DiT-based models by introducing noise patterns that degrade the image quality of resulting images. We hope that our revisions have addressed your concerns, and we kindly request that you consider reflecting these changes in your updated review scores.

---

> > > > ### Comment · Reviewer_e2Ae · 2024-12-02
> > > >
> > > > I will keep my score of 6, as my questions are primarily for clarification and addressing missing experiments.

---

> > > > > ### Author Response · Authors · 2024-12-04
> > > > >
> > > > > Dear Reviewer e2Ae,
> > > > >
> > > > > Thank you for your insightful feedback and thoughtful engagement throughout the review process.

---

### Author Response · Authors · 2024-11-23

# Gentle Reminder for Discussions


Dear Reviewer,


We appreciate your time and effort for reviewing our work again. We have posted clarifications and additional evaluations results to address your raised concerns. It would be grateful to let us know if there are more questions and concerns that we can address.


Thank you.

---

### Meta-Review · Area_Chair_bGbe · 2024-12-18

**Metareview:**

This paper proposes a novel defense method AdvPaint to generate adversarial perturbations that can disrupt the adversary’s inpainting tasks. The method distracts semantic understanding and prompt interactions by maximizing the distance between the cross-attention queries, and self-attention qkv between the perturbed and clean image. The method shows effectiveness for Stable Diffusion and Diffusion Transformer models.

The paper studies an important problem of avoiding malicious abuse of generative models for inpaining tasks. The paper proposes a novel method to address the problem. The paper is well-written.The reviewers initially raised several concerns about this paper, including effectiveness on Diffusion Transformers, insufficient discussion of related work, effectiveness under defense mechanisms, etc. The authors have provided detailed responses to the concerns and addressed most of the concerns raised by the reviewers. Given the consensus among all the reviewers, the AC recommends the acceptance of this paper.

**Additional Comments On Reviewer Discussion:**

The reviewers raised several points of the paper:
- Reviewer e2Ae pointed out the weaknesses of not evaluating the method on Diffusion Transformer model and foreground inpainting prompt is a noun phrase.
- Reviewer BUBL pointed out the weaknesses that discussion with related work is insufficient, the method is complex without comprehensive experiments, the assumptions may not align with real-world scenarios.
- Reviewer 5BhD pointed out the weaknesses of the effectiveness under countermeasures, lack of experimental details, and lack of comparison results under black-box setting.
- Reviewer 37iq pointed out the issues of fixed devision of target region and background, effect of varying noise levels and PGD steps, and lack of experiments on Diffusion Transformers.

The authors have provided detailed responses to these concerns and revised the paper accordingly. The reviewers were satisfied with the rebuttal and raised their ratings to recommend acceptance of the paper. Thus, AC is willing to recommend acceptance.

---

### Decision · Program_Chairs · 2025-01-22

Accept (Poster)